# Transformers are uninterpretable with myopic methods: a case study with bounded Dyck grammars

**Kaiyue Wen**
Tsinghua University
wenky20@mails.tsinghua.edu.cn

**Yuchen Li**
Carnegie Mellon University
yuchenl4@cs.cmu.edu

**Bingbin Liu**
Carnegie Mellon University
bingbinl@cs.cmu.edu

**Andrej Risteski**
Carnegie Mellon University
aristesk@andrew.cmu.edu

## Abstract

Transformer interpretability aims to understand the algorithm implemented by a learned Transformer by examining various aspects of the model, such as the weight matrices or the attention patterns. In this work, through a combination of theoretical results and carefully controlled experiments on synthetic data, we take a critical view of methods that exclusively focus on individual parts of the model, rather than consider the network as a whole. We consider a simple synthetic setup of learning a (bounded) Dyck language. Theoretically, we show that the set of models that (exactly or approximately) solve this task satisfy a structural characterization derived from ideas in formal languages (the pumping lemma). We use this characterization to show that the set of optima is qualitatively rich; in particular, the attention pattern of a single layer can be "nearly randomized", while preserving the functionality of the network. We also show via extensive experiments that these constructions are not merely a theoretical artifact: even with severe constraints to the architecture of the model, vastly different solutions can be reached via standard training. Thus, interpretability claims based on inspecting individual heads or weight matrices in the Transformer can be misleading.

## 1 Introduction

Transformer-based models power many leading approaches to natural language processing. With their growing deployment in various applications, it is increasingly essential to understand the inner working of these models. Towards addressing this, there have been great advancement in the field of interpretability presenting various types of evidence (Clark et al., 2019; Vig & Belinkov, 2019; Wiegreffe & Pinter, 2019; Nanda et al., 2023; Wang et al., 2023), some of which, however, can be misleading despite being highly intuitive (Jain & Wallace, 2019; Serrano & Smith, 2019; Rogers et al., 2020; Grimsley et al., 2020; Brunner et al., 2020; Meister et al., 2021).

In this work, we aim to understand the theoretical limitation of certain interpretability methods by characterizing the set of viable solutions. We focus on myopic interpretability methods, i.e. methods based on examining individual components only. We adopt a particular toy setup in which Transformers are trained to generate *Dyck grammars*, a classic type of formal language grammar consisting of balanced parentheses of multiple types. Dyck is a useful sandbox, as it captures properties like long-range dependency and hierarchical tree-like structure that commonly appear in natural and programming language syntax, and has been an object of interest in many theoretical studies (Hahn, 2020; Yao et al., 2021; Liu et al., 2022b, 2023). Dyck is canonically parsed using

37th Conference on Neural Information Processing Systems (NeurIPS 2023).

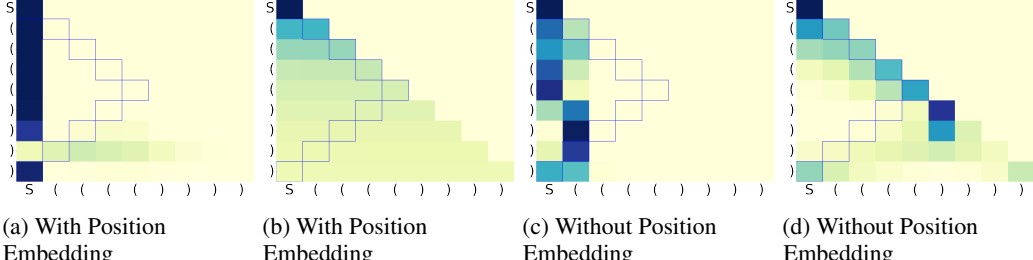

| (a) With Position Embedding | (b) With Position Embedding | (c) Without Position Embedding | (d) Without Position Embedding |

Figure 1: **Second-layer attention patterns of two-layer Transformers on Dyck**: typical attention patterns do *not* exactly match the intuitively interpretable stack-like pattern prescribed in Ebrahimi et al. (2020); Yao et al. (2021). The blue boxes indicate the locations of the last unmatched open brackets, as they would appear in a stack-like pattern. All models reach $\geq 97\%$ accuracy (defined in Section 4.1). In the heatmap, darker color indicates larger value.

a stack-like data structure. Such stack-like patterns (Figure 1) have been observed in the attention heads (Ebrahimi et al., 2020), which was later bolstered by mathematical analysis in Yao et al. (2021).

From a representational perspective and via explicit constructions of Transformer weights, recent work (Liu et al., 2023; Li et al., 2023) show that Transformers are sufficiently expressive to admit very different solutions that perform equally well on the training distribution. Thus, the following questions naturally arise:

(Q1) Do Transformer solutions found empirically match the theoretical constructions given in these representational results (Figure 1)? In particular, are interpretable stack-like pattern in Ebrahimi et al. (2020) the norm or the exception in practice?

(Q2) More broadly, can we understand in a principled manner the fundamental obstructions to reliably "reverse engineering" the algorithm implemented by a Transformer by looking at individual attention patterns?

(Q3) Among models that perform (near-)optimally on the training distribution, even if we cannot fully reverse engineer the algorithm implemented by the learned solutions, can we identify properties that characterize performance beyond the training distribution?

**Our contributions.** We first prove several theoretical results to provide evidence for why individual components (e.g. attention patterns or weights) of a Transformer should not be expected to be interpretable. In particular, we prove:

- A **perfect balance** condition (Theorem 1) on the attention pattern that is sufficient and necessary for 2-layer Transformers with a *minimal first layer* (Assumption 1) to predict optimally on Dyck of *any* length. We then show that this condition permits abundant *non-stack-like* attention patterns that do not necessarily reflect any structure of the task, including *uniform* attentions (Corollary 1).
- An **approximate balance** condition (Theorem 3), the *near-optimal* counterpart of the condition above, for predicting on *bounded*-length Dyck. Likewise, non-stack-like attention patterns exist.
- **Indistinguishability from a single component** (Theorem 2), proved via a *Lottery Ticket Hypothesis* style argument that any Transformer can be approximated by pruning a larger random Transformer, implying that interpretations based exclusively on local components may be unreliable.

We further accompany these theoretical findings with an extensive set of empirical investigations.

*Is standard training biased towards interpretable solutions?* While both stack-like and non-stack like patterns can process Dyck theoretically, the inductive biases of the architecture or the optimization process may prefer one solution over the other in practice. In Section 4.1, based on a wide range of Dyck distributions and model architecture ablations, we find that Transformers that generalize near-perfectly in-distribution (and reasonably well out-of-distribution) do *not* typically produce stack-like attention patterns, showing that the results reported in prior work (Ebrahimi et al., 2020) should not be expected from standard training.

*Do non-interpretable solutions perform well in practice?* Our theory predicts that balanced (or even uniform) attentions suffice for good in- and out-of-distribution generalization. In Section 4.2, we

empirically verify that with standard training, the extent to which attentions are balanced is positively correlated with generalization performance. Moreover, we can guide Transformers to learn more balanced attention by regularizing for the balance condition, leading to better length generalization.

## 1.1 Related Work

There has been a flourishing line of work on interpretability in natural language processing. Multiple "probing" tasks have been designed to extract syntactic or semantic information from the learned representations (Raganato & Tiedemann, 2018; Liu et al., 2019; Hewitt & Manning, 2019; Clark et al., 2019). However, the effectiveness of probing often intricately depend on the architecture choices and task design, and sometimes may even result in misleading conclusions (Jain & Wallace, 2019; Serrano & Smith, 2019; Rogers et al., 2020; Brunner et al., 2020; Prasanna et al., 2020; Meister et al., 2021). While these challenges do not completely invalidate existing approaches (Wiegreffe & Pinter, 2019), it does highlight the need for more rigorous understanding of interpretability.

Towards this, we choose to focus on the synthetic setup of Dyck whose solution space is easier to characterize than natural languages, allowing us to identify a set of feasible solutions. While similar representational results have been studied in prior work (Yao et al., 2021; Liu et al., 2023; Zhao et al., 2023), our work emphasizes that theoretical constructions do not resemble the solutions found in practice. Moreover, the multiplicity of valid constructions suggest that understanding Transformer solutions require analyzing the optimization process, which a number of prior work has made progress on (Jelassi et al., 2022; Li et al., 2023; Deng et al., 2023).

Finally, it is worth noting that the challenges highlighted in our work do not contradict the line of prior work that aim to improve *mechanistic interpretability* into a trained model or the training process (Elhage et al., 2021; Olsson et al., 2022; Nanda et al., 2023; Chughtai et al., 2023; Li et al., 2023), which aim to develop circuit-level understanding of a particular model or the training process.

We defer discussion on additional related work to Appendix A.

## 2 Problem Setup

**Dyck languages** A Dyck language (Schützenberger, 1963) is generated by a context-free grammar, where the valid strings consist of balanced brackets of different types (for example, "[()]" is valid but "([)]" is not). $\mathsf{Dyck}_k$ denote the Dyck language defined on $k$ types of brackets. The alphabet of $\mathsf{Dyck}_k$ is denoted as $[2k] \equiv \{1, 2, \cdots, 2k\}$, where for each type $t \in [k]$, tokens $2t - 1$ and $2t$ are a pair of corresponding open and closed brackets. Dyck languages can be recognized by a push-down automaton. For a string $w$ and $i \leq j \in \mathbb{Z}_+$, we use $w_{i:j}$ to denote the substring of $w$ between position $i$ and position $j$ (both ends included). For a valid prefix $w_{1:i}$, the *grammar depth* of $w_{1:i}$ is defined as the depth of the stack after processing $w_{1:i}$:

$$\mathrm{depth}(w_{1:i}) = \#\text{Open Brackets in } w_{1:i} - \#\text{Closed Brackets in } w_{1:i}.$$

We overload $\mathrm{depth}(w_{1:i})$ to also denote the grammar depth of the bracket at position $i$. For example, in each pair of matching brackets, the closing bracket is one depth smaller than the open bracket. We will use $\tau_{i,d}$ to denote a token of type $i \in [2k]$ placed at grammar depth $d \in \mathbb{N}$.

We consider *bounded-depth* Dyck languages following Yao et al. (2021). Specifically, $\mathsf{Dyck}_{k,D}$ is a subset of $\mathsf{Dyck}_k$ such that the depth of any prefix of a word is bounded by $D$,

$$\mathsf{Dyck}_{k,D} := \{w_{1:n} \in \mathsf{Dyck}_k \mid \max_{i \in [n]} \mathrm{depth}(w_{1:i}) \leq D\}. \tag{1}$$

While a bounded grammar depth might seem restrictive, it suffices to capture many practical settings. For example, the level of recursion occurring in natural languages is typically bounded by a small constant (Karlsson, 2007; Jin et al., 2018). We further define the *length-$N$ prefix set* of $\mathsf{Dyck}_{k,D}$ as

$$\mathsf{Dyck}_{k,D,N} = \{w_{1:N} \mid \exists n \geq N, w_{N+1:n} \in [2k]^{n-N}, s.t. \ w_{1:n} \in \mathsf{Dyck}_{k,D}\}. \tag{2}$$

Our theoretical setup uses the following data distribution $\mathcal{D}_{q,k,D,N}$:

**Definition 1** (Dyck distribution). *The distribution $\mathcal{D}_{q,k,D,N}$, specified by $q \in (0,1)$, is defined over* $\mathsf{Dyck}_{k,D,N}$ *such that* $\forall w_{1:N} \in \mathsf{Dyck}_{k,D,N}$,

$$\mathbb{P}(w_{1:N}) \propto (q/k)^{\#\{i|w_i \text{ is open, depth}(w_{1:i})>1\}} \cdot (1-q)^{\#\{i|w_i \text{ is closed, depth}(w_{1:i})<D-1\}}. \tag{3}$$

That is, $q \in (0, 1)$ denote the probability of seeing an open bracket at the next position, except for two corner cases: 1) the next bracket has to be open if the current grammar depth is 0 (1 after seeing the open bracket); 2) the next bracket has to be closed if the current grammar depth is $D$.

**Training Objectives.** Given a model $f_\theta$ parameterized by $\theta$, we train with a *next-token prediction* language modeling objective on a given $\mathcal{D}_{q,k,D,N}$. Precisely, given a loss function $l(\cdot, \cdot) \to \mathbb{R}$, $f_\theta$ is trained to minimize the loss function $\min_\theta \mathcal{L}(\theta; \mathcal{D}_{q,k,D,N})$ with

$$\mathcal{L}(\theta; \mathcal{D}_{q,k,D,N}) = \mathbb{E}_{w_{1:N} \sim \mathcal{D}_{q,k,D,N}} \Big[ \frac{1}{N} \sum_{i=1}^{N} l(f_\theta(w_{1:i-1}), z(w_i)) \Big] \tag{4}$$

in which $z(w_i) \in \{0, 1\}^{2k}$ denotes the one-hot embedding of token $w_i$. We will omit the distribution $\mathcal{D}_{q,k,D,N}$ when it is clear from the context. We will also consider a $\ell_2$-regularized version $\mathcal{L}^{\text{reg}}(\theta) = \mathcal{L}(\theta) + \lambda \frac{\|\theta\|_2^2}{2}$ with parameter $\lambda > 0$.

For our theory, we will consider the mean squared error as the loss function: [1]

$$l := l_{sq}(x, z_i) = \|x - z_i\|_2^2. \tag{5}$$

In our experiments, we apply the cross entropy loss following common practice.

**Transformer Architecture.** We consider a general formulation of Transformer in this work: the $l$-th layer is parameterized by $\theta^{(l)} := \{W_Q^{(l)}, W_K^{(l)}, W_V^{(l)}, \text{param}(\mathrm{g}^{(l)})\} \in \Theta$, where $W_K^{(l)}, W_Q^{(l)} \in \mathbb{R}^{m_a \times m}$, and $W_V^{(l)} \in \mathbb{R}^{m \times m}$ are the key, query, and value matrices of the attention module; $\text{param}(\mathrm{g}^{(l)})$ are parameters of a feed-forward network $\mathrm{g}^{(l)}$, consisting of fully connected layers, (optionally) LayerNorms and residual links. Given $X \in \mathbb{R}^{m \times N}$, the matrix of $m$-dimensional features on a length-$N$ sequence, the $l$-th layer of a Transformer computes the function

$$f_l(X; \theta^{(l)}) = \mathrm{g}^{(l)} \Big( \mathrm{LN} \Big( W_V^{(l)} X \underbrace{\sigma \Big( \mathcal{C} + (W_K^{(l)} X)^\top (W_Q^{(l)} X) \Big)}_{\text{attention pattern}} \Big) + X \Big), \tag{6}$$

where $\sigma$ is the column-wise softmax operation defined as $\sigma(A)_{i,j} = \frac{\exp(A_{i,j})}{\sum_{k=1}^{N} \exp(A_{k,j})}$, $\mathcal{C}$ is the causal mask matrix defined as $\mathcal{C}_{i,j} = -\inf \cdot \mathbb{1}[i > j]$ where $\inf$ denotes infinity. We call $\sigma \Big( \mathcal{C} + (W_K^{(l)} X)^\top (W_Q^{(l)} X) \Big)$ the *Attention Pattern* of the Transformer layer $l$. LN represents column-wise LayerNorm operation, whose $j_{th}$ output column is defined as

$$\mathrm{LN}_{C_{LN}}(A)_{:,j} = \frac{\mathcal{P}_\perp A_{:,j}}{\max\{\|\mathcal{P}_\perp A_{:,j}\|_2, C_{LN}\}}, \mathcal{P}_\perp = \mathcal{I}_m - \frac{1}{m} \mathbf{1}\mathbf{1}^\top. \tag{7}$$

Here $\mathcal{P}_\perp$ denotes the projection orthogonal to the $\mathbf{1}\mathbf{1}^\top$ subspace [2] and $C_{LN}$ is called the normalizing constant for LayerNorm.

We will further define the *attention output* at the $l$-th layer as

$$a_l(X; \theta^{(l)}) = W_V^{(l)} X \sigma \Big( \mathcal{C} + (W_K^{(l)} X)^\top (W_Q^{(l)} X) \Big). \tag{8}$$

When $C_{LN} = 0$, we will also consider the *unnormalized attention output* as

$$\tilde{a}_l(X; \theta^{(l)}) = W_V^{(l)} X \tilde{\sigma} \Big( \mathcal{C} + (W_K^{(l)} X)^\top (W_Q^{(l)} X) \Big). \tag{9}$$

where $\tilde{\sigma}(A)_{i,j} = \exp(A_{i,j})$ and it holds by definition that $\mathrm{LN}_0(\tilde{a}_l(X; \theta^{(l)})) = \mathrm{LN}_0(a_l(X; \theta^{(l)}))$.

An $L$-layer Transformer $\mathcal{T}_L$ consists of a composition of $L$ of the above layers, along with a word embedding matrix $W_E \in \mathbb{R}^{m \times 2k}$ and a linear decoding head with weight $W_{\text{Head}} \in \mathbb{R}^{2k \times w}$. When

---

[1]The challenge of applying our theory to cross-entropy loss is that for some prefixes, their grammatical immediate continuations strictly exclude certain tokens in the vocabulary (e.g. "]" cannot immediately follow "{"), so the optimal cross-entropy loss can only be attained if some parameters are set to infinity. However, when label smoothing is added, the optima is finite again, and analysis similar to ours could apply.

[2]this is just a compact way to write the standard mean subtraction operation

inputting a sequence of tokens into Transformer, we will append a *starting token* $t_S$ that is distinct from any token in the language at the beginning of the sequence. Let $\mathcal{Z} \in \mathbb{R}^{2k \times (N+1)}$ denote the one-hot embedding of a length-$N$ sequence, then $\mathcal{T}_L$ computes for $\mathcal{Z}$ as

$$\mathcal{T}(\mathcal{Z}) = W_{\text{Head}} \Big[ f_L(\cdots(f_1(W_E \mathcal{Z}))) \Big]_{1:2k,(N+1)}. \tag{10}$$

## 3 Theoretical Analyses

Many prior works have looked for intuitive interpretations of Transformer solutions by studying the attention patterns of particular heads or some individual components of a Transformer (Clark et al., 2019; Vig & Belinkov, 2019; Dar et al., 2022). However, we show in this section why this methodology can be insufficient even for the simple setting of Dyck. Namely, for Transformers that generalize well on Dyck (both in-distribution and out-of-distribution), neither attention patterns nor individual local components are guaranteed to encode structures specific for parsing Dyck. We further argue that the converse is also insufficient: when a Transformer does produce interpretable attention patterns, there could be limitations of such interpretation as well, as discussed in Appendix B. Together, our results provide theoretical evidence that careful analyses (beyond heuristics) are required when interpreting the components of a learned Transformer.

### 3.1 Interpretability Requires Inspecting More Than Attention Patterns

This section focuses on Transformers with 2 layers, which are sufficient for processing Dyck (Yao et al., 2021). We will show that even under this simplified setting, attention patterns alone are not sufficient for interpretation. In fact, we will further restrict the set of 2-layer Transformers by requiring the first-layer outputs to only depend on information necessary for processing Dyck:

**Assumption 1** (Minimal First Layer). *We consider 2-layer Transformers with a* minimal *first layer $f_1$. That is, let $\boldsymbol{Z} \in \mathbb{R}^{2k \times (N+1)}$ denote the one-hot embeddings of any input sequence $t_S, t_1, \ldots, t_N \in [2k]$, then the $(j+1)_{th}$ column of the output $f_1(W^E \boldsymbol{Z})$ only depends on the type and depth of $t_j$, $\forall j \in [N]$.*

Assumption 1 requires the first layer output to depend only on the bracket type and depth, disregarding any other information such as positions; one such example is given by Yao et al. (2021). The construction of a minimal first layer can vary, hence we *directly parameterize its output* instead:

**Definition 2** (Minimal first layer embeddings). *Given a minimal first layer, $\boldsymbol{e}(\tau_{t,d}) \in \mathbb{R}^m$ denotes its output embedding of $\tau_{t,d}$ for $t \in [2k]$, $d \in [D]$. $\boldsymbol{e}(t_S) \in \mathbb{R}^m$ is the embedding of the starting token.*

It is important to note that while the minimal first layer is a strong condition, it does not weaken our results: We will show that the function class allows for a rich set of solutions, none of which are necessarily interpretable. Relaxing to more complex classes will only expand the solution set, and hence our conclusion will remain valid. See Appendix C.2 for more technical details.

#### 3.1.1 Perfect Balance Condition: Ideal Generalization of Unbounded Length

Some prior works have tried to understand the model by inspecting the attention patterns (Ebrahimi et al., 2020; Clark et al., 2019; Vig & Belinkov, 2019). However, we will show that the attention patterns alone are too flexible to be helpful, even for the restricted class of a 2-layer Transformer with a minimal first layer (Assumption 1) and even on a language as simple as Dyck. In particular, the Transformer only needs to satisfy what we call the *balanced condition*:

**Definition 3** (Balance condition). *A 2-layer Transformer (Equation (10)) with a minimal first layer (Assumption 1 and Definition 2) is said to satisfy the* balance condition, *if for any $i, j_1, j_2 \in [k]$ and $d', d_1, d_2 \in [D]$,*

$$(\boldsymbol{e}(\tau_{2i-1,d'}) - \boldsymbol{e}(\tau_{2i,d'-1}))^\top (W_K^{(2)})^\top W_Q^{(2)} (\boldsymbol{e}(\tau_{2j_1,d_1}) - \boldsymbol{e}(\tau_{2j_2,d_2})) = 0. \tag{11}$$

The following result shows that under minor conditions the balance condition is both necessary and sufficient:

**Theorem 1** (Perfect Balance). *Consider a two-layer Transformer $\mathcal{T}$ (Equation (10)) with a minimal first layer (Assumption 1) and $C_{LN} = 0$ (Equation (7)). Let $\mathcal{O}$ denote the* optimal prediction scenario, *that is, when the first layer embeddings $\{e(\tau_{i,d})\}_{d\in[D],i\in[2k]}$ (Definition 2) and second layer parameters $\theta^{(2)}$ satisfy*

$$\theta := \{e(\tau_{i,d})\}_{d\in[D],i\in[2k]}, \theta^{(2)}\} = \arg\min_{\tilde{\theta}} \mathcal{L}(\tilde{\theta}; \mathcal{D}_{q,k,D,N}), \forall N,$$

*where the objective $\mathcal{L}$ is defined in Equation (4). Then,*

- *Equation (11) a necessary condition of $\mathcal{O}$, if $W_V^{(2)}$ satisfies $\mathcal{P}_\perp W_V^{(2)} e(\tau_{t,d}) \neq 0, \forall t \in [k], d \in [D]$.*

- *Equation (11) is a sufficient condition of $\mathcal{O}$, if the set of $2k + 1$ encodings $\{e(\tau_{2i-1,d}), e(\tau_{2i,d})\}_{i\in[k]} \cup \{e(t_\mathcal{S})\}$ are linearly independent for any $d \in [D]$, and the projection function $\mathrm{g}^{(2)}$ is a 6-layer MLP [3] with $O(k^2D^2)$ width.*

*Remark*: Recall from Equation (7) that $\mathcal{P}_\perp$ projects to the subspace orthogonal to $\mathbf{1}\mathbf{1}^\top$. The assumption in the necessary condition can be intuitively understood as requiring all tokens to have nonzero contributions to the prediction after the LayerNorm.

Recall that $e(\tau_{2i-1,d'}), e(\tau_{2i,d'-1})$ denote the first-layer outputs for a matching pair of brackets. Intuitively, Equation (11) says that since matching brackets should not affect future predictions, their embeddings should balance out each other. The balance condition Equation (11) is "perfect" in the sense that the theory assumes the model can minimize the loss for any length $N$; we will see an approximate version later in Theorem 3.

*Proof of the necessity of the balance condition.* The key idea is reminiscent of the pumping lemma for regular languages. For any prefix $p$ ending with a closed bracket $\tau_{2j,d}$ for $d \geq 1$ and containing brackets of all depths in $[D]$, let $p_\beta$ be the prefix obtained by inserting $\beta$ pairs of $\{\tau_{2i-1,d'}, \tau_{2i,d'-1}\}$ for arbitrary $i \in [k]$ and $d' \in [D]$. Denote the *projection of the unnormalized attention output* by

$$u(\tau_{t_1,d_1}, \tau_{t_2,d_2}) := \mathcal{P}_\perp \exp\left(e(\tau_{t_1,d_1})^\top (W_K^{(2)})^\top W_Q^{(2)} e(\tau_{t_2,d_2})\right) W_V^{(2)} e(\tau_{t_1,d_1}). \tag{12}$$

We ignored the normalization in softmax above, since the attention output will be normalized directly by LayerNorm according to Equation (6).

By Equation (10), there exists a vector $v \in \mathbb{R}^m$ such that for any $\beta \in \mathbb{N}$, the next-token logits given by Transformer $\mathcal{T}$ are

$$\mathcal{T}(p_\beta) = W_{\text{Head}} g^{(2)} \left( \frac{v + \beta(u(\tau_{2j,d}, \tau_{2i,d'-1}) + u(\tau_{2j,d}, \tau_{2i-1,d'}))}{\|v + \beta(u(\tau_{2j,d}, \tau_{2i,d'-1}) + u(\tau_{2j,d}, \tau_{2i-1,d'}))\|_2} + e(\tau_{2j,d}) \right). \tag{13}$$

The proof proceeds by showing a contradiction. Suppose $u(\tau_{2j,d}, \tau_{2i,d'-1}) + u(\tau_{2j,d}, \tau_{2i-1,d'}) \neq 0$. Based on the continuity of the projection function and the LayerNorm Layer, we can show that $\lim_{\beta\to\infty} \mathcal{T}(p_\beta)$ depend only on grammar depths $d, d'$ and types $2j, 2i - 1, 2i$. However, these are not sufficient to determine the next-token probability from $p_\beta$, since the latter depends on the type of the last unmatched open bracket in $p$. This contradicts the assumption that the model can minimize the loss for any length $N$. Hence we must have

$$u(\tau_{2j,d}, \tau_{2i,d'-1}) + u(\tau_{2j,d}, \tau_{2i-1,d'}) = 0. \tag{14}$$

Finally, as we assumed that $\mathcal{P}_\perp W_V^{(2)} e(\tau_{t,d}) \neq 0$, we conclude that

$$(e(\tau_{2i-1,d'}) - e(\tau_{2i,d'-1}))^\top (W_K^{(2)})^\top W_Q^{(2)} e(\tau_{2j+1,d}) = \ln\left(\frac{\|\mathcal{P}_\perp W_V e(\tau_{2i,d'-1})\|_2}{\|\mathcal{P}_\perp W_V e(\tau_{2i-1,d'})\|_2}\right),$$

where the right hand side is independent of $j, d$, concluding the proof for necessity. The proof of sufficiency are given in Appendix C.1. □

---

[3] The 6 layers are by our construction. We will first use 4 layers to convert the input of the projection function to a triplet indicating the type and depth of the last token and the type of the last unmatched bracket when the last token is a closed bracket. We will then use another 2 layers to predict the next token probability based on the triplet. This construction may be improved.

Note that the perfect balance condition is an orthogonal consideration to interpretability. For example, even the uniform attention satisfies the condition and can solve Dyck: [4]

**Corollary 1.** *There exists a 2-layer Transformer with uniform attention and no position embedding (but with causal mask and a starting token [5] ) that generates the Dyck language of arbitrary length.*

Since uniform attention patterns are hardly reflective of any structure of Dyck, Corollary 1 proves that attention patterns can be oblivious about the underlying task, violating the "faithfulness" criteria for an interpretation (Jain & Wallace, 2019). We will further show in Appendix B.1 that empirically, seemingly structured attention patterns may not accurately represent the inherent structure of the task.

*Extension to approximate balance condition*: Theorem 1 assumes the model reaches the optimal loss for Dyck prefixes of any length. However, in practice, due to finite samples and various sources of randomness, training often does not end exactly at a population optima. In this case, the condition in Theorem 1 is not precisely met. However, even for models that *approximately* meet those conditions, we will prove that when the second-layer projection function $g^{(2)}$ is Lipschitz, a similar condition as in Equation (14) is still necessary. Details are deferred to Appendix C.4.

### 3.2 Interpretability Requires Inspecting More Than Any Single Weight Matrix

Another line of interpretability works involves inspecting the weight matrices of the model (Li et al., 2016; Dar et al., 2022; Eldan & Li, 2023). Some of the investigations are done locally, neglecting the interplay between different parts of the model. Our result in this section shows that from a representational perspective, isolating single weights can also be misleading for interpretability. For this section only, we will assume the linear head $W_{\text{Head}}$ is identity for simplicity. To consider the effect of pruning, we will also extend the parameterization of LayerNorm module (Equation (7)) as

$$\text{LN}_{C_{LN}}[b](A)_{:,j} = b\frac{\mathcal{P}_\perp A_{:,j}}{\max\{\|\mathcal{P}_\perp A_{:,j}\|_2, \epsilon\}} + (1-b)A_{:,j},$$

which corresponds to a weighted residual branch; note that the original LayerNorm corresponds to $\text{LN}_C[1]$. Let $\hat{\theta}$ denote the set of parameters of this extended parameterization.

We define the *nonstructural pruning* [6] as:

**Definition 4** (Nonstructural pruning)**.** *Under the extended parameterization, a nonstructural pruning of a Transformer with parameter $\hat{\theta}$ is a Transformer with the same architecture and parameter $\hat{\theta}'$, so that for any weight matrix $W$ in $\hat{\theta}$, the corresponding matrix $W'$ in $\hat{\theta}'$ has $W'_{i,j} \in \{W_{i,j}, 0\}, \forall i, j$.*

To measure the quality of the pruning, define the $\epsilon$-approximation:

**Definition 5** ($\epsilon$-approximation)**.** *Given two metric spaces $A, B$ with the same metric $\|\cdot\|$, a function $f: A \to B$ is an $\epsilon$-approximation of function $g$ with respect to that metric, if and only if,*

$$\forall x \in A, \|f(x) - g(x)\| \leq \epsilon\|x\|.$$

The metric, unless otherwise specified, will be the 2-norm for vectors and the 1, 2-norm for matrices:

**Definition 6.** *The $1, 2$-norm of a matrix $A$ is the max row norm, i.e. $\|A\|_{1,2} = \max_{i \in [d']} \|A_{:,i}\|_2$.*

With these definitions, we are ready to state the main result of this section:

**Theorem 2** (Indistinguishability From a Single Component)**.** *Consider any $L$-layer Transformer $\mathcal{T}$ (Equation (10)) with embedding dimension $m$, attention dimension $m_a$, and projection function $g$ as 2-layer ReLU MLP with width $w$. For any $\delta \in (0, 1)$ and $N \in \mathbb{N}^+$, consider a $4L$-layer random Transformer $\mathcal{T}_{large}$ with embedding dimension $m_{\text{large}} = O(m\log(Lm/\delta))$, attention dimension $m_{\text{large},a} = O(m_a L \log\frac{m_a mLN}{\epsilon\delta})$, and projection function $g_{\text{large}}$ as 4-layer ReLU MLP with width $w_{\text{large}} = O(\max\{m, w\}L \log\frac{wmLN}{\epsilon\delta})$.*

---

[4] This is verified empirically: the uniform-attention models have attention weights fix to 0 and are to fit the distribution almost perfectly ($> 99\%$ accuracy).

[5] Here the starting token is necessary because otherwise, the Transformer with uniform attention will have the same outputs for prefix $p$ and prefix $p \oplus p$, in which $\oplus$ denotes concatenation, i.e. $p \oplus p$ means the same string $p$ repeated twice.

[6] This is as opposed to *structural pruning*, which prunes entire rows/columns of weight matrices.

*Assume that $\|W\|_2 \leq 1$ for every weight matrix $W$ in $\mathcal{T}$, and suppose the weights are randomly sampled as $W_{i,j} \sim U(-1, 1)$ for every $W \in \mathcal{T}_{large}$. Then, with probability $1 - \delta$ over the randomness of $\mathcal{T}_{large}$, there exists a nonstructural pruning (Definition 4) of $\mathcal{T}_{large}$, denoted as $\tilde{\mathcal{T}}_{large}$, which $\epsilon$-approximates $\mathcal{T}$ with respect to $\| \cdot \|_{1,2}$ for any input $X \in \mathbb{R}^{m \times N}$ satisfying $\|X\|_{1,2} \leq 1$.* [7]

**Proof sketch: connection to Lottery Tickets.** Theorem 2 can be interpreted as a lottery ticket hypothesis (Frankle & Carbin, 2018; Malach et al., 2020) for randomly initialized Transformers, which can be of independent interest. The proof repeatedly uses an extension of Theorem 1 of Pensia et al. (2020), where it 1) first prunes the $(2l - 1)$-th and $2l$-th layers of $\mathcal{T}_{large}$ to approximate $\mathcal{T}^{(l)}$ for each $l \in [L]$ (Lemma 6), and 2) then prunes the remaining $2L + 1$ to $4L$-th layers of $\mathcal{T}_{large}$ to approximate the identity function. The full proof is deferred to Appendix C.5.

Noting that the layers used to approximate the identity can appear at arbitrary depth in $\mathcal{T}_{large}$, a direct corollary of Theorem 2 is that one cannot distinguish between two functionally different Transformers by inspecting any single weight matrix only:

**Corollary 2.** *Let $\mathcal{T}_1, \mathcal{T}_2$ and $\mathcal{T}_{large}$ follow the same definition and assumptions as $\mathcal{T}$ and $\mathcal{T}_{large}$ in Theorem 2. Pick any weight matrix $W$ in $\mathcal{T}_{large}$, then with probability $1 - \delta$ over the randomness of $\mathcal{T}_{large}$, there exist two Transformers $\mathcal{T}_{Large,1}, \mathcal{T}_{Large,2}$ pruned from $\mathcal{T}_{large}$, such that $\mathcal{T}_{Large,i}$ $\epsilon$-approximate $\mathcal{T}_i$, $\forall i \in \{1, 2\}$, and $\mathcal{T}_{Large,1}, \mathcal{T}_{Large,2}$ coincide on the pruned versions of $W$.*

Hence, one should be cautious when using methods based solely on individual components to interpret the overall function of a Transformer.

# 4 Experiments

Our theory in Section 3 proves the existence of abundant *non-stack-like* attention patterns, all of which suffice for (near-)optimal generalization on Dyck. However, could it be that stack-like solutions are more frequently discovered empirically, due to potential *implicit biases* in the architecture and the training procedure? In this section, we show there is no evidence for such implicit bias in standard training (Section 4.1). Additionally, we propose a regularization term based on the balance condition (Theorem 1), which leads to better length generalization (Section 4.2).

## 4.1 Different Attention Patterns Can Be Learned To Generate Dyck

We empirically verify our theoretical findings that Dyck solutions can give rise to a variety of attention patterns, by evaluating the accuracy of predicting the last bracket of a prefix (Equation 2) given the rest of the prefix. We only consider prefixes ending with a closing bracket, so that there exists a unique correct closing bracket which a correct parser should be able to determine. The experiments in this section are based on Transformers with 2 layers and 1 head, hidden dimension 50 and embedding dimension 50, trained using Adam. Additional results for three-layer Transformers are provided in Appendix D.3. The training data consists of valid $\text{Dyck}_{2,4}$ sequences of length less than 28 generated with $q = 0.5$. When tested in-distribution, all models are able to achieve $\geq 97\%$ accuracy.

**Variation in attention patterns** First, as a response to (Q1), we observe that attention patterns of Transformers trained on Dyck are not always stack-like (Figure 1). In fact, the attention patterns differ even across different random initialization. Moreover, while Theorem 1 implies that position encoding is not necessary for a Transformer to generate Dyck, [8] adding the position encoding [9] does affect the attention patterns (Figures 1c and 1d).

---

[7]Here the input and output dimension of $\tilde{\mathcal{T}}_{large}$ is actually $m_{large}$ which is larger than $m$; additional dimensions are padded with zeroes. The norm constraint can be easily extended to an arbitrary constant.

[8]This is verified empirically, as Transformers with no positional encoding achieve $\geq 97\%$ accuracy.

[9]We use the linear positional encoding following Yao et al. (2021): for the $i_{th}$ position, the encoding is defined to be $e_p(i) := i/T_{\max}$ for some $T_{\max}$.

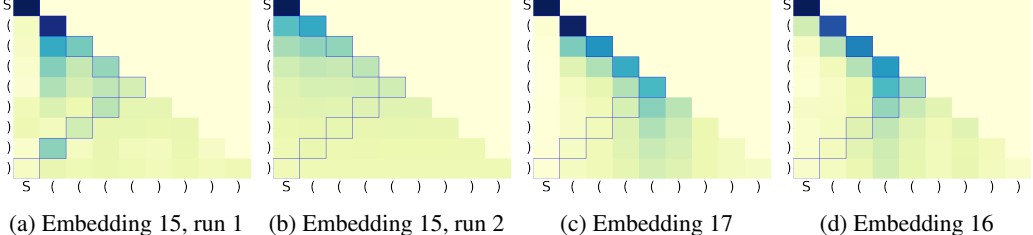

| (a) Embedding 15, run 1 | (b) Embedding 15, run 2 | (c) Embedding 17 | (d) Embedding 16 |

Figure 2: **Second-layer attention patterns of two-layer Transformers with a minimal first layer**: (a), (b) are based on embedding 15 with different learning rates, where the attention patterns show much variance as Theorem 1 predicts. (c), (d) are based on embedding 17 and 16. Different embedding functions lead to diverse attention patterns, most of which are not stack-like.

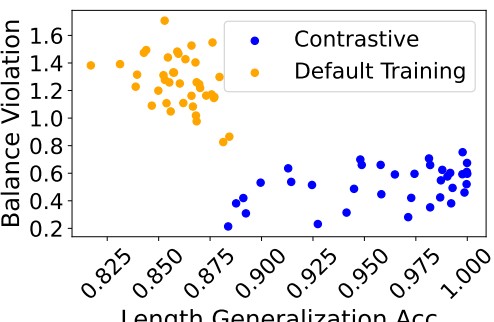

Figure 3: **Relationship Between Balance Violation and Length Generalization.** Accuracy from Transformers with minimal first layer with embedding 15, using both standard training and contrastive regularization (Equation (18)). Standard training leas to high balance violations which negatively correlate with length generalization performance. Contrastive regularization helps reduce the balance violation and improve the length generalization performance.

Specifically, for 2-layer Transformers with a minimal first layer, we experiment with three different types of embeddings $e$: let $o_t$ denote the one-hot embedding where $o_t[t] = 1$,

$$e(\tau_{t,d}) = o_{(t-1)D+d} \in \mathbb{R}^{2kD}, \tag{15}$$

$$e(\tau_{t,d}) = o_t \oplus o_d \in \mathbb{R}^{2k+D}, \tag{16}$$

$$e(\tau_{t,d}) = o_t \oplus [\cos(\theta_d), \sin(\theta_d)] \in \mathbb{R}^{2k+2}, \theta_d = \arctan(d/(D+2-d)), \tag{17}$$

where $\oplus$ denotes vector concatenation. Equation (15) is the standard one-hot embedding for $\tau_{t,d}$; Equation (16) is the concatenation of one-hot embedding of types and depths. Finally, Equation (17) is the embedding constructed in Yao et al. (2021). As shown in Figure 2, the attention patterns learned by Transformers exhibit large variance between different choices of architectures and learning rates, and most learned attention patterns are not stack-like.

**Quantifying the variation** We now quantify the variation in attention by comparing across multiple random initializations. We define the *attention variation* between two attention patterns $A_1, A_2$ as $\text{Variation}(A_1, A_2) = \|A_1 - A_2\|_F^2$, for $A_1, A_2 \in \mathbb{R}^{N \times N}$ over an length-$N$ input sequence. We report the *average attention variation* of each architecture based on 40 random initializations.

On the prefix $[[[[]]]]((((()))) $ [10], we observe that for standard two layer training, the average attention variation is 2.20 with linear position embedding, and is 2.27 without position embedding. Both numbers are close to the random baseline value of 2.85 [11], showing that the attention head learned by different initializations indeed tend to be very different. We also experiment with Transformer with a minimal first layer and the embedding in Equation (15), where the average variation is reduced to 0.24. We hypothesize that the structural constraints in this setting provide sufficiently strong inductive bias that limit the variation.

---

[10]This prefix contains brackets of all types and depths. Results with different prefixes are provided in Appendix D.3.

[11]The random baseline is calculated by generating purely random attention patterns (from the simplex, i.e. random square matrices s.t. each row sums up to 1) and calculate the average attention variation between them.

## 4.2 Guiding The Transformer To Learn Balanced Attention

In our experiments, we observe that although models learned via standard training that can generalize well in distribution, the *length generalization* performance is far from optimal. This implies that the models do not correctly identify the parsing algorithm for Dyck when learning from finite samples. A natural question is: can we guide Transformers towards correct algorithms, as evidenced by improved generalization performance on longer Dyck sequences?

In the following, we measure length generalization performance by the model accuracy on valid Dyck prefixes with length randomly sampled from $400$ to $500$, which corresponds to around 16 times the length of the training sequences. Inspired by results in Section 3, we propose a regularization term to encourage more balanced attentions, which leads to better length generalization.

**Regularizing for balance violation improves length generalization accuracy** We denote the *balance violation* of a Transformer as $\beta := \mathbb{E}_{d,d',i,j} \left[ S_{d,d',i,j}/P_{d,j} \right]$ for $S, P$ defined in Equations (31) and (33). Theorem 1 predicts that for models with a minimal first layer, perfect length generalization requires $\beta$ to be zero. Inspired by this observation, we design a contrastive training objective to reduce the balance violation, which ideally would lead to improved length generalization. Specifically, let $p_r$ denote a prefix of $r$ nested pairs of brackets of for $r \sim U([D])$, and let $\mathcal{T}(s \mid p_r \oplus s)$ denote the logits for $s$ when $\mathcal{T}$ takes as input the concatenation of $p_r$ and $s$. We define the *contrastive regularization term* $R_{\text{contrastive}}(s)$ as the mean squared error between the logits of $\mathcal{T}(s)$ and $\mathcal{T}(s \mid p_r \oplus s)$, taking expectation over $r$ and $p_r$:

$$\mathbb{E}_{r \sim U([D]), p_r} \left[ \| \mathcal{T}(s \mid p_r \oplus s) - \mathcal{T}(s) \|_F^2 \right]. \tag{18}$$

Following the same intuition as in the proof of Theorem 1, if the model can perfectly length-generalize, then the contrastive loss will be zero. Models trained with contrastive loss show reduced balance violation as well as improved length generalization performance, as shown in Figure 3.

## 5 Conclusion

Why interpreting individual components sometimes leads to misconceptions? Through a case study of the Dyck grammar, we provide theoretical and empirical evidence that even in this simple and well-understood setup, Transformers can implement a rich set of non-interpretable solutions. This is reflected both by diverse attention patterns and by the absence of task-specific structures in local components. Our results directly imply similar conclusions for more complex Transformer models; see Appendix C.2 for technical details. Together, this work provides definite proof that myopic interpretability, i.e. methods based on examining individual components only, are not sufficient for understanding the functionality of a trained Transformer.

Our results do not preclude that interpretable attention patterns can emerge; however, they do suggest that interpretable patterns can be infrequent. We discuss the implications for multi-head, overparameterized Transformers trained on more complex data distributions in Appendix B. Moreover, our current results pertain to the existence of solutions; an interesting next step is to study how "inductive biases" given by the synergy of the optimization algorithm and the architecture affect the solutions found.

## Acknowledgement

This work was in part supported by NSF awards IIS-2211907, CCF-2238523, and Amazon Research Award.

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

# Contents

# A   Additional Related Work

**Interpreting Transformer solutions**   Prior empirical works show that Transformers trained on natural language data can produce representations that contain rich syntactic and semantic information, by designing a wide range of "probing" tasks (Raganato & Tiedemann, 2018; Liu et al., 2019; Hewitt & Manning, 2019; Clark et al., 2019; Tenney et al., 2019; Hewitt & Liang, 2019; Kovaleva et al., 2019; Lin et al., 2019; Wu et al., 2020; Belinkov, 2022) (or other approaches using the attention weights or parameters in neurons directly Vig & Belinkov, 2019; Htut et al., 2019; Sun & Marasović, 2021; Eldan & Li, 2023). However, there is no canonical way to probe the model, partially due to the huge design space of probing tasks, and even a slight change in the setup may lead to very different (sometimes even seemingly contradictory) interpretations of the result (Hewitt & Liang, 2019). In this work, we tackle such ambiguity through a different perspective—by developing formal (theoretical) understanding of solutions learned by Transformers. Our results imply that it may be challenging to try to interpret Transformer solutions based on individual parameters (Li et al., 2016; Dar et al., 2022), or based on constructive proofs (unless the Transformer is specially trained to be aligned with a certain algorithm, as in Weiss et al., 2021).

**Interpreting attention patterns**   Prior works (Jain & Wallace, 2019; Serrano & Smith, 2019; Rogers et al., 2020; Grimsley et al., 2020; Brunner et al., 2020; Prasanna et al., 2020; Meister et al., 2021; Bolukbasi et al., 2021; Haab et al., 2023, *inter alia*) present negative results on deriving explanations from attention weights using approaches by Vig & Belinkov (2019); Kobayashi et al. (2020, *inter alia*). However, Wiegreffe & Pinter (2019) argues to the contrary by pointing out flaws in the experimental design and arguments of some of the prior works; they also call for theoretical analysis on the issue. Hence, a takeaway from these prior works is that expositions on explainability based on attention requires clearly defining the notion of explainability adopted (often task-specific). In our work, we restrict our main theoretical analysis to the fully defined data distribution of Dyck language (Definition 1), and define "interpretable attention pattern" as the stack-like pattern proposed in prior theoretical (Yao et al., 2021) and empirical (Ebrahimi et al., 2020) works. These concrete settings and definitions allow us to mathematically state our results and provide theoretical reasons.

**Theoretical understanding of representability**   Methodologically, our work joins a long line of prior works that characterize the solution of neural networks via the lens of simple synthetic data, from class results on RNN representability (Siegelmann & Sontag, 1992; Gers & Schmidhuber, 2001; Weiss et al., 2018; Suzgun et al., 2019; Merrill, 2019; Hewitt et al., 2020), to the more recent Transformer results on parity (Hahn, 2020), Dyck (Yao et al., 2021), topic model (Li et al., 2023), and formal grammars in general (Bhattamishra et al., 2020a; Li & Risteski, 2021; Zhang et al., 2022; Liu et al., 2023; Zhao et al., 2023). Our work complements prior works by showing that although representational results can be obtained via intuitive "constructive proofs" that assign values to the weight matrices, the model does not typically converge to those intuitive solutions in practice. Similar messages are conveyed in Liu et al. (2023), which presents different types of constructions using different numbers of layers. In contrast, we show that there exist multiple different constructions even when the number of layers is kept the same.

There are also theoretical results on Transformers in terms of Turing completeness (Bhattamishra et al., 2020b; Perez et al., 2021), universal approximatability (Yun et al., 2020), and statistical sample complexity (Wei et al., 2021; Edelman et al., 2022), which are orthogonal to our work.

**Transformer optimization**   Given multiple global optima, understanding Transformer solutions requires analyzing the training dynamics. Recent works theoretically analyze the learning process of Transformers on simple data distributions, e.g. when the attention weights only depend on the position information (Jelassi et al., 2022), or only depend on the content (Li et al., 2023). Our work studies a syntax-motivated setting in which both content and position are critical. We also highlight that Transformer solutions are very sensitive to detailed changes, such as positional encoding, layer norm, sharpness regularization (Foret et al., 2020), or pre-training task (Liu et al., 2022a). On a related topic but towards different goals, a series of prior works aim to improve the training process of Transformers with algorithmic insights (Nguyen & Salazar, 2019; Xiong et al., 2020; Liu et al., 2020; Zhang et al., 2020; Li & Gong, 2021, *inter alia*). An end-to-end theoretical characterization of the training dynamics remains an open problem; recent works that propose useful techniques towards this goal include Gao et al., 2023; Deng et al., 2023.

**Mechanistic interpretability**    Finally, it is worth noting that the challenges highlighted in our work do not contradict the line of prior works that aim to improve *mechanistic interpretability* into a trained model or the training process (Cammarata et al., 2020; Elhage et al., 2021; Olsson et al., 2022; Nanda et al., 2023; Chughtai et al., 2023; Li et al., 2023; Wang et al., 2023; Zhong et al., 2023): although we prove that components (e.g. attention scores) of trained Transformers do not generally admit intuitive interpretations based on the data distribution, it is still possible to develop circuit-level understanding about a particular model, or measures that closely track the training process, following these prior works.

# B  Are interpretable attention patterns useful?

Our results Section 3 and Section 4.1 demonstrate that Transformers are sufficiently expressive that a (near-)optimal loss on Dyck languages can be achieved by a variety of attention patterns, many of which may not be interpretable.

However, multiple prior works have shown that for multi-layer multi-head Transformers trained on natural language datasets, it is often possible to locate attention heads that produce interpretable attention patterns (Vig & Belinkov, 2019; Htut et al., 2019; Sun & Marasović, 2021). Hence, it is also illustrative to consider the *"converse question"* of (Q1): when some attention heads do learn to produce attention patterns that suggest intuitive interpretations, what benefits can they bring?

We discuss this through two perspectives:

- **Reliability of interpretation:** Is the Transformer necessarily implementing a solution consistent with such interpretation based on the attention patterns? (Section B.1)
- **Usefulness for task performance:** Are those interpretable attention heads more important for the task than other uninterpretable attention heads? (Section B.2)

We present preliminary analysis on these questions, and motivate future works on the interpretability of attention patterns using rigorous theoretical analysis and carefully designed experiments.

## B.1  Can interpretable attention patterns be misleading?

We show through a simple argument that interpretations based on attention patterns can sometimes be misleading, as we formalize in the following proposition:

**Proposition 1.** *Consider an L-layer Transformer $\mathcal{T}$ (Equation (10)). For any $W_K^{(l)}, W_Q^{(l)} \in \mathbb{R}^{m_a \times m}$ ($l \in [L]$), there exist $W_{\mathrm{Head}} \in \mathbb{R}^{2k \times w}$ and $b_{\mathrm{Head}} \in \mathbb{R}^{2k}$ such that $\mathcal{T}(\mathcal{Z}) = 0, \forall \mathcal{Z}$.*

While its proof is trivial (simply setting $W_{\mathrm{Head}} = 0$ and $b_{\mathrm{Head}} = 0$ suffices), Proposition 1 implies that the solution represented by the Transformer could possibly be independent of the attention patterns in all the layers (1 through $l$). Hence, it could be misleading to interpret Transformer solutions solely based on these attention patterns.

Empirically, Transformers trained on Dyck indeed sometimes produce misleading attention patterns.

We present one representative example in Figure 4, and Figure 5, in which *all interpretable attention patterns are misleading*.

We also present additional results in Figure 6, in which *some interpretable attention patterns are misleading, and some are not*.

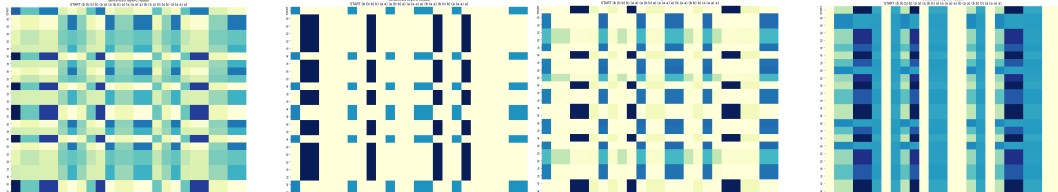

Figure 4: **Even interpretable attention patterns can be misleading**: For a 4-layer Transformer trained on Dyck with the *copying* task (with $> 96\%$ validation accuracy), i.e. the output should be exactly the same as the input, the attention patterns in some layers seem interpretable: (layer 2) attending to bracket type a) or (b; (layer 3) attending to closing bracketss; (layer 4) neve attending to bracket type a); However, none of them are informative of the copying task. This is possible because Transformers can use the residual connections (or weights MLPs or the value matrices) to solve copying, bypassing the need of using attention.

Similar message has been conveyed in prior works Bolukbasi et al. (2021), and future works may aim to achieve the *faithfulness*, *completeness*, and *minimality* conditions in Wang et al. (2023).

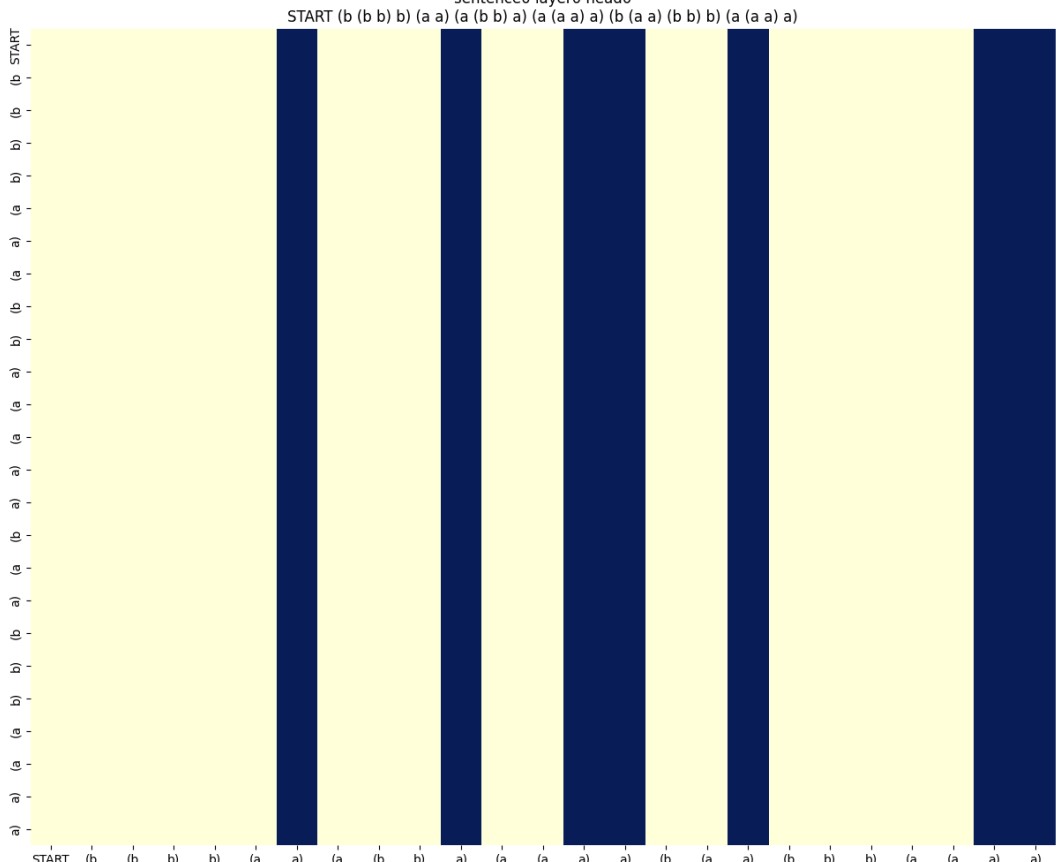

Figure 5: **Even interpretable attention patterns can be misleading**: For a 1-layer Transformer trained on Dyck with the *copying* task (with $> 90\%$ validation accuracy), i.e. the output should be exactly the same as the input, the attention pattern seems to be attending to closing brackets only, but that is not informative of the copying task.

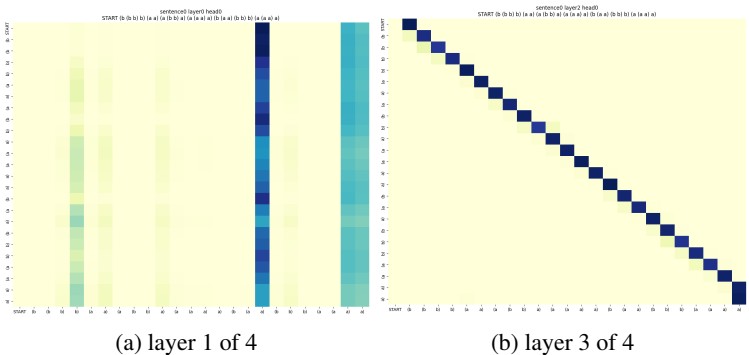

(a) layer 1 of 4          (b) layer 3 of 4

Figure 6: **Even interpretable attention patterns can be misleading**: For a 4-layer Transformer trained on Dyck with the *copying* task (with $> 96\%$ validation accuracy), i.e. the output should be exactly the same as the input, both types of attention patterns are common: (a) attending to closing bracketss, which is uninformative of the copying task; (b) attending to the current position, which solves the copying task.

## B.2 Are attention heads with interpretable patterns more important?

Kovaleva et al. (2019) observes that, when the "importance" of an attention head is defined as the performance drop the model suffers when the head is disabled, then for most tasks they test, the most important attention head in each layer *does not* tend to be interpretable.

However, experiments by Voita et al. (2019) led to a seemingly contradictory observation: when attention heads are systematically pruned by finetuning the Transformer with a relaxation of $L_0$-penalty (i.e. encouraging the number of remaining attention heads to be small), most remaining attention heads that survive the pruning can be associated with certain functionalities such as positional, syntactic, or attending to rare tokens.

These works seem to bring mixed conclusions to our question: are interpretable attention heads more important for a task than uninterpretable ones? We interpret these results by conjecturing that the definition of "importance" (reflected in their experimental design) plays a crucial role:

- When the importance of an attention head is defined *treating all other attention heads as fixed*, motivating experiments that prune/disable certain heads while keeping other heads unchanged (Michel et al., 2019; Kovaleva et al., 2019), the conclusion may be mostly pessimistic: mostly no strong connection between interpretability and importance.

- On the other hand, when the importance of an attention head is defined *allowing all other attention heads to adapt to its change*, motivating experiments that jointly optimize all attention heads while penalizing the number of heads (Voita et al., 2019), the conclusion may be more optimistic: the heads obtained as a result of this optimization tend to be interpretable.

We think the following trade-offs apply:

- On one hand, the latter setting is more practical, since Transformers are typically not trained to explicitly ensure that the model performs well when a single attention head is individually disabled; rather, it would be more intuitive to think of a group of attention heads as jointly representing some transformation, so when one head is disabled, other heads should be fine-tuned to adapt to the change.

- On the other hand, when all other heads change too much during such fine-tuning, the resulting set of attention heads no longer admit an unambiguous one-to-one map with the original set of (unpruned) attention heads. As a result, the interpretability and importance obtained from the set of pruned heads do not necessarily imply those properties of the original heads.

A comprehensive study of this question involves multi-head extensions of our theoretical results (Section 3), and carefully-designed experiments that take the above-mentioned trade-offs into consideration. We think these directions are interesting future work.

# C Omitted Proofs in Section 3

## C.1 Proof of Theorem 1

The key step is already shown in Section 3. We will restate the proof rigorously here.

**Theorem 1** (Perfect Balance). *Consider a two-layer Transformer $\mathcal{T}$ (Equation (10)) with a minimal first layer (Assumption 1) and $C_{LN} = 0$ (Equation (7)). Let $\mathcal{O}$ denote the* optimal prediction scenario, *that is, when the first layer embeddings $\{e(\tau_{i,d})\}_{d\in[D],i\in[2k]}$ (Definition 2) and second layer parameters $\theta^{(2)}$ satisfy*

$$\theta := \{e(\tau_{i,d})\}_{d\in[D],i\in[2k]}, \theta^{(2)}\} = \arg\min_{\tilde{\theta}} \mathcal{L}(\tilde{\theta}; \mathcal{D}_{q,k,D,N}), \forall N,$$

*where the objective $\mathcal{L}$ is defined in Equation (4). Then,*

- *Equation (11) a necessary condition of $\mathcal{O}$, if $W_V^{(2)}$ satisfies $\mathcal{P}_\perp W_V^{(2)} e(\tau_{t,d}) \neq 0, \forall t \in [k], d \in [D]$.*

- *Equation (11) is a sufficient condition of $\mathcal{O}$, if the set of $2k + 1$ encodings $\{e(\tau_{2i-1,d}), e(\tau_{2i,d})\}_{i\in[k]} \cup \{e(t_{\mathcal{S}})\}$ are linearly independent for any $d \in [D]$, and the projection function $\mathrm{g}^{(2)}$ is a 6-layer MLP [12] with $O(k^2 D^2)$ width.*

*Proof.* We prove the **sufficiency of the balanced condition** below; the proof for the *necessity* has been given in Section 3.1.

We will denote the dimension of $e(\tau_{t,d})$ as $m$.

For any $i \in [k], d' \in [D]$, by Equation (11), we can assume that there exists $a_{i,d'} \in \mathbb{R}$ such that for any $j \in [k], d \in [D]$, it holds that,

$$a_{i,d'} \triangleq \left(e\left(\tau_{2i-1,d'}\right) - e\left(\tau_{2i,d'-1}\right)\right)^\top \left(W_K^{(2)}\right)^\top W_Q^{(2)} e\left(\tau_{2j,d}\right). \tag{19}$$

We will first define the possible index sets of $\tau_{t,d}$ as $\mathcal{I} = \{(2t, d) \mid t \in [k], 0 \leq d \leq D - 1\} \cup \{(2t - 1, d) \mid t \in [k], 1 \leq d \leq D\}$, and we will define the rank of $(t, d)$ as

$$r(t, d) \triangleq \#\{(t_1, d_1) \mid t_1 < t \text{ or } t_1 = t, d_1 \leq d, (t_1, d_1) \in \mathcal{I}\} \tag{20}$$

Then it is clear that $r(t, d)$ is a one-to-one mapping from $\mathcal{I}$ to $[2kD]$. We will then define the collection of all $e(\tau_{t,d})$ as $\boldsymbol{E}$, satisfying that $\boldsymbol{E}_{:,r(t,d)} = e(\tau_{t,d}), \boldsymbol{E}_{:,2kD+1} = e(t_{\mathcal{S}})$.

Because $e(\tau_{t,d})$ are linearly independent, for any $(i, d) \neq (j, d') \in \mathcal{I}$, it holds that $e(\tau_{i,d}) - e(\tau_{j,d'}) \neq 0$. Then based on Lemma 16, there exists a set of orthonormal vectors $\{\mathrm{b}_i\}_{i\in[m-2]}$, such that for any $(i, d), (j, d') \in \mathcal{I}$, it holds that

$$\sum_{i=1}^{m-2} \mathrm{b}_i \mathrm{b}_i^\top \left(e(\tau_{i,d}) - e(\tau_{j,d'})\right) \neq \left(e(\tau_{i,d}) - e(\tau_{j,d'})\right) \tag{21}$$

$$\mathrm{b}_i^\top 1^m = 0 \tag{22}$$

We will further construct matrix $\boldsymbol{O}$ as [13]

$$\boldsymbol{O}_{:,r(2t,d-1)} = -\exp(a_{t,d})\mathrm{b}_{tD+d},$$
$$\boldsymbol{O}_{:,r(2t-1,d)} = \mathrm{b}_{tD+d}. \tag{23}$$
$$\boldsymbol{O}_{:,2kD+1} = 0.$$

---

[12] The 6 layers are by our construction. We will first use 4 layers to convert the input of the projection function to a triplet indicating the type and depth of the last token and the type of the last unmatched bracket when the last token is a closed bracket. We will then use another 2 layers to predict the next token probability based on the triplet. This construction may be improved.

[13] Recall the definition of $r$ in Equation (20). Comparing $\boldsymbol{O}_{:,r(2t,d-1)}$ and $\boldsymbol{O}_{:,r(2t-1,d)}$: the idea is that a pair of matched brackets are represented by the same direction (i.e. the direction along $\mathrm{b}_{tD+d}$), just with different norms.

for $t \in [k], d \in [D]$.

We can then choose $W_V^{(2)} \in \mathbb{R}^{m \times m}$ such that

$$W_V^{(2)} \boldsymbol{E} = \boldsymbol{O} \tag{24}$$

Such $W_V^{(2)}$ is guaranteed to exist, because $\boldsymbol{E}$ is of full column rank by the linear independence assumption.

Now based on this construction, we will show that the last column of unnormalized attention output (Equation (9)) depends only on the sequence of unmatched brackets when the last token is a closed bracket with depth $d$ greater than or equal to 1. [14]

For any valid Dyck prefix $p$ of length $n$ ending with a closed bracket $\tau_{2j,d}$ satisfying $d \geq 1$, suppose the list of unmatched open brackets in $p$ is $[\tau_{2j_1-1,1}, \tau_{2j_2-1,2}, \ldots, \tau_{2j_d-1,d}]$. Then, the remaining tokens in $p$ are pairs of matching brackets. Denote them by $\tau_{2t_k-1,d_k}, \tau_{2t_k,d_k-1}$ for $k \in [K]$. Then the input of the second layer of Transformer $X$, up to a permutation is

$$XP = [\boldsymbol{e}(\tau_{2t_1-1,d_1}), \boldsymbol{e}(\tau_{2t_1,d_1-1}), \ldots, \boldsymbol{e}(\tau_{2t_K-1,d_K}), \boldsymbol{e}(\tau_{2t_K,d_K-1}), \boldsymbol{e}(\tau_{2j_1-1,1}), \ldots \boldsymbol{e}(\tau_{2j_d-1,d}), \boldsymbol{e}(t_S)].$$

We will focus on the last column of the unnormalized attention output

$$
\begin{aligned}
\tilde{a}_2(X; \theta^{(2)})_{:,n+1} &= \mathcal{P}_\perp \Big[ W_V^{(2)} X \cdot \tilde{\sigma} \Big( \mathcal{C} \cdot (W_K^{(2)} X)^\top (W_Q^{(2)} X) \Big) \Big]_{:,n+1} \\
&= \sum_{s=1}^{n+1} \mathcal{P}_\perp (W_V^{(2)} X)_{:,s} \Big[ \tilde{\sigma} \Big( \mathcal{C} \cdot (W_K^{(2)} X)^\top (W_Q^{(2)} X) \Big) \Big]_{s,n+1} \\
&= \sum_{s=1}^{n+1} \mathcal{P}_\perp (W_V^{(2)} X)_{:,s} \exp \Big( \Big( (W_K^{(2)} X)^\top (W_Q^{(2)} X) \Big)_{s,n+1} \Big) \\
&= \sum_{s=1}^{n+1} \mathcal{P}_\perp (W_V^{(2)} X)_{:,s} \exp \Big( (W_K^{(2)} X)_{:,s}^\top (W_Q^{(2)} X)_{:,n+1} \Big) \\
&= \sum_{k=1}^{K} \big( u(\tau_{2t_k,d_k-1}, \tau_{2j,d}) + u(\tau_{2t_k-1,d_k}, \tau_{2j,d}) \big) + \sum_{s=1}^{d} u(\tau_{2j_s-1,s}, \tau_{2j,d}) \quad (25)
\end{aligned}
$$

in which the last line is by definition of $u(\cdot, \cdot)$ in Equation (12).

For any indices $s, j_s, j, d$, we can simplify the expression for $u(\tau_{2j_s-1,s}, \tau_{2j,d})$ by observing that

$$
\begin{aligned}
u(\tau_{2j_s-1,s}, \tau_{2j,d}) &= \mathcal{P}_\perp \exp \Big( \boldsymbol{e}(\tau_{2j_s-1,s})^\top (W_K^{(2)})^\top W_Q^{(2)} \boldsymbol{e}(\tau_{2j,d}) \Big) W_V^{(2)} \boldsymbol{e}(\tau_{2j_s-1,s}) \text{ by Eq 12} \\
&= \mathcal{P}_\perp \exp \Big( \boldsymbol{e}(\tau_{2j_s-1,s})^\top (W_K^{(2)})^\top W_Q^{(2)} \boldsymbol{e}(\tau_{2j,d}) \Big) \boldsymbol{O}_{:,r(2j_s-1,s)} \quad \text{by Eq 24} \\
&= \mathcal{P}_\perp \exp \Big( \boldsymbol{e}(\tau_{2j_s-1,s})^\top (W_K^{(2)})^\top W_Q^{(2)} \boldsymbol{e}(\tau_{2j,d}) \Big) \mathrm{b}_{j_s D+s} \quad \text{by Equation (23)} \\
&= \exp \Big( \boldsymbol{e}(\tau_{2j_s-1,s})^\top (W_K^{(2)})^\top W_Q^{(2)} \boldsymbol{e}(\tau_{2j,d}) \Big) \mathrm{b}_{j_s D+s} \quad \text{by Equation (22).} \quad (26)
\end{aligned}
$$

Likewise by Equation (12), Equation (24), Equation (23), Equation (22)

$$u(\tau_{2j_s,s-1}, \tau_{2j,d}) = -\exp \Big( \boldsymbol{e}(\tau_{2j_s,s-1})^\top (W_K^{(2)})^\top W_Q^{(2)} \boldsymbol{e}(\tau_{2j,d}) \Big) \exp(a_{j_s,s}) \mathrm{b}_{j_s D+s} \tag{27}$$

---

[14] When depth $d = 0$, all brackets are matched, the groundtruth next-token distribution is the prior distribution over the open brackets. Because in Equation (11) $d_1, d_2 \geq 1$, we handle the depth $d = 0$ case separately in Case 2 "t is even, $d = 0$" towards the end of this proof. In the following, we focus on cases with depth $d \geq 1$.

By Equation (26) and Equation (27),

$$
\begin{aligned}
& u(\tau_{2t_k,d_k-1}, \tau_{2j,d}) + u(\tau_{2t_k-1,d_k}, \tau_{2j,d}) \\
&= \exp\left(\boldsymbol{e}(\tau_{2t_k-1,d_k})^\top (W_K^{(2)})^\top W_Q^{(2)} \boldsymbol{e}(\tau_{2j,d})\right) \mathrm{b}_{t_k D+d_k} \\
&\quad - \exp\left(\boldsymbol{e}(\tau_{2t_k,d_k-1})^\top (W_K^{(2)})^\top W_Q^{(2)} \boldsymbol{e}(\tau_{2j,d})\right) \exp(a_{t_k,d_k}) \mathrm{b}_{t_k D+d_k} \\
&= \Big[ \exp\left(\boldsymbol{e}(\tau_{2t_k-1,d_k})^\top (W_K^{(2)})^\top W_Q^{(2)} \boldsymbol{e}(\tau_{2j,d})\right) \\
&\quad - \exp\left(\boldsymbol{e}(\tau_{2t_k,d_k-1})^\top (W_K^{(2)})^\top W_Q^{(2)} \boldsymbol{e}(\tau_{2j,d}) + a_{t_k,d_k}\right) \Big] \mathrm{b}_{t_k D+d_k} \\
&= 0
\end{aligned}
\tag{28}
$$

in which the last line is because the terms inside $\big[\cdots\big]$ cancel each other, because by Equation (19)

$$
\boldsymbol{e}(\tau_{2t_k-1,d_k})^\top (W_K^{(2)})^\top W_Q^{(2)} \boldsymbol{e}(\tau_{2j,d}) = \boldsymbol{e}(\tau_{2t_k,d_k-1})^\top (W_K^{(2)})^\top W_Q^{(2)} \boldsymbol{e}(\tau_{2j,d}) + a_{t_k,d_k}
$$

Plugging Equation (28) and Equation (26) into Equation (25),

$$
\begin{aligned}
\tilde{a}_2(X;\theta^{(2)})_{:,n+1} &= \sum_{s=1}^{d} u(\tau_{2j_s-1,s}, \tau_{2j,d}) \\
&= \sum_{s=1}^{d} \exp\left(\boldsymbol{e}(\tau_{2j_s-1,s})^\top (W_K^{(2)})^\top W_Q^{(2)} \boldsymbol{e}(\tau_{2j,d})\right) \mathrm{b}_{j_s D+s}
\end{aligned}
\tag{29}
$$

Therefore, $\tilde{a}_2(X;\theta^{(2)})_{:,n+1}$ lies in the span of $\{\mathrm{b}_{j_s D+s}\}_{s\in[d]}$. We will from now on assume $\langle \mathrm{LN}(\tilde{a}_2(X;\theta^{(2)})_{:,n}), \mathrm{b}_{j_s D+s}\rangle > M$ for all possible choices of $p$ ending with a closed bracket with grammar depth at least 1 for some constant $M \in (0,1)$. Here $M$ exists because

$$
\langle \mathrm{LN}(\tilde{a}_2(X;\theta^{(2)})_{:,n}), \mathrm{b}_{j_s D+s}\rangle = \frac{\exp\left(\boldsymbol{e}(\tau_{2j_s-1,s})^\top (W_K^{(2)})^\top W_Q^{(2)} \boldsymbol{e}(\tau_{2j,d})\right)}{\sqrt{\sum_{s'=1}^{d} \exp\left(2\boldsymbol{e}(\tau_{2j_{s'}-1,s})^\top (W_K^{(2)})^\top W_Q^{(2)} \boldsymbol{e}(\tau_{2j,d})\right)}} > 0,
$$

for all possible combination of $j_k, k \in [d]$ and $s$, and there are only finite number of such combinations.

**Constructing the projection function** $\mathrm{g}^{(2)}$ We will finally show there exists a 6-layer MLP $\mathrm{g}^{(2)}$ with width $O(D^2 k^2)$, such that for any dyck prefix $q$ with $n$ being the length of $q$, $X$ being the input of the second layer given $q$ and $\mathbb{P}(p)$ being the groundtruth next-token probability vector given $q$ [15], it holds that, $\mathrm{g}^{(2)}\big(\mathrm{LN}(\tilde{a}_2(X;\theta^{(2)})_{:,n+1}) + X_{:,n+1}\big) = \mathbb{P}(q)$.

We will assume the last token of $q$ is $\tau_{t,d}$. Suppose that $\mathrm{b}_{m-1}, \mathrm{b}_m$ is an orthonormal basis of the normal space of span$\{\mathrm{b}_1, .., \mathrm{b}_{m-2}\}$, then we can first observe that for $U = \mathrm{b}_m \mathrm{b}_m^\top + \mathrm{b}_{m-1} \mathrm{b}_{m-1}^\top$, it holds that

$$
U(\mathrm{LN}(\tilde{a}_2(X;\theta^{(2)})_{:,n+1}) + X_{:,n+1}) = U\boldsymbol{e}(\tau_{t,d}).
$$

is unique for every $t, d$. Then based on Lemma 15, there exists a 2-layer MLP with width $4kD$ that maps $U(\mathrm{LN}(\tilde{a}_2(X;\theta^{(2)})_{:,n+1}) + X_{:,n+1})$ to $(t,d)$. This implies that there exists a 2-layer MLP with width $4kD$ that maps $\mathrm{LN}((\tilde{a}_2(X;\theta^{(2)})_{:,n}) + X_{:,n}$ to $(t,d)$.

Further, let matrix $U' = \sum_{j=1}^{Dk} \boldsymbol{o}_j \mathrm{b}_j^\top$ where $\boldsymbol{o}_j$ is the $Dk$ dimension one-hot vector with the $j-$th entries being 1. Then when $t$ is an even number and $d \geq 1$, based on Equation (29) and the definition of $M$,

$$
U'(\mathrm{LN}(\tilde{a}_2(X;\theta^{(2)})_{:,n+1}) + X_{:,n+1})_{t'D+d'} \begin{cases} = 0, & \tau_{2t'-1,d'} \text{ is not an unmatched open brackets in } p. \\ > M, & \tau_{2t'-1,d'} \text{ is an unmatched open brackets in } p. \end{cases}
$$

---

[15]That is $\mathbb{P}(q)_t = \mathbb{P}(\text{The next token of } q \text{ has type } t)$

Then based on Lemma 18, there exists 2-layer MLP with width $kD$ that operates on $\left(U'(\mathrm{LN}(\tilde{a}_2(X;\theta^{(2)})_{:,n+1}) + X_{:,n+1})_{t'D+d'}\right)_{t'\in[k]}$ for a fixed $d'$ and output the nonzero index in it, if such index exists. Hence, we can choose the weight of the first and second layer of $\mathrm{g}^{(2)}$, such that the output of the second layer is $(t,d) \oplus x$, where $2x_{d'} - 1$ is the type of the unmatched open brackets with grammar depth $d'$ if $t$ is an even number, $d \geq d' \geq 1$.

Now based on Lemma 17, we can choose the third and fourth layer of $\mathrm{g}^{(2)}$ to perform indexing and let the output of the fourth layer be $(t,d,y)$, where $y = x_d$ when $d \geq 1$. [16] Notice that this triplet contains all the necessary information to infer $\mathbb{P}(q)$ because it uniquely determines the type of last unmatched open bracket,

1. If $t$ is odd (i.e. the last bracket is open), and then the type of last unmatched open bracket is $t$.

2. If $t$ is even and $d = 0$, then all the brackets is matched.

3. If $t$ is even and $d \geq 1$, then the type of last unmatched bracket is $y$.

One may finally construct a 2-layer MLP $f$ that maps $(t,d,y)$ to the corresponding probability vector. As the input of g has bounded norm,

$$\|\mathrm{LN}(\tilde{a}_2(X;\theta^{(2)})_{:,n+1}) + X_{:,n+1}\|_2 \leq 1 + \max_{t,d}\|\boldsymbol{e}(\tau_{t,d})\|,$$

the output of the constructed 4 layers also has a bounded norm. Hence, we can assume there exists constant $M' > 1$, such that $y \leq M'$. Now we will discuss by the value of $t$,

1. $t$ is odd, then one can neglect the third dimension and the correct probability is determined by $d$ and can be represented by a width-$2D$ network based on Lemma 15.

2. $t$ is even. When $d = 0$, one can construct a width-1 network mapping any $y$ to the correct probability distribution as it is unique. When $d \geq 1$, one can construct a width-$2K$ network mapping $x_d \in [K]$ to the correct probability distribution based on Lemma 15. Then by Lemma 19, one can construct a width-$4KD$ network that maps $(d,y)$ to the corresponding probability distribution.

Putting together and using Lemma 19 again, one can construct a width-$8K^2D$ network that maps $(t,d,y)$ to the correct next token probability prediction. The proof is then completed.

$\square$

## C.2 Implication of our results to larger models

Recall that the main conclusion of our paper is that interpretability based on a single Transformer component (e.g. an attention pattern or an MLP block) can be unreliable, since the set of optimal solutions can give rise to a large set of attention patterns and pruned MLP weights. Section 3 has demonstrated this with simple two-layer Transformers. The simplicity of this architecture choice is intentional, since our theory on two-layer Transformers directly implies similar conclusions for larger models, as we discuss in this section.

Intuitively, when moving to more complex architectures, the set of solutions can only grow and complicate interpretability further, hence our main conclusion still stands. For example, even though Theorem 1 and Theorem 3 are stated for 2-layer Transformers only, the constructed solutions can be trivially extended to multiple layers by e.g. letting the higher layers perform the identity function, or removing Assumption 1 and allowing the model to flexibly use or ignore positional information. More precisely:

- For Transformers with greater width, our Theorem 1 applies directly, since the construction does not depend on the width.

- For Transformers with greater depth, it suffices to show that additional layers can perform the identity function. To this end, one can utilize the residue link in the Transformer layer and choose the value matrix to be zero and the FFN (with or without residue connection) to be identity. This

---

[16]When $d = 0$, $y$ does not matter since there is no unmatched open brackets.

construction is implicitly assuming LayerNorm will map zero vector to zero vector, which is true for the common PyTorch implementation and for our paper. Also, it is worth noting that this holds for both the architecture we considered in the paper and the standard GPT-2 architecture.

## C.3 Proof of Corollary 1

**Corollary 1.** *There exists a 2-layer Transformer with uniform attention and no position embedding (but with causal mask and a starting token [17] ) that generates the Dyck language of arbitrary length.*

*Proof.* We will first construct a uniform attention first layer that can generate the embedding in Equation (15). Suppose $Z$ is the one-hot embeddings of a prefix $p$ of length $n$, where each token of type $t$ for $t \in [2k]$ is encoded as $o_t$ and the starting token is encoded as $o_{2k+1}$. Then it holds that

$$\left[ Z\sigma\Big( \mathcal{C} \cdot (W_K^{(1)} Z)^\top (W_Q^{(1)} Z) \Big) \right]_{:,n+1} = \sum_{i=1}^{2k} \#\{\text{token of type } t \text{ in } p\} o_t + o_{2k+1}. \tag{30}$$

Then we can choose $W_V^{(1)}$ such that for $x \in \mathbb{R}^{2k+1}$,

$$(W_V^{(1)} x)_1 = \sum_{i=1}^{k} x_{2i-1} - x_{2i},$$

$$(W_V^{(1)} x)_2 = x_{2k+1},$$

$$(W_V^{(1)} x)_i = 0, \forall i \geq 3.$$

Hence it holds That

$$\left[ W_V^{(1)} Z\sigma\Big( \mathcal{C} \cdot (W_K^{(1)} Z)^\top (W_Q^{(1)} Z) \Big) \right]_{:,n+1} = \#\{\text{depth of } p_n\} o_1 + o_2.$$

It is then easy to check $\text{LN}\left( \left[ W_V^{(1)} Z\sigma\Big( \mathcal{C} \cdot (W_K^{(1)} Z)^\top (W_Q^{(1)} Z) \Big) \right]_{:,n+1} \right) + Z_{:,n+1}$ is uniquely determined by the type and depth of $p_n$ without repetition. Then by Lemma 15, there exists a 2-layer ReLU MLP with width $O(k^2 D^2)$ that can map $\text{LN}\left( \left[ W_V^{(1)} Z\sigma\Big( \mathcal{C} \cdot (W_K^{(1)} Z)^\top (W_Q^{(1)} Z) \Big) \right]_{:,n+1} \right) + Z_{:,n+1}$ to the embedding in Equation (15). It is then easy to see that the condition in Theorem 1 is satisfied as $W_K^{(2)} = W_Q^{(2)} = 0$. Hence the second layer can be constructed to let the Transformer to output the correct next token probability. □

## C.4 Approximate Balance Condition For Finite Length Training Data

Theorem 1 assumes the model reaches the optimal loss for Dyck prefixes of any length. However, in practice, due to finite samples and various sources of randomness, training often does not end exactly at a population optima. In this case, the condition in Theorem 1 is not precisely met. However, even for models that *approximately* meet those conditions, we will prove that when the second-layer projection function $\text{g}^{(2)}$ is Lipschitz, a similar condition as in Equation (14) is still necessary.

We will show this by bounding the amount of deviations from the perfect balance. The idea is that for two long prefixes that differ in only the last open bracket, correct next token prediction requires the Transformer outputs on these prefixes to be sufficiently different, hence the part irrelevant to the prediction (i.e. matched brackets) should not have a large contribution.

To formalize this intuition, let's define two quantities: 1) $S_{d,d',i,j}$ which measures the effect from one matching pair, and 2) $P_{d,j}$ which measures the effect on the last position from all tokens in a prefix.

---

[17]Here the starting token is necessary because otherwise, the Transformer with uniform attention will have the same outputs for prefix $p$ and prefix $p \oplus p$, in which $\oplus$ denotes concatenation, i.e. $p \oplus p$ means the same string $p$ repeated twice.

Let $u$ be defined as in Equation (12). $S_{d,d',i,j}$ is defined as

$$S_{d,d',i,j}[\theta^{(2)}] = u(\tau_{2j,d}, \tau_{2i,d'-1}) + u(\tau_{2j,d}, \tau_{2i-1,d'}), \tag{31}$$

which measures how much a matching pair of brackets $(\tau_{2i,d'-1}, \tau_{2i-1,d'})$ changes the input to the LayerNorm upon seeing the last token $\tau_{2j,d}$. Note that under the perfect balance condition, $S_{d,d',i,j}[\theta^{(2)}] = 0$.

The second quantity $P_{d,j}[\theta^{(2)}]$ is defined via an intermediate quantity $Q(2j, d, \tilde{t})$: for any $i \in [k], d \in [D]$ and a length-$(d-1)$ prefix $\tilde{t} \in [2k]^{d-1}$, $Q(i, d, \tilde{t})$ is defined as

$$Q(i, d, \tilde{t}) := u(\tau_{2i,d-1}, t_{\mathcal{S}}) + \sum_{1 \le d' < d} u(\tau_{2i,d-1}, \tau_{\tilde{t}_{d'},d'}) \tag{32}$$
$$+ u(\tau_{2i,d-1}, \tau_{2i-1,d}) + u(\tau_{2i,d-1}, \tau_{2i,d-1}),$$

where $\tilde{t}_{d'}$ denotes the $d'_{th}$ entry of $\tilde{t}$. Intuitively, $Q(i, d, \tilde{t})$ denotes the unnormalized second-layer attention output at the last position, given the input sequence $\tilde{t} \oplus \tau_{2i-1,d}\tau_{2i,d-1}$, [18]

For results in this subsection, it suffices to consider prefixes consisting only of open brackets. Let $t := \arg\min_{\tilde{t} \in \{2i-1\}_{i \in [k]}^{d-1}} \|Q(2j, d, \tilde{t})\|_2$, and let $t'$ denote the prefix that minimizes $\|Q(2j, d, \tilde{t})\|_2$ subject to the constraint that $t'$ differs from $t$ at the last (i.e. $(d-1)_{th}$) position, i.e.

$$t' = \arg\min_{\tilde{t}' \in \{2i-1\}_{i \in [k]}^{d-1}, t'_{d-1} \neq t_{d-1}} Q(2j, d, \tilde{t}').$$

Such choices of $t, t'$ guarantees that the two prefixes differ at the last open bracket and hence must have different next-word distributions. Finally, define

$$P_{d,j}[\bar{\theta}^{(2)}] = \|Q(2j, d, t')\|. \tag{33}$$

In the following theorem, $P_{d,j}$ will be used as a reference to upper bound $S_{d,d',i,j}[\theta^{(2)}]$, meaning that the model should not be sensitive to the insertion of a matching pair of brackets.

**Theorem 3** (Approximate Balance). *Consider a 2-layer Transformer $\mathcal{T}$ (Equation (10)) with a minimal first layer (Assumption 1) and a $\gamma$-Lipschitz $g^{(2)}$ for $\gamma > 0$, trained on sequences of length $N$ with the mean squared loss (Equation (5)).*

*Suppose the loss is approximately optimal, namely, the set of second-layer weights $\bar{\theta}_N^{(2)}$ satisfies $\mathcal{L}(\mathcal{T}[\bar{\theta}_N^{(2)}], \mathcal{D}_{q,k,D,N}) \le q^{-N}\epsilon$, for any positive integer $N > 8D$ and sufficiently small $\epsilon > 0$. Then, there exists a constant $C_{\gamma,\epsilon,D}$, such that $\forall 0 \le d' \le D, 1 \le d \le D, i, j \in [k]$, it holds that*

$$\|S_{d,d',i,j}[\bar{\theta}_N^{(2)}]\| \le \frac{C_{\gamma,\epsilon,D}}{N} P_{d,j}[\bar{\theta}_N^{(2)}]. \tag{34}$$

Intuitively, Theorem 3 states that when the loss $\mathcal{L}(\theta)$ is sufficiently small, $S_{d,d',i,j}[\theta^{(2)}]$ must be small relative to $P_{d,j}[\bar{\theta}_N^{(2)}]$. Inequality 34 can be interpreted as a relaxation of Equation (14), which is equivalent to $S_{d,d',i,j}[\theta^{(2)}] = 0$. The proof of Theorem 3 shares a similar intuition as Theorem 1 and is given in Appendix C.4.1.

A direct corollary of Theorem 3 additionally consider weight decay, in which case approximate balance condition still holds, as the regularization strength goes to 0:

**Corollary 3.** *Consider the setting where a Transformer with a fixed minimal first layer is trained to minimize $\mathcal{L}_\lambda^{reg} = \mathcal{L}_\theta(x) + \lambda \frac{\|\theta\|_2^2}{2}$, which is the squared loss with $\lambda$ weight decay. Suppose $g^{(1)}$ of the Transformer is a 2-layer fully connected network and $g^{(2)}$ of the Transformer is a 6-layer fully connected network. Then, there exists constant $C > 0$, such that if a set of parameters $\theta_{\lambda,N}$ minimizes $\mathcal{L}_\lambda^{reg}$, then it holds $\forall 0 \le d' \le D, 1 \le d \le D, i, j \in [k]$ that,*

$$\forall N, \exists \lambda_N, \text{ such that } \forall \lambda \in [0, \lambda_N], S_{d,d',i,j}[\theta_{\lambda,N}^{(2)}] \le \frac{C}{N} P_{d,i}[\theta_{\lambda,N}^{(2)}].$$

---

[18]We use $s \oplus t$ to denote the concatenation of two strings $s, t$, same as in Equation (15)-(16), and use $\tau_i\tau_j$ to denote the concatenation of two tokens $\tau_i, \tau_j$.

### C.4.1 Proof of Theorem 3

*Proof.* The key idea is similar to the proof of necessity in Theorem 1. That is, we will construct two input sequences with different next-word distributions, and show that the approximate balance condition must hold so that inserting (a bounded number of) pairs of matching brackets does not collapse the two predicted distributions given by the Transformer.

**Constructing the input sequences.**

Let $\boldsymbol{t} := \arg\min_{\tilde{\boldsymbol{t}} \in [k]^{d-1}} \|Q(2j, d, \tilde{\boldsymbol{t}})\|_2$, and let $\boldsymbol{t}'$ denote the prefix that minimizes $\|Q(2j, d, \tilde{\boldsymbol{t}})\|_2$ subject to the constraint that $\boldsymbol{t}'$ must differ from $\boldsymbol{t}$ in the last (i.e. $(d-1)_{th}$) position, i.e.

$$\boldsymbol{t}' = \arg\min_{\tilde{\boldsymbol{t}}' \in [k]^{d-1}, \boldsymbol{t}'_{d-1} \neq \boldsymbol{t}_{d-1}} Q(2j, d, \tilde{\boldsymbol{t}}').$$

The motivation for such choices of $\boldsymbol{t}, \boldsymbol{t}'$ is that since they differ at least by the last position which is an open bracket, they must lead to different next-word distributions. Note also that $P_{d,j}[\bar{\theta}^{(2)}] = \|Q(2j, d, \boldsymbol{t}')\|$.

With the above definition of $\boldsymbol{t}, \boldsymbol{t}'$, consider two valid Dyck prefixes $p_1$ and $p_2$ with length no longer than $N$, defined as follows: for any $d, d' \in [D], i, j \in [k]$, consider a common prefix $p = \underbrace{\tau_{2i-1} \ldots \tau_{2i-1}}_{d' \text{ open brackets}} \underbrace{\tau_{2i-1}\tau_{2i} \ldots \tau_{2i-1}\tau_{2i}}_{(\lfloor \frac{N}{2} \rfloor - d' - d - 1) \text{ pairs}} \underbrace{\tau_{2i} \ldots \tau_{2i}}_{d' \text{ closed brackets}}$, where $\tau_i$ denotes a token with type $i$ whose depth is implicit from the context. Set $p_1, p_2$ as

$$p_1 = p \oplus \boldsymbol{t} \oplus \tau_{2j-1}\tau_{2j},$$
$$p_2 = p \oplus \boldsymbol{t}' \oplus \tau_{2j-1}\tau_{2j}.$$

That is, $p_1, p_2$ differ in the last unmatched open bracket. In the following, we will show that the approximate balance condition must hold for the predictions on $p_1, p_2$ to be sufficiently different.

**Bounding the difference in Transformer outputs.** For a Transformer $\mathcal{T}$ with second layer parameters $\bar{\theta}_N^{(2)}$, its outputs on $p_1, p_2$ satisfy

$$\|\mathcal{T}[\bar{\theta}_N^{(2)}](p_1) - \mathcal{T}[\bar{\theta}_N^{(2)}](p_2)\|_2 \tag{35}$$

$$\geq \|p_1 - p_2\|_2 - \|\mathcal{T}[\bar{\theta}_N^{(2)}](p_1) - p_1\|_2 - \|\mathcal{T}[\bar{\theta}_N^{(2)}](p_2) - p_2\|_2 \tag{36}$$

$$\geq \frac{1}{\sqrt{2k}}\|p_1 - p_2\|_1 - \left(\|\mathcal{T}[\bar{\theta}_N^{(2)}](p_1) - p_1\|_1 + \|\mathcal{T}[\bar{\theta}_N^{(2)}](p_2) - p_2\|_1\right) \tag{37}$$

$$\geq \frac{1}{\sqrt{2k}}\text{TV}(p_1, p_2) - o_\epsilon(1) \quad (\text{since } \mathcal{L}(\mathcal{T}[\bar{\theta}_N^{(2)}], \mathcal{D}_{q,k,D,N}) \leq q^{-N}\epsilon) \tag{38}$$

$$= \Omega(1), \tag{39}$$

where $\text{TV}(p_1, p_2)$ denotes the TV distance in the next-word distributions from $p_1$ and $p_2$, and $o_\epsilon(1)$ means the term will go to zero as $\epsilon$ goes to zero. The TV distance is lower bounded by the construction of $p_1, p_2$, where $\boldsymbol{t}, \boldsymbol{t}'$ differ at the last open bracket. The error $\epsilon$ is upper bounded because of the assumption on $\bar{\theta}_N^{(2)}$, i.e. $\mathcal{L}(\mathcal{T}[\bar{\theta}_N^{(2)}], \mathcal{D}_{q,k,D,N}) \leq q^{-N}\epsilon$ with sufficiently small $\epsilon$.

Define by $A_p$ the contribution of $p$ to the attention output (before LayerNorm) of the last position of $p_1, p_2$:

$$A_p = \sum_{1 \leq d'' < d'} \left(u(\tau_{2j,d-1}, \tau_{2i,d''-1}) + u(\tau_{2j,d-1}, \tau_{2i-1,d''})\right)$$
$$+ \lfloor \frac{N - 2d' - 2d}{2} \rfloor \left(u(\tau_{2j,d-1}, \tau_{2i,d'}) + u(\tau_{2j,d-1}, \tau_{2i-1,d'+1})\right). \tag{40}$$

The attention outputs (before LayerNorm) of $p_1, p_2$, denoted by $A(p_1)$ and $A(p_2)$, satisfy that

$$\mathcal{P}_\perp A(p_1) = \mathcal{P}_\perp(A_p + Q(2j, d, \boldsymbol{t})),$$
$$\mathcal{P}_\perp A(p_2) = \mathcal{P}_\perp(A_p + Q(2j, d, \boldsymbol{t}')). \tag{41}$$

Note that for any prefix $p'$,

$$\mathcal{T}[\bar{\theta}_N^{(2)}](p') = \mathrm{g}^{(2)}\big(\mathrm{LN}_{C_{LN}}(\mathcal{P}_\perp A(p'))\big) + \boldsymbol{e}(\tau_{2i,d'}) \tag{42}$$

$$= \mathrm{g}^{(2)}\Big(\frac{\mathcal{P}_\perp A(p')}{\|\mathcal{P}_\perp A(p')\|}\Big) + \boldsymbol{e}(\tau_{2i,d'}), \tag{43}$$

where $\mathrm{g}^{(2)}$ is $\gamma$-Lipschitz. Hence the Lipschitz constant with respect to and we have

$$\Big\|\frac{\mathcal{P}_\perp A(p_1)}{\|\mathcal{P}_\perp A(p_1)\|_2} - \frac{\mathcal{P}_\perp A(p_2)}{\|\mathcal{P}_\perp A(p_2)\|_2}\Big\|_2 \geq \frac{\mathrm{TV}(p_1,p_2) - o_\epsilon(1)}{\sqrt{2k}\gamma} = \Omega_{\frac{1}{\gamma},\epsilon}(1). \tag{44}$$

We show that $A_p$ should not be too much larger in norm than $Q(2j,d,\boldsymbol{t})$ or $Q(2j,d,\boldsymbol{t}')$. First, let's state a helper lemma about the contrapositive:

**Lemma 1.** *For any $\epsilon > 0$, there exists a constant $R_\epsilon$, such that for any $a,b \in \mathbb{R}^d$ and any $r \in \mathbb{R}^d$ such that $\|r\|_2 \geq R_\epsilon \cdot \max\{\|a\|_2, \|b\|_2\}$, it holds that*

$$\Big\|\frac{a+r}{\|a+r\|_2} - \frac{b+r}{\|b+r\|_2}\Big\|_2 \leq \epsilon.$$

*Proof.* Denote $r_0 := \max\{\|a\|_2, \|b\|_2\}$. Then $R_\epsilon := \frac{4r_0}{\epsilon} + 1$ suffices:

$$\Big\|\frac{r+a}{\|r+a\|_2} - \frac{r+b}{\|r+b\|_2}\Big\| \leq \|r\| \cdot \Big|\frac{1}{\|r+a\|} - \frac{1}{\|r+b\|}\Big| + \frac{\|a\|}{\|r+a\|} + \frac{\|b\|}{\|r+b\|}$$

$$\leq \|r\| \cdot \Big(\frac{1}{\|r\|-r_0} - \frac{1}{\|r\|+r_0}\Big) + \frac{2r_0}{\|r\|-r_0}$$

$$= \frac{2r_0}{\|r\|-r_0} \cdot \Big(\frac{\|r\|}{\|r\|+r_0} + 1\Big) \leq \frac{4r_0}{\|r\|-r_0} \leq \frac{4r_0}{R_\epsilon - r_0} \leq \epsilon.$$

$\square$

Lemma 1 implies that if $A_p$ is too large, then the output on $p_1, p_2$ (Equation (44)) won't be sufficiently different. Let $P_{d,j}[\bar{\theta}_N^{(2)}]$ be defined as in Equation (33) and let $R_\epsilon$ be the constant in Lemma 1, we need to bound $\|\mathcal{P}_\perp A_p\|$ by

$$\|\mathcal{P}_\perp A_p\|_2 \leq R_\epsilon \|P_{d,j}[\bar{\theta}_N^{(2)}]\|_2. \tag{45}$$

As Equation (45) holds for $p$ with any $d, d'$, if one choose $d' = 1$, this shows

$$\|u(\tau_{2j,d-1}, \tau_{2i,1}) + u(\tau_{2j,d-1}, \tau_{2i-1,2})\|_2 \leq \frac{4R_\epsilon \|P_{d,j}[\bar{\theta}_N^{(2)}]\|_2}{N}. \tag{46}$$

Further, it holds that for any $1 < d' \leq d - 1$,

$$\Big\| \sum_{1 \leq d'' < d'} \big(u(\tau_{2j,d-1}, \tau_{2i,d''-1}) + u(\tau_{2j,d-1}, \tau_{2i-1,d''})\big)$$

$$+ \Big\lfloor\frac{N - 2d' - 2d}{2}\Big\rfloor \big(u(\tau_{2j,d-1}, \tau_{2i,d'}) + u(\tau_{2j,d-1}, \tau_{2i-1,d'+1})\big) \Big\|_2 \leq R_\epsilon \|P_{d,j}[\bar{\theta}_N^{(2)}]\|_2.$$

$$\Big\| \sum_{1 \leq d'' < d'+1} \big(u(\tau_{2j,d-1}, \tau_{2i,d''-1}) + u(\tau_{2j,d-1}, \tau_{2i-1,d''})\big)$$

$$+ \Big\lfloor\frac{N - 2d' - 2d - 2}{2}\Big\rfloor \big(u(\tau_{2j,d-1}, \tau_{2i,d'+1}) + u(\tau_{2j,d-1}, \tau_{2i-1,d'+2})\big) \Big\|_2 \leq R_\epsilon \|P_{d,j}[\bar{\theta}_N^{(2)}]\|_2.$$

The triangle inequality then yields,

$$\lfloor \frac{N - 2d' - 2d - 2}{2} \rfloor \| \left( u(\tau_{2j,d-1}, \tau_{2i,d'+1}) + u(\tau_{2j,d-1}, \tau_{2i-1,d'+2}) \right)$$
$$- \left( u(\tau_{2j,d-1}, \tau_{2i,d'}) + u(\tau_{2j,d-1}, \tau_{2i-1,d'+1}) \right) \|_2 \leq 2R_\epsilon \| P_{d,j}[\bar{\theta}_N^{(2)}] \|_2.$$

Because $N \geq 8D$, we have that $\lfloor \frac{N-2d'-2d-2}{2} \rfloor \geq \frac{N}{8}$, hence it holds that

$$\| \left( u(\tau_{2j,d-1}, \tau_{2i,d'+1}) + u(\tau_{2j,d-1}, \tau_{2i-1,d'+2}) \right)$$
$$- \left( u(\tau_{2j,d-1}, \tau_{2i,d'}) + u(\tau_{2j,d-1}, \tau_{2i-1,d'+1}) \right) \|_2 \leq \frac{16 R_\epsilon \| P_{d,j}[\bar{\theta}_N^{(2)}] \|_2}{N}.$$

Combined with Equation (46), one can conclude that,

$$S_{d,d',i,j} = \| u(\tau_{2j,d-1}, \tau_{2i,d'-1}) + u(\tau_{2j,d-1}, \tau_{2i-1,d'-1}) \| \leq \frac{16 D R_\epsilon}{N} \| P_{d,j}[\bar{\theta}_N^{(2)}] \|_2. \quad (47)$$

The proof is then completed. $\qquad \square$

*Proof of Corollary 3.* This proof is in fact a direct combination of Theorems 1 and 3. By Theorem 1 we know there exists a weight $\theta^{(2)*}$ that can reach zero loss for arbitrarily length $N$. Then it holds that $\|\theta_{\lambda,N}\|_2 \leq \|\theta^{(2)*}\|$ as $\theta_{\lambda,N}$ minimizes the regularized loss. Noticing that bounded weight implies bounded Lipschitzness of $g^{(2)}$, the rest follows as Theorem 3. $\qquad \square$

## C.5 Proof of Theorem 2

We now show the limitation of interpretability from a single component, using a Lottery-Ticket-style argument by pruning from large random Transformers.

**Theorem 2** (Indistinguishability From a Single Component). *Consider any $L$-layer Transformer $\mathcal{T}$ (Equation (10)) with embedding dimension $m$, attention dimension $m_a$, and projection function g as 2-layer ReLU MLP with width $w$. For any $\delta \in (0,1)$ and $N \in \mathbb{N}^+$, consider a $4L$-layer random Transformer $\mathcal{T}_{large}$ with embedding dimension $m_{\text{large}} = O(m \log(Lm/\delta))$, attention dimension $m_{\text{large},a} = O(m_a L \log \frac{m_a m L N}{\epsilon \delta})$, and projection function $g_{\text{large}}$ as 4-layer ReLU MLP with width $w_{\text{large}} = O(\max\{m,w\} L \log \frac{wmLN}{\epsilon \delta})$.*

*Assume that $\|W\|_2 \leq 1$ for every weight matrix $W$ in $\mathcal{T}$, and suppose the weights are randomly sampled as $W_{i,j} \sim U(-1,1)$ for every $W \in \mathcal{T}_{large}$. Then, with probability $1-\delta$ over the randomness of $\mathcal{T}_{large}$, there exists a nonstructural pruning (Definition 4) of $\mathcal{T}_{large}$, denoted as $\tilde{\mathcal{T}}_{large}$, which $\epsilon$-approximates $\mathcal{T}$ with respect to $\| \cdot \|_{1,2}$ for any input $X \in \mathbb{R}^{m \times N}$ satisfying $\|\boldsymbol{X}\|_{1,2} \leq 1$.* [19]

*Proof.* We will first introduce some notation. For vector $x \in \mathbb{R}^a$ and $y \in \mathbb{R}^b$, we will use $x \oplus y$ to denote their concatenation. We will use $0^a$ to denote the all-zero vector with dimension $a$. We will also assume without loss of generality that $w \geq 2m$. [20]

We will use $\bar{\boldsymbol{X}}$ to denote $\begin{bmatrix} \boldsymbol{X} \\ 0^{(m_{\text{large}} - m') \times N} \end{bmatrix}$ for $\boldsymbol{X} \in \mathbb{R}^{m' \times N}$ with $m' \leq m_{\text{large}}$.

In the following, a *random network* refers to a network whose weights have entries sampled from a uniform distribution, i.e. $W_{i,j} \sim U(-1,1)$ for every weight $W$ in the random network.

We will first recall Lemma 2 from Pensia et al. (2020) which shows that a pruned 2-layer random network can approximate a linear function.

**Lemma 2** (Approximating a linear function; Theorem 1 of Pensia et al. (2020) restated). *Let $W \in \mathbb{R}^{m' \times m}, \|W\|_2 = O(1)$, then for $\sigma \in \{\text{ReLU}, \mathcal{I}\}$, where $\mathcal{I}$ represents the identity operator, for a random network $g(x) = W_2 \sigma(W_1 x)$ with $W_2 \in \mathbb{R}^{m' \times h}, W_1 \in \mathbb{R}^{h \times m}$ for hidden dimension*

---

[19] Here the input and output dimension of $\tilde{\mathcal{T}}_{\text{large}}$ is actually $m_{\text{large}}$ which is larger than $m$; additional dimensions are padded with zeroes. The norm constraint can be easily extended to an arbitrary constant.

[20] We can always pad dimensions if $w$ is too small.

$h = O(m \log(\frac{mm'}{\min\{\epsilon,\delta\}}))$, with probability $1 - \delta$, there exists boolean masking matrices $M_1, M_2$, such that for any $x \in \mathbb{R}^w$,

$$\|(M_2 \odot W_2)\sigma\big((M_1 \odot W_1)x\big) - Wx\| \le \epsilon\|x\|_2,$$

where $\odot$ denotes the Hadamard product.

We then derive two approximation results Lemmas 3 and 4 based on Lemma 2.

**Lemma 3.** *Under the setting of Theorem 2, with probability $1 - 2\delta/3$, for any $l \in [L], l' \in [4L-1]$, let $\mathcal{T}^{(l)}$ be the $l$-th layer of $\mathcal{T}$, there exists a pruning of the $(l'-1)$-th and the $(l')$-th layer $\mathcal{T}_{large}^{(l'-1)}, \mathcal{T}_{large}^{l'}$, named $\tilde{\mathcal{T}}_{large}^{(l'-1)}, \tilde{\mathcal{T}}_{large}^{l'}$ such that when defined on domain $\|\boldsymbol{X}\|_{1,2} \le 2L, \boldsymbol{X} \in \mathbb{R}^{m \times N}$,*

1. *$\tilde{\mathcal{T}}_{large}^{(l'-1)}$ is independent of the last $m_{large} - m$ rows of the input.*

2. *$\tilde{\mathcal{T}}_{large}^{l'} \circ \tilde{\mathcal{T}}_{large}^{(l'-1)}\big(\bar{\boldsymbol{X}}\big)$ is an $\big(\frac{C}{1000L^2}\big)^{4L-3} \epsilon$-approximation of $\overline{\mathcal{T}^{(l)}(\boldsymbol{X})}$ with respect to $1, 2$-norm.*

**Lemma 4.** *Under the setting of Theorem 2, for any matrix $W \in \mathbb{R}^{4m \times 4m}, \|W\|_2 \le 1$, with probability $1 - \delta/4$, for any $l' \in [4L]$, there exists a pruning of the $l$-th layer $\mathcal{T}_{large}^{(l')}$, named $\tilde{\mathcal{T}}_{large}^{(l')}$, such that when defined on domain $\boldsymbol{X} \in \mathbb{R}^{m \times N}$,*

1. *$\tilde{\mathcal{T}}_{large}^{(l')}$ is independent of the last $m_{large} - 4m$ rows of the input.*

2. *$a(x) = \tilde{\mathcal{T}}_{large}^{(l')}\big(\bar{\boldsymbol{X}}\big)$ is an $\big(\frac{C}{1000L^2}\big)^{4L} \epsilon$-approximation of $\hat{g}(\boldsymbol{X}) = \overline{W\boldsymbol{X}}$ with respect to $1, 2$-norm.*

The proof of Lemmas 3 and 4 is deferred to Appendix C.5.1 We can now prove the theorem.

We will first show with induction that if we 1) prune the $(2l-1)$-th and $2l$-th layers of $\mathcal{T}_{large}$ to approximate $\mathcal{T}^{(l)}$ for each $l \in [L]$, and 2) prune the $2L+1$ to $4L$-th layers of $\mathcal{T}_{large}$ to approximate identity, then the pruned large transformer will be an $\epsilon$-approximation of $\mathcal{T}$ for any input $\|\boldsymbol{X}\|_{1,2} \le 1$.

We will perform induction on $l$: Let $\mathcal{T}^{(1:l)}$ define the composition of layer 1 to $l$, i.e. $\mathcal{T}^{(1:l)}(\boldsymbol{X}) := \mathcal{T}^{(l)} \circ \mathcal{T}^{(l-1)} \circ \cdots \circ \mathcal{T}^{(1)}(\boldsymbol{X})$, and define $\epsilon_l := \big(\frac{C}{1000L^2}\big)^{4L-3-l} \epsilon$. Suppose that $\mathcal{T}_{large}^{(1:2l)}$ is an $\epsilon_l$-approximation of $\mathcal{T}^{(1:l)}$. Note that $\|\mathcal{T}^{(1:l)}(\boldsymbol{X})\|_{1,2} \le (l+1)$, since each attention output has a bounded norm of 1 and every weight matrix in projection function g has spectral norm smaller than 1, hence the norm will at most increment 1 (due to residual connection) after each layer. We have that

$$\left\|\tilde{\mathcal{T}}_{large}^{(1:2l)}\big(\bar{\boldsymbol{X}}\big)\right\|_{1,2} \le 4l \le 4L.$$

Then according to Lemma 13, $\mathcal{T}^{(l+1)}$ is $(1 + 200L^2/C)$-Lipschitz on the set of intermediate outputs $\{\big(\tilde{\mathcal{T}}_{large}^{(1:2l)}(\bar{\boldsymbol{X}})\big)_{1:m} \mid \|\boldsymbol{X}\|_{1,2} \le 1\}$. We also have that $\mathcal{T}^{(1:l)}(\boldsymbol{X})$ is $(1 + 200L^2/C)^l$-Lipschitz. Now we can apply Lemma 5 to show that $\mathcal{T}_{large}^{(1:2l+2)}$ can $\epsilon'$-approximate $\mathcal{T}^{(1:l+1)}$ with

$$\epsilon' = \epsilon_l(1 + 200L^2/C) + \epsilon\left(\frac{C}{1000L^2}\right)^{4L-3}(1 + 200L^2/C)^l + \epsilon_l\left(\frac{C}{1000L^2}\right)^{4L-3}\epsilon$$

$$\le \left(\frac{C}{1000L^2}\right)^{4L-4-l}\epsilon = \epsilon_{l+1}.$$

The induction is then completed and we have the composition of $\tilde{\mathcal{T}}_{large}^i$ for $i \in [2L]$ $\epsilon_L$-approximates the composition of $\mathcal{T}$ with $\epsilon_L = \big(\frac{C}{1000L^2}\big)^{3L-3}\epsilon$. We will then perform another induction showing that the composition of $\tilde{\mathcal{T}}_{large}^i$ for $i \in [2L+l]$ $\epsilon_{l+L}$-approximates $\mathcal{T}$ with $\epsilon_{l+L} = \big(\frac{C}{1000L^2}\big)^{3L-3-l}\epsilon$. Suppose the statement holds for $L - 1 \ge l \ge 0$.

The induction step is similar, because we have $\mathcal{T}$ is $(1 + 200L^2/C)^L$ Lipschitz, by Lemma 5, it holds that the composition of $\mathcal{T}_{\text{large}}^i$ for $i \in [2L + l + 1]$ $\epsilon'$-approximates $\mathcal{T}$ with,

$$\epsilon' = \epsilon_{l+L} + \epsilon \left(\frac{C}{1000L^2}\right)^{4L} (1 + 200L^2/C)^L + \epsilon_{l+L}\epsilon \left(\frac{C}{1000L^2}\right)^{4L}$$

$$\leq \epsilon \left(\frac{C}{1000L^2}\right)^{3L-4-l} \epsilon = \epsilon_{L+l+1}.$$

This concludes the induction and prove the first claim of the theorem. For the second claim, notice that through similar induction steps, we can prune arbitrary layer of $\mathcal{T}_{\text{large}}$ to approximate identity function and obtain the same approximation rate, this concludes the proof for the second claim. $\square$

### C.5.1 Helper lemmas for Theorem 2

**Error Analysis** Our first lemma shows that the composition of $\epsilon$-approximation can approximate the composition of the original function.

**Lemma 4.** *Under the setting of Theorem 2, for any matrix $W \in \mathbb{R}^{4m \times 4m}, \|W\|_2 \leq 1$, with probability $1 - \delta/4$, for any $l' \in [4L]$, there exists a pruning of the $l$-th layer $\mathcal{T}_{\text{large}}^{(l')}$, named $\tilde{\mathcal{T}}_{\text{large}}^{(l')}$, such that when defined on domain $\boldsymbol{X} \in \mathbb{R}^{m \times N}$,*

1. *$\tilde{\mathcal{T}}_{\text{large}}^{(l')}$ is independent of the last $m_{\text{large}} - 4m$ rows of the input.*

2. *$a(x) = \tilde{\mathcal{T}}_{\text{large}}^{(l')}(\bar{\boldsymbol{X}})$ is an $\left(\frac{C}{1000L^2}\right)^{4L}\epsilon$-approximation of $\hat{g}(\boldsymbol{X}) = \overline{W\boldsymbol{X}}$ with respect to $1, 2$-norm.*

*Proof.* One can prune the value matrix on layer $l'$ to zero and the rest is a direct consequence of Lemmas 2 and 20. $\square$

**Lemma 5.** *Given three metric spaces $A, B, C$ equipped with same metric $\| \cdot \|$. Suppose $f_1 : A \to B, f_2 : B \to C$ are $\epsilon_1, \epsilon_2$-approximations of $g_1, g_2$ with respect to $\| \cdot \|$, where $g_1$ is a Lipschitz function with constant $\lambda_1$ with respect to $\| \cdot \|$ and $\|g_2(x)\| \leq \lambda_2 x$, then it holds that, $f_1 \circ f_2$ is an $\epsilon'$-approximation of $g_1 \circ g_2$, with $\epsilon' = (\lambda_2 + \epsilon_1)(\lambda_1 + \epsilon_2) - \lambda_1 \lambda_2$*

*Proof.* For any $x \in \mathbb{R}^{d_1}$, it holds that,

$$\|f_1(x) - g_1(x)\| \leq \epsilon_1 \|x\|.$$

This then suggests that,

$$\begin{aligned}
&\|f_2(f_1(x)) - g_2(g_1(x))\| \\
\leq &\|f_2(f_1(x)) - g_2(f_1(x))\| + \|g_2(f_1(x)) - g_2(g_1(x))\| \\
\leq &\epsilon_2 \|f_1(x)\| + \lambda_2 \|f_1(x) - g_1(x)\| \\
\leq &\epsilon_2 \|g_1(x)\| + (\lambda_2 + \epsilon_2)\|f_1(x) - g_1(x)\| \\
\leq &(\epsilon_2\lambda_1 + \epsilon_1\lambda_2 + \epsilon_1\epsilon_2)\|x\|
\end{aligned}$$

$\square$

**Approximating ReLU MLP** We will first show an extension of Lemma 2, illustrating that a pruned wide 4-layer ReLU MLP can approximate any 2-layer ReLU MLP.

**Lemma 6.** *Consider any 2-layer ReLU MLP $g : \mathbb{R}^{4m} \to \mathbb{R}^{4m}$ parameterized by $W_1 \in \mathbb{R}^{4m \times w}, W_2 \in \mathbb{R}^{w \times 4m}, \|W_1\|_2, \|W_2\|_2 \leq 2\sqrt{2}$, for any $\delta, \epsilon \in (0, 1)$, consider a random 4-layer ReLU MLP $f$ with input and output dimension $4m$ and width $w' = O(w \log(\frac{wm}{\min\{\epsilon, \delta\}}))$ parameterized by $W_{\text{large},i}$, with probability $1 - \delta$ over the randomness of weight of $f$, there exists a nonstructural pruning of $f$ named $\tilde{f}$, such that $\tilde{f}$ is an $\epsilon$-approximation of $f$ with respect to $2$-norm.*

*Proof.* Choose $\epsilon_0 = \epsilon/8$. We only need to show there exists boolean matrices $M_1, M_2, M_3, M_4$, such that,

$$\left\| \left( M_4 \odot W_{\mathrm{large},4} \mathrm{ReLU}\left( (M_3 \odot W_{\mathrm{large},3}) \mathrm{ReLU}\left( (M_2 \odot W_{\mathrm{large},2}) \mathrm{ReLU}\left( (M_1 \odot W_{\mathrm{large},1}) x \right) \right) \right) \right) \right.$$
$$\left. - W_2 \mathrm{ReLU}\left( W_1 \boldsymbol{X} \right) \right\|_2 \le \epsilon.$$

By Lemma 2, there exists boolean matrices $M_1 \in \mathbb{R}^{w' \times 4m}$ and $M_2' \in \mathbb{R}^{w \times w'}$, such that for any $x \in \mathbb{R}^{4m}$,

$$\left\| \left( \begin{bmatrix} M_2' \\ 0^{(w'-w) \times w'} \end{bmatrix} \odot W_{\mathrm{large},2} \right) \mathrm{ReLU}\left( (M_1 \odot W_{\mathrm{large},1}) x \right) - \begin{bmatrix} W_1 x \\ 0^{w'-w} \end{bmatrix} \right\|_2 \le \epsilon_0 \|x\|_2.$$

Hence we can choose $M_2 = \begin{bmatrix} M_2' \\ 0^{(w'-w) \times w'} \end{bmatrix}$ and have $f_1(x) = \mathrm{ReLU}\left( (M_2 \odot W_{\mathrm{large},2}) \mathrm{ReLU}\left( (M_1 \odot W_{,1}) x \right) \right)$ is $\epsilon_0$-approximation of $g_1(x) = \begin{bmatrix} \mathrm{ReLU}(W_1 x) \\ 0^{w'-w} \end{bmatrix}$.

Again by Lemma 2, there exists boolean matrices $M_3' \in \mathbb{R}^{w' \times w}$ and $M_4 \in \mathbb{R}^{4m \times w'}$, such that for any $y \in \mathbb{R}^w$,

$$\| (M_4 \odot W_{\mathrm{large},4}) \mathrm{ReLU}\left( [M_3', 0^{w' \times (w'-w)}] \begin{bmatrix} y \\ 0^{w'-w} \end{bmatrix} \right) \le \epsilon_0 \|y\|_2$$

Hence we can choose $M_3 = [M_3', 0^{w' \times (w'-w)}]$, and have $f_2(x) = \mathrm{ReLU}\left( (M_4 \odot W_{\mathrm{large},4}) \mathrm{ReLU}\left( (M_3 \odot W_{\mathrm{large},3}) x \right) \right)$ is $\epsilon_0$-approximation of $g_2(x) = W_2 x$.

It is also easy to check $g_1$ and $g_2$ are both $2\sqrt{2}$-lipschitz and $g_1(0) = 0$. By Lemma 5, we conclude that $\tilde{f} = f_1 \odot f_2$ is $\epsilon'$-approximation of $g = g_1 \odot g_2$, with $\epsilon' = 4\sqrt{2}\epsilon_0 + \epsilon_0^2 \le \epsilon$. $\qquad\square$

This lemma then yields the following corollaries.

**Corollary 1.** *Under the setting of Theorem 2, with probability $1 - \delta/4$, for any $l \in [L], l' \in [4L]$, there exists a pruning of the projection function $\mathrm{g}_{\mathrm{large}}^{(l')}$, named $\mathrm{g}_{\mathrm{large}}^{(\tilde{l}')}$, such that*

1. *$\mathrm{g}_{\mathrm{large}}^{(\tilde{l}')}$ is independent of the last $m_{\mathrm{large}} - m$ dimension of the input.*

2. *$a(x) = \mathrm{g}_{\mathrm{large}}^{(\tilde{l}')} \left( \begin{bmatrix} x \\ 0^{m_{\mathrm{large}} - m} \end{bmatrix} \right)$ is an $\left( \frac{C}{1000L^2} \right)^{4L} \epsilon$-approximation of $\hat{g}(x) = \begin{bmatrix} \mathrm{g}^{(l)}(x) \\ 0^{m_{\mathrm{large}} - m} \end{bmatrix}$ with respect to $2-$norm.*

*Proof.* One can construct such pruning by pruning the last $m_{\mathrm{large}} - m$ rows of the weight of the last layer and the last $m_{\mathrm{large}} - m$ columns of the weight of the first layer of $\mathrm{g}_{\mathrm{large}}^{(l')}$ to zero and then apply Lemma 6. $\qquad\square$

**Approximating Attention Patterns**   We will now show that the attention pattern can be approximated by pruning random Transformer layers.

**Lemma 7.** *For any $\delta, \epsilon \in (0, 1)$, for any $W \in \mathbb{R}^m, \|W\|_2 \le 1$, for two random matrix $W_1, W_2 \in \mathbb{R}^{m' \times m}$ where $m' = O(m \log(\frac{m}{\min\{\epsilon, \delta\}}))$, suppose $\boldsymbol{X} \in \mathbb{R}^{m \times N}$, then there exists nonstructural pruning of $W_1, W_2$, named $\tilde{W}_1, \tilde{W}_2$, such that*

$$\| \boldsymbol{X}^\top \tilde{W}_1^\top \tilde{W}_2 \boldsymbol{X} - \boldsymbol{X}^\top W \boldsymbol{X} \|_\infty \le \epsilon \|\boldsymbol{X}\|_{1,2}^2$$

*Here we adopt $\|\|_\infty$ in vector sense, meaning the entry with largest absolute value.*

*Proof.* Suppose without loss of generality, $\|\boldsymbol{X}\|_{:,i} \le 1$. According to Lemma 2, there exists nonstructural pruning of $W_1, W_2$, named $\tilde{W}_1, \tilde{W}_2$, such that for any $x \in \mathbb{R}^m, \|x\|_2 \le 1$,

$$\| \tilde{W}_1^\top \tilde{W}_2 x - W x \|_2 \le \epsilon.$$

This then suggests that,

$$\|y^\top (\tilde{W}_1^\top \tilde{W}_2 x - Wx)\|_2 \le \epsilon\|y\|_2 \le \epsilon.$$

This concludes the proof. $\qquad\square$

The next lemma shows how error propogates through the softmax operators.

**Lemma 8.** *For any dimension $d$, suppose $x, y \in \mathbb{R}^d$ satisfies $\|x - y\|_\infty \le \epsilon$, then it holds that,*

$$\sum_{i=1}^d \Big| \frac{\exp(x_i)}{\sum_{i=1}^n \exp(x_i)} - \frac{\exp(y_i)}{\sum_{i=1}^n \exp(y_i)} \Big| \le \exp(2\epsilon) - 1.$$

*Proof.* One can observe that,

$$\exp(-\epsilon)\exp(x_i) \le \exp(y_i) \le \exp(\epsilon)\exp(x_i)$$

This then suggests,

$$\frac{\exp(x_i)}{\sum_{i=1}^n \exp(x_i)}\exp(-2\epsilon) \le \frac{\exp(y_i)}{\sum_{i=1}^n \exp(y_i)} \le \exp(2\epsilon)\frac{\exp(x_i)}{\sum_{i=1}^n \exp(x_i)}.$$

Hence,

$$\sum_{i=1}^d \Big| \frac{\exp(x_i)}{\sum_{i=1}^n \exp(x_i)} - \frac{\exp(y_i)}{\sum_{i=1}^n \exp(y_i)} \Big| \le \max\{\exp(2\epsilon) - 1, 1 - \exp(-2\epsilon)\} = \exp(2\epsilon) - 1.$$

This concludes the proof. $\qquad\square$

**Approximating Attention Module**  We will need the following lemma showing there exists a pruning of the value matrix in $\mathcal{T}_{\text{large}}$ such that it has eigenvalues with magnitude $\Theta(1)$.

**Lemma 9.** *For a matrix $W \in \mathbb{R}^{m_{\text{large}} \times m_{\text{large}}}$, with probability at least $1 - \frac{\delta}{10L}$, there exists a pruning of $W$, named $W'$, such that all the nonzero entries is contained in a $d \times d$ submatrix of $W'$ that satisfies that (1) all its eigenvalues are within $(\frac{1}{2}, 1)$, (2) the index of row specifying the submatrix and the index of column specifying the submatrix are disjoint.*

*Proof.* As $w_{\text{large}} = \Omega(m \log(\frac{dL}{\delta}))$, hence we can split $W_{1:\lceil m_{\text{large}}/2 \rceil, \lceil m_{\text{large}}/2 \rceil + 1:m_{\text{large}}}$ into $(m \times (m$ blocks, each with width at least $O(\log(\frac{m}{\delta}))$ [21]. Within each block, with probability $1 - \frac{\delta}{10Lm_{\text{large}}}$, there exists at least one entry that has value at least $\frac{1}{2}$. We can then choose $d$ disjoint entries in $W$ that are all at least $\frac{1}{2}$, indexed with $\{(a_i, b_i)\}_{i \in [d]}$ where $a_i < a_j$ and $b_i < b_j$ for $i < j$. We can then prune all other entries to zero. Consider the submatrix defined by entries $(a, b)$ for $a \in \{a_i\}_{i \in m}$ and $b \in \{b_i\}_{i \in m}$. Then, this submatrix will be diagonal and contains eigenvalues within $(\frac{1}{2}, 1)$. Further $\{a_i\}_{i \in m}$ and $\{b_i\}_{i \in m}$ must be disjoint because $a_i \le \lceil m_{\text{large}}/2 \rceil < b_i$. The proof is then completed. $\qquad\square$

We will also prove that LayerNorm with nonzero normalization constant is Lipschitz.

**Lemma 10.** *For LayerNorm function defined as* $\text{LN}(x) = \frac{\mathcal{P}_\perp x}{\max\{\|\mathcal{P}_\perp x\|_2, C\}}, x \in \mathbb{R}^m$, *for any* $x, y \in \mathbb{R}^m$, *it holds that,*

$$\Big\| \text{LN}(x) - \text{LN}(y) \Big\|_2 \le 2\|x - y\|_2/C.$$

*Proof.* We will proceed by a case analysis:

1. If $\|\mathcal{P}_\perp x\|_2, \|\mathcal{P}_\perp y\|_2 \le C$, then $\Big\|\text{LN}(x) - \text{LN}(y)\Big\|_2 = \frac{\|\mathcal{P}_\perp x - \mathcal{P}_\perp y\|_2}{C} \le \frac{1}{C}\|x - y\|_2$.

2. If $\|\mathcal{P}_\perp x\|_2, \|\mathcal{P}_\perp y\|_2 > C$, then $\Big\|\text{LN}(x) - \text{LN}(y)\Big\|_2 = \frac{\|\mathcal{P}_\perp x - \mathcal{P}_\perp y\|_2}{\|\mathcal{P}_\perp y\|_2} + \Big|1 - \frac{\|\mathcal{P}_\perp x\|_2}{\|\mathcal{P}_\perp y\|_2}\Big| \le \frac{2}{C}\|x - y\|_2$.

---

[21] $O(\cdot)$ hides absolute constants arising from the change of basis in the logarithm.

3. If $\|\mathcal{P}_\perp x\|_2 < C$ and $\|\mathcal{P}_\perp y\|_2 > C$, then $\left\|\mathrm{LN}(x) - \mathrm{LN}(y)\right\|_2 = \frac{\|\mathcal{P}_\perp x - \mathcal{P}_\perp y\|_2}{\|\mathcal{P}_\perp y\|_2} + \left|\frac{\|\mathcal{P}_\perp x\|_2}{C} - \frac{\|\mathcal{P}_\perp x\|_2}{\|\mathcal{P}_\perp y\|_2}\right| \le \frac{2}{C}\|x - y\|_2$.

The cases exhaust all possibilities, thus the proof is completed. $\qquad\square$

Finally, we will need a lemma showing how error accumulates when we consider both attention patterns and the value matrices.

**Lemma 11.** *For any dimension $d$ and positive number $N$, for $P, Q \in \mathbb{R}^{d \times d}$ satisfying that $\|P\|_2, \|Q\|_2 \le 1$, for any $x \in \mathbb{R}^{d \times N}$, if matrix $A \in \mathbb{R}^{N \times N}, B \in \mathbb{R}^{d \times N}$ satisfy that,*

$$\|A - \sigma(x^\top Q x)\|_{1,1} \le \epsilon_1.$$
$$\|B - Px\|_{1,2} \le \epsilon_2.$$
$$\forall i, k \in [N] \sum_{j \in [N]} A_{j,i} = 1, A_{k,i} \ge 0.$$

*Then it holds that,*

$$\|BA - Px\sigma(x^\top Q x)\|_{1,2} \le (\epsilon_1 \|PX\|_{1,2} + \epsilon_2).$$
$$\|\mathrm{LN}_C(BA) - \mathrm{LN}_C(Px\sigma(x^\top Q x))\|_{1,2} \le 2(\epsilon_1 \|PX\|_{1,2} + \epsilon_2)/C.$$

*Proof.* For any $i \in N$, we will have

$$\left\|(BA)_{:,i} - \left(Px\sigma(x^\top Q x)\right)_{:,i}\right\|_2$$
$$= \left\|\sum_{j \in [N]} A_{j,i} B_{:,j} - \left(\sigma(x^\top Q x)\right)_{j,i} (PX)_{:,j}\right\|_2$$
$$\le \left\|\sum_{j \in [N]} A_{j,i}(PX)_{:,j} - \left(\sigma(x^\top Q x)\right)_{j,i} (PX)_{:,j}\right\|_2 + \left\|\sum_{j \in [N]} A_{j,i} (PX - B)_{:,j}\right\|_2$$
$$\le \|PX\|_{1,2} \sum_{j \in [N]} |A_{j,i} - \left(\sigma(x^\top Q x)\right)_{j,i} + \|PX - B\|_{1,2}$$
$$\le \|PX\|_{1,2}\|A - \sigma(x^\top Q x)\|_{1,1} + \|PX - B\|_{1,2} \le \epsilon_1 \|PX\|_{1,2} + \epsilon_2.$$

The rest follows from Lemma 10

$\qquad\square$

A LayerNorm of larger dimension can be made to be functionally equivalent to a LayerNorm of a smaller dimension. Precisely:

**Lemma 12.** *Given any dimension $d < d'$, it holds that for any $x \in \mathbb{R}^d$,*

$$\mathrm{LN}_C\left(\begin{bmatrix} \mathcal{P}_\perp x \\ 0^{d'-d} \end{bmatrix}\right) = \begin{bmatrix} \mathrm{LN}_C(x) \\ 0 \end{bmatrix}.$$

*Proof.* The proof follows directly from definition. $\qquad\square$

We will now formally define attention module.

**Definition 7** (Attention Module). *We will define attention module $a(\boldsymbol{X} \mid W_V, W_K, W_Q)$ as*

$$a(\boldsymbol{X}) = \mathrm{LN}_C\left(W_V \boldsymbol{X} \sigma(\boldsymbol{X}^\top W_K^\top W_Q \boldsymbol{X})\right).$$

**Lemma 13.** *Attention module is lipschitz with respect to $1, 2$-norm for bounded input. Precisely, consider attention module (Definition 7) parameterized by $\|W_V\|_2, \|W_K\|_2, \|W_Q\|_2 \le 1$ with input domain $\|\boldsymbol{X}\|_{1,2} \le 4L$, $a(\boldsymbol{X})$ is $200L^2/C$-lipschitz with respect to $1, 2-$norm.*

*Proof.* We have that

$$a(\boldsymbol{X}) = \mathrm{LN}_C\left(W_V \boldsymbol{X} \sigma(\boldsymbol{X}^\top W_K^\top W_Q \boldsymbol{X})\right).$$

Choose $\epsilon$ to be a sufficiently small constant, such that, $\exp(32L\epsilon) - 1 \le 64L\epsilon$. Consider $\boldsymbol{X}$ and $\tilde{\boldsymbol{X}}$ satisfying that $\|\boldsymbol{X} - \tilde{\boldsymbol{X}}\|_{1,2} \le \epsilon$ and $\|\boldsymbol{X}\|_{1,2} \le 4L, \|\tilde{\boldsymbol{X}}\|_{1,2} \le 4L$, we will have

$$\left| \left( \boldsymbol{X}^\top W_K^\top W_Q \boldsymbol{X} - (\tilde{\boldsymbol{X}})^\top W_K^\top W_Q(\tilde{\boldsymbol{X}}) \right)_{i,j} \right|$$

$$= \left| (\boldsymbol{X}_{:,i} - \tilde{\boldsymbol{X}}_{:,i})^\top W_K^\top W_Q \boldsymbol{X}_{:,j} + (\tilde{\boldsymbol{X}}_{:,i})^\top W_K^\top W_Q(\boldsymbol{X}_{:,j} - \tilde{\boldsymbol{X}}_{:,j}) + (\boldsymbol{X}_{:,i} - \tilde{\boldsymbol{X}}_{:,i})^\top W_K^\top W_Q(\boldsymbol{X}_{:,j} - \tilde{\boldsymbol{X}}_{:,j}) \right|$$

$$\le 8L\epsilon + \epsilon^2 \le 16L\epsilon.$$

By Lemma 8, this implies,

$$\|\sigma(\boldsymbol{X}^\top W_K^\top W_Q \boldsymbol{X}) - \sigma((\tilde{\boldsymbol{X}})^\top W_K^\top W_Q(\tilde{\boldsymbol{X}}))\|_{1,1} \le \exp(32L\epsilon) - 1 \le 64L\epsilon.$$

We also have

$$\|W_V\left(\boldsymbol{X} - \tilde{\boldsymbol{X}}\right)\|_{1,2} \le \epsilon.$$

$$\|W_V \boldsymbol{X}\|_{1,2} \le 4L$$

Lemma 11 then implies that

$$\|a(\boldsymbol{X}) - a(\tilde{\boldsymbol{X}})\|_{1,2} \le 200L^2\epsilon/C.$$

This then concludes the proof. $\qquad\square$

We can now prove that a large Transformer Layer and an attention module of the larger Transformer can be pruned to approximate the attention module of a smaller Transformer Layer module.

**Lemma 14.** *Under the setting of Theorem 2, with probability $1 - \delta/2$, for any $l \in [L], l' \in [4L - 1]$, let $a^{(l)}$ be the attention module on the $l$-th layer of $\mathcal{T}$, there exists a pruning of the $(l' - 1)$-th layer $\mathcal{T}_{large}^{(l'-1)}$, named $\tilde{\mathcal{T}}_{large}^{(l'-1)}$ and the attention module on $l'$-th layer $a_{large}^{l'}$ named $\tilde{a_{large}^{l'}}$, such that when defined on domain $\|\boldsymbol{X}\|_{1,2} \le 2L$,*

1. *$\tilde{\mathcal{T}}_{large}^{(l'-1)}$ is independent of the last $m_{\mathrm{large}} - m$ rows of the input.*

2. *$\left( \tilde{a_{large}^{l'}} \circ \tilde{\mathcal{T}}_{large}^{(l'-1)} \left( \begin{bmatrix} x \\ 0^{(m_{\mathrm{large}}-m) \times N} \end{bmatrix} \right) \right)_{1:m}$ is an $\left( \frac{C}{1000L^2} \right)^{4L-1} \epsilon$-approximation of $a^{(l)}(x)$ with respect to $1, 2$-norm.*

3. *$\left( \tilde{\mathcal{T}}_{large}^{(l'-1)} \left( \begin{bmatrix} x \\ 0^{(m_{\mathrm{large}}-m) \times N} \end{bmatrix} \right) \right)_{1:m}$ is an $\left( \frac{C}{1000L^2} \right)^{4L} \epsilon$-approximation of $\boldsymbol{X}$ with respect to $1, 2$-norm.*

*Proof.* We will use the shorthand $\epsilon_0 = \left( \frac{C}{1000L^2} \right)^{4L} \epsilon$ and prune in the following order. It holds that for $\epsilon \le 1, \exp(8L^2\epsilon_0) - 1 \le 16L^2\epsilon_0$.

1. We will prune $W_V^{\mathrm{large},(l')}$ according to Lemma 9 and name the pruned matrix $W_V^{\tilde{\mathrm{large}},(l')}$. By Lemma 9, all the nonzero entries is contained in a $d \times d$ submatrix of $W'$ that satisfies that all its eigenvalues are within $(\frac{1}{2}, 1)$. We will assume WLOG the submatrix is the one specified by row $1 \ldots d$ and column $d + 1 \ldots 2d$ and name the submatrix as $W$.

2. We will then prune $\mathcal{T}_{\mathrm{large}}^{(l'-1)}$ according to Lemma 4 to output $\epsilon_0$-approximation of $\boldsymbol{X} \in \mathbb{R}^{m \times N} \rightarrow$
$\begin{bmatrix} \boldsymbol{X} \\ W^{-1}\mathcal{P}_\perp W_V^{(l)}\boldsymbol{X} \\ 0^{(m_{\mathrm{large}}-2m) \times N} \end{bmatrix}$. As $W$ is defined as the submatrix pruned by $W_V^{(t+1)}$, it holds that

$$W_V^{\tilde{\mathrm{large}},(l')} \begin{bmatrix} \boldsymbol{X} \\ W^{-1}\mathcal{P}_\perp W_V^{(l)}\boldsymbol{X} \\ 0^{(m_{\mathrm{large}}-m) \times N} \end{bmatrix} = \begin{bmatrix} \mathcal{P}_\perp W_V^{(l)}\boldsymbol{X} \\ 0^{(m_{\mathrm{large}}-m) \times N} \end{bmatrix}.$$

3. Finally we will prune $W_K^{\text{large},(l')}, W_Q^{\text{large},(l')}$ according to Lemma 7 to approximate $(W_K^{(l)})^\top W_Q^{(l)}$ up to $\epsilon_0$ error.

we can now calculate the approximation error. For any $\boldsymbol{X} \in \mathbb{R}^{m \times N}, \|\boldsymbol{X}\|_{1,2} \leq 2L$, suppose

$$\mathcal{T}^{(\tilde{l'}-1)}(\boldsymbol{X}) = \begin{bmatrix} \boldsymbol{X} + \delta_1 \\ W^{-1}\mathcal{P}_\perp W_V^{(l)}\boldsymbol{X} + \delta_2 \\ 0^{(m_{\text{large}}-2m)\times N} \end{bmatrix}$$

Then by our constrution, it holds that $\forall i \in \{1, 2\}, \|\delta_i\|_{1,2} \leq \epsilon_0 \|\boldsymbol{X}\|_{1,2}$.

We would then have

$$W_V^{\tilde{\text{large}},(l')}\mathcal{T}^{(\tilde{l'}-1)}(\boldsymbol{X}) = \begin{bmatrix} \mathcal{P}_\perp W_V^{(l)}\boldsymbol{X} + W_V^{\tilde{\text{large}},(l')}\delta_2 \\ 0^{(m_{\text{large}}-m)\times N} \end{bmatrix} \tag{48}$$

By our construction, it holds that $\|W_V^{\tilde{\text{large}},(l')}\delta_2\|_{1,2} \leq 2\|\delta_2\|_{1,2} \leq 2\epsilon_0\|\boldsymbol{X}\|_{1,2}$.

Further, by the construction of $W_K^{\tilde{\text{large}},(l')}, W_Q^{\tilde{\text{large}},(l')}$, it holds that,

$$\left\| \left( W_K^{\tilde{\text{large}},(l')}\mathcal{T}^{(\tilde{l'}-1)}(\boldsymbol{X}) \right)^\top \left( W_Q^{\tilde{\text{large}},(l')}\mathcal{T}^{(\tilde{l'}-1)}(\boldsymbol{X}) \right) \right.$$
$$\left. - (W_K^{(l)}\boldsymbol{X} + W_K^{(l)}\delta_1)^\top(W_Q^{(l)}\boldsymbol{X} + W_Q^{(l)}\delta_1) \right\|_\infty \leq \epsilon_0 \tag{49}$$

As for any $i, j \in [N]$

$$\left| \left( (W_K^{(l)}\boldsymbol{X} + W_K^{(l)}\delta_1)^\top(W_Q^{(l)}\boldsymbol{X} + W_Q^{(l)}\delta_1) - (W_K^{(l)}\boldsymbol{X})^\top W_Q^{(l)}\boldsymbol{X} \right)_{i,j} \right|$$
$$\leq \left| (W_K^{(l)}\boldsymbol{X}_{:,i})^\top(W_Q^{(l)}\delta_1)_{:,j} \right| + \left| (W_K^{(l)}\delta_1)_{:,i}^\top(W_Q^{(l)}\boldsymbol{X})_{:,j} \right| + \left| (W_K^{(l)}\delta_1)_{:,i}^\top(W_Q^{(l)}\delta_1)_{:,j} \right|$$
$$\leq \|\boldsymbol{X}\|_{1,2}^2(2\epsilon_0 + \epsilon^2) \leq 4\|\boldsymbol{X}\|_{1,2}^2\epsilon_0.$$

combined with Equation (49),

$$\left\| \left( W_K^{\tilde{\text{large}},(l')}\mathcal{T}^{(\tilde{l'}-1)}(\boldsymbol{X}) \right)^\top \left( W_Q^{\tilde{\text{large}},(l')}\mathcal{T}^{(\tilde{l'}-1)}(\boldsymbol{X}) \right) - (W_K^{(l)}\boldsymbol{X})^\top W_Q^{(l)}\boldsymbol{X} \right\|_\infty \leq \epsilon_0(1 + 4\|\boldsymbol{X}\|_{1,2}^2). \tag{50}$$

By Lemma 8, this implies

$$\left\| \sigma\left( \left( W_K^{\tilde{\text{large}},(l')}\mathcal{T}^{(\tilde{l'}-1)}(\boldsymbol{X}) \right)^\top \left( W_Q^{\tilde{\text{large}},(l')}\mathcal{T}^{(\tilde{l'}-1)}(\boldsymbol{X}) \right) \right) \right.$$
$$\left. - \sigma\left( (W_K^{(l)}\boldsymbol{X})^\top W_Q^{(l)}\boldsymbol{X} \right) \right\|_{1,1} \leq 4\epsilon_0(1 + 4\|\boldsymbol{X}\|_{1,2}^2). \tag{51}$$

By Lemma 11, Equations (48) and (51) imply,

$$\left\| W_V^{\tilde{\text{large}},(l')}\mathcal{T}^{(\tilde{l'}-1)}(\boldsymbol{X})\sigma\left( \left( W_K^{\tilde{\text{large}},(l')}\mathcal{T}^{(\tilde{l'}-1)}(\boldsymbol{X}) \right)^\top \left( W_Q^{\tilde{\text{large}},(l')}\mathcal{T}^{(\tilde{l'}-1)}(\boldsymbol{X}) \right) \right) \right.$$
$$\left. - \begin{bmatrix} \mathcal{P}_\perp W_V^{(l)}\boldsymbol{X}\sigma\left( (W_K^{(l)}\boldsymbol{X})^\top W_Q^{(l)}\boldsymbol{X} \right) \\ 0^{(m_{\text{large}}-m)\times N} \end{bmatrix} \right\|_{1,2} \leq 8\epsilon_0(1 + 4\|\boldsymbol{X}\|_{1,2}^2)\|\boldsymbol{X}\|_{1,2} \leq 80L^2\epsilon_0.$$

Now according to Lemmas 10 and 12, it holds that

$$\left\| a_{\text{large}}^{\tilde{l'}} \circ \tilde{\mathcal{T}}_{\text{large}}^{(l'-1)}\left( \begin{bmatrix} x \\ 0^{(m_{\text{large}}-m)\times N} \end{bmatrix} \right)_{1:m} - a^{(l)}(x) \right\|_{1,2} \leq 160L^2\epsilon_0/C.$$

This concludes the proof. $\qquad \square$

**Approximating Transformer Layers**  We will finally show that two random Transformer layers can be pruned to approximate a given Transformer layer.

**Lemma 3.** *Under the setting of Theorem 2, with probability $1-2\delta/3$, for any $l \in [L], l' \in [4L-1]$, let $\mathcal{T}^{(l)}$ be the $l$-th layer of $\mathcal{T}$, there exists a pruning of the $(l'-1)$-th and the $(l')$-th layer $\mathcal{T}_{large}^{(l'-1)}, \mathcal{T}_{large}^{l'}$, named $\tilde{\mathcal{T}}_{large}^{(l'-1)}, \tilde{\mathcal{T}}_{large}^{l'}$ such that when defined on domain $\|X\|_{1,2} \le 2L, X \in \mathbb{R}^{m \times N}$,*

1. *$\tilde{\mathcal{T}}_{large}^{(l'-1)}$ is independent of the last $m_{\mathrm{large}} - m$ rows of the input.*

2. *$\tilde{\mathcal{T}}_{large}^{l'} \circ \tilde{\mathcal{T}}_{large}^{(l'-1)} \left(\bar{X}\right)$ is an $\left(\frac{C}{1000L^2}\right)^{4L-3} \epsilon$-approximation of $\overline{\mathcal{T}^{(l)}(X)}$ with respect to $1, 2$-norm.*

*Proof.* We will prune the $(l'-1)$-th layer and the attention module of the $l'$-th layer according to Lemma 14 to approximate $a^{(l)}$ and the projection function of the $l'$-th layer according to Corollary 1. Notice that $\left\|a^{(l)}(X) + X\right\|_{1,2} \le (\frac{2}{C}+1)\|X\|_{1,2}$ and $g^{(l)}$ is $1-$lipschitz, according to Lemma 5,

$$\left(\tilde{\mathcal{T}}_{\mathrm{large}}^{l'} \odot \tilde{\mathcal{T}}_{\mathrm{large}}^{(l'-1)} \left(\begin{bmatrix} x \\ 0^{(m_{\mathrm{large}}-m) \times N} \end{bmatrix}\right)\right)_{1:m} \quad \text{is an } \epsilon'\text{-approximation of } \mathcal{T}^{(l)}(x), \text{ with}$$

$$\epsilon' \le (\frac{2}{C}+1)\left(\frac{C}{1000L^2}\right)^{4L}\epsilon + \left(\frac{C}{1000L^2}\right)^{4L-2}\epsilon + \left(\frac{C}{1000L^2}\right)^{8L-2}\epsilon^2 \le \left(\frac{C}{1000L^2}\right)^{4L-3}\epsilon.$$

This concludes the proof. $\qquad\square$

## C.6  Technical Lemmas

**Lemma 15.** *Given any dimension $d$ and number of samples $n$, for any size-$n$ dataset $\{(x_i, y_i)\}_{i \in [n]}$ with $x_i \in \mathbb{R}^d$ and $y_i \in \mathbb{R}$, there exists a width-$2n$ two-layer MLP $f : \mathbb{R}^d \to \mathbb{R}$ with ReLU activation such that, $f(x_i) = y_i$ for any $i \in [n]$.*

*Proof.* We will first choose direction $w \in \mathbb{R}^d, \|w\|_2 = 1$ and margin $\gamma > 0$ such that for any $i \ne j$ in $[n]$, it holds that,

$$\left|\langle w, x_i - x_j \rangle\right| \ge 2\gamma.$$

We will assume WLOG $w^\top x_i$ is increasing in $i$.

Then we will construct an auxilliary series $z_i$ for $i \in [n]$ such that,

$$z_1 = y_1/\gamma$$

$$z_i = y_i/\gamma - 2\sum_{j=1}^{i-1} z_j, i \in \{2, \dots n\}.$$

Finally consider the following two-layer MLP with ReLU activation,

$$f(x) = \sum_{i=1}^{n} z_i \mathrm{ReLU}\left(\langle w, x - x_i \rangle + \gamma\right) - z_i \mathrm{ReLU}\left(\langle w, x - x_i \rangle - \gamma\right),$$

we will show that $f(x_i) = y_i$ for any $i \in [n]$. Notice that

$$z_j \mathrm{ReLU}\left(\langle w, x_i - x_j \rangle + \gamma\right) - z_j \mathrm{ReLU}\left(\langle w, x_i - x_j \rangle - \gamma\right) = \begin{cases} 0, & j > i, \\ \gamma z_i, & j = i, \\ 2\gamma z_j, & j < i. \end{cases}$$

Thus it holds,

$$f(x_i) = \sum_{j=1}^{n} z_j \mathrm{ReLU}\left(\langle w, x_i - x_j \rangle + \gamma\right) - z_j \mathrm{ReLU}\left(\langle w, x_i - x_j \rangle - \gamma\right)$$

$$= \sum_{j=1}^{i-1} 2\gamma z_j + \gamma z_i = y_i.$$

$$\square$$

**Lemma 16.** *Given any sets $\{x_i\}_{i \in m}$ satisfying that $x_i \in \mathbb{R}^n$ and $x_i \neq 0$, there exists a set of orthonormal vectors $\{u_j\}_{j \in [n-2]}$ of $\mathbb{R}^n$ such that (1) $u_j^\top 1^n = 0$ for any $j \in [n-2]$ and (2) $\sum_{j \in [n-2]} u_j^\top x_i u_j \neq x_i$ for any $i \in [m]$.*

*Proof.* There exists a vector $v \in \mathbb{R}^n$ such that $v^\top x_i \neq 0$ for any $i \in [m]$. We can then construct an orthonormal basis $\{u_j\}_{j \in [n-2]}$ of $\mathbb{R}^n$ as the basis of the normal space of $\text{span}(v, 1^n)$. Then the lemma holds. $\qquad\square$

**Lemma 17.** *Given any dimension $n$ and constant $M$, there exists a 2-layer width-$2n$ ReLU network $f : \mathbb{R}^{n+1} \to \mathbb{R}$ such that for any $x \in [0, M]^n, y \in [n]$, $f(x \oplus y) = x_y$.*

*Proof.* The construction is as followed, we will choose $f$ as

$$f(x \oplus y) = \sum_{i=1}^n \text{ReLU}(x_i + M(y - i)) - \sum_{i=1}^n \text{ReLU}(x_i + M(y - i - 1)) - M(y - 1).$$

Then as we have

$$\text{ReLU}(x_i + M(y - i)) - \sum_{i=1}^n \text{ReLU}(x_i + M(y - i - 1)) = \begin{cases} M, & i \leq y - 1; \\ x_i, & i = y; \\ 0, & i \geq y + 1. \end{cases}$$

The proof is completed. $\qquad\square$

**Lemma 18.** *Given any dimension $n$ and constant $M > 0$, there exists a 2-layer width-$2n$ ReLU network $f : \mathbb{R}^n \to \mathbb{R}$ such that for any $x \in \mathbb{R}^n$ satisfying there exists $i \in [n]$, $x_i > M$ and $\forall j \neq i, x_j = 0$, it holds that $f(x) = i$.*

*Proof.* The construction is as followed, we will choose $f$ as

$$f(x) = \sum_{i=1}^m i \left( \text{ReLU}(x_i) - \text{ReLU}(x_i - M) + M \right) / M.$$

The proof is completed. $\qquad\square$

**Lemma 19.** *Given any dimension $n$ and natural numbers $K, m, M$, if there exists $K$ different 2-layer width-$m$ ReLU networks $f_k : \mathbb{R}^n \to \mathbb{R}$, then there exists a 2-layer width-$2Km$ ReLU network $f : \mathbb{R}^{n+1} \to \mathbb{R}$, such that $f(\begin{bmatrix} k \\ x \end{bmatrix}) = f_k(x)$ when $x \in [0, M]^n$.*

*Proof.* Suppose that

$$f_k(x) = \sum_{i=1}^m a_{k,i} \text{ReLU}(w_{k,i}^\top x + b_{k,i}) + b_k.$$

Then we can construct

$$f(\begin{bmatrix} y \\ x \end{bmatrix}) = \sum_{k=1}^K \sum_{i=1}^m a_{k,i} \text{ReLU}(w_{k,i}^\top x + b_{k,i} + M(y - k)) - a_{k,i} \text{ReLU}(w_{k,i}^\top x + b_{k,i} + M(y - k - 1))$$
$$+ b_k - c_{k,i} \text{ReLU}(y + 1 - k),$$

where $c_{k,i}$ satisfies

$$\forall i, k', \sum_{k=1}^{k'} c_{k,i}(k' + 1 - k) = M \sum_{k=1}^{k'-1} a_{k,i}.$$

The proof is then completed. $\qquad\square$

**Lemma 20.** *Given any dimension $n$ and $W \in \mathbb{R}^{n \times n}$, $\|W\|_2 \leq 2$, there exists a 2-layer width-$2n$ ReLU network $f : \mathbb{R}^n \to \mathbb{R}$ such that for any $x \in \mathbb{R}^n$, it holds that $f(x) = Wx$ and both weight matrices parameterizing $f$ has spectral norm less than $2\sqrt{2}$.*

*Proof.* The construction is straightforward, one can choose

$$f(x) = [I_n, -I_n]^\top \text{ReLU}\left(\begin{bmatrix} Wx \\ -Wx \end{bmatrix}\right).$$

$\square$

## C.7 Discussion on Architecture Choices

The reader may notice that Equation (6) is not the same as the standard GPT architecture,

$$f_l(X; \theta^{(l)}) = g^{(l)}\left(\text{LN}\left(W_V^{(l)} X \sigma\left(\mathcal{C} + (W_K^{(l)} X)^\top (W_Q^{(l)} X)\right) + X\right)\right). \tag{52}$$

We will shortly discuss the impact of considering Equation (52) here.

With similar arguments to Theorem 3 and the necessity part of Theorem 1, one can prove that similar balance conditions should also hold for a transformer with a layer specified by Equation (52) and a minimal first layer that can nearly perfectly generate bounded Dyck languages.

However, the sufficiency part of Theorem 1 no longer holds, when the balance condition holds, the last column of the term $W_V^{(2)} X \sigma\left(\mathcal{C} + (W_K^{(2)} X)^\top (W_Q^{(2)} X)\right)$ will converge to zero when input length converges to infinity. Hence, if not all $e(\tau_{t,d})$ where $\tau_{t,d}$ is a closed bracket aligns with $1^m$, then it is impossible for the model to perfectly generate Dyck for arbitrary length. Although it remains possible to refine a sharper condition for standard GPT architecture to perfectly generate Dyck Language, we find considering Equation (6) more elegant in theory. We also verify with experiments that our architecture with standard training can learn bounded Dyck language to more than $97\%$ accuracy. Also, the learned attention patterns are also similarly not interpretable as standard architectures.

# D Experiments

## D.1 Training Details

For Figure 1, we train 2-layer standard GPT on $\mathrm{Dyck}_{2,4}$ with sequence length no longer than 28. For $(a)$, we train with hidden dimension and network width 200 and learning rate 3e-4. For $(b), (c), (d)$, we train with hidden dimension and FFN width 50 and learning rate 3e-3.

For Figure 2, for $(a)$, we train 1-layer transformer without residual link, FFN and the final LayerNorm before the linear head. The hidden dimensions and FFN widths are fixed as 500. For $(a)$, we train the network with learning rate 1e-2 and for $(b), (c), (d)$ we train the network with learning rate 3e-3.

## D.2 Additional Results on Dyck Prefix

In the experiment presented in the main text, we perform experiments on complete Dyck sequences, which is a special case of Dyck prefixes. In this section, we present additional experiments on Dyck prefixes $\mathrm{Dyck}_{2,4,28}$.

**Attention Patterns** We first perform experiments on attention patterns. The qualitative results are shown in Figures 7 and 9. We can observe that the attention patterns are still diverse and do not commonly show stack-like patterns. We also calculate the *attention variation* [22], and find that the attention variation is $0.34$, based on 30 models with a minimal first layer and different random seeds. In contrast, for models with a standard first layer and without position encodings, the attention variation is surprisingly high, reaching $14.51$. The high value is caused by the large distance between attention patterns like Figure 7 (c) and (d); that is, between patterns that attend more to the current positions, and patterns that attend more heavily to the initial position. The difference is even increased when we consider longer sequence (Figure 8). Similarly, the variation is also high for models with linear position embedding, reaching $11.92$. This shows that the attention patterns are still diverse and do not commonly show stack-like patterns.

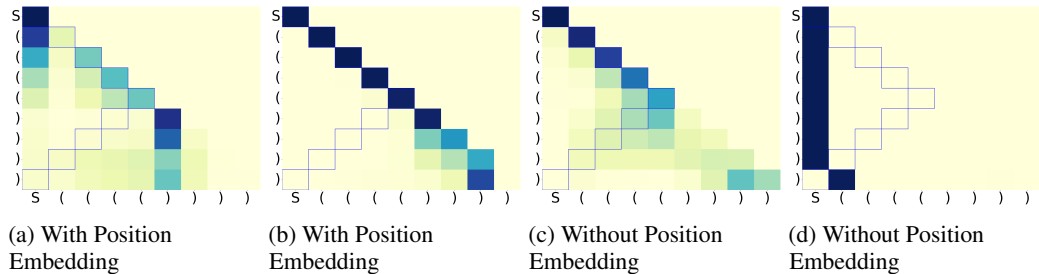

| (a) With Position Embedding | (b) With Position Embedding | (c) Without Position Embedding | (d) Without Position Embedding |

Figure 7: **Second-layer attention patterns of two-layer Transformers on Dyck Prefix**: Models for (a),(b) are under the same setup but different random seeds; similarly for (c),(d). All models reach $\geq 97\%$ accuracy (defined in Section 4.1). In the heatmap, darker color indicates larger value. As we can observe, the attention patterns still show much variance.

**Balanced Violations** We also test the relationship with the balance violation with length generalization on Dyck prefixes, similar to Figure 3. We observe that although the negative correlation is not presented as in the case of Dyck sequences, contrastive regularization still helps reduce the balance violation and significantly improve the length generalization performance. This shows that for Dyck prefixes, while the balance violation may not be predictive of the length generalization performance, it is still possible to reduce the balance violation and improve the length generalization performance. The results are shown in Figure 10.

---

[22]Recall from Section 4.1 that the attention variation between two attention patterns $A_1, A_2 \in \mathbb{R}^{N \times N}$ is defined as $\mathrm{Variation}(A_1, A_2) = \|A_1 - A_2\|_F^2$.

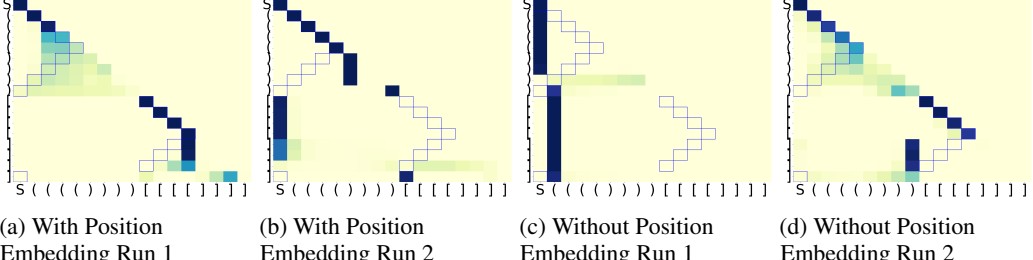

(a) With Position
Embedding Run 1

(b) With Position
Embedding Run 2

(c) Without Position
Embedding Run 1

(d) Without Position
Embedding Run 2

Figure 8: **Second-layer attention patterns of two-layer Transformers on Longer Dyck Prefix**: Models for (a),(b) are under the same setup but different random seeds. All models reach $\geq 97\%$ accuracy (defined in Section 4.1). In the heatmap, darker color indicates larger value.

### D.3 Extended Experiments

We include more experiments on the attention variation of different Dyck languages and architectures. The results are summarized in Table 1.

| #types $k$ | Grammar depth $m$ | #Layers $l$ | Layer 1 | Layer 2 | Layer 3 |
|---|---|---|---|---|---|
| 2 | 4 | 2 | $0.047_{(0.006)}$ | $7.721_{(0.908)}$ | |
| 2 | 4 | 3 | $0.070_{(0.013)}$ | $5.072_{(0.645)}$ | $24.063_{(1.166)}$ |
| 2 | 8 | 2 | $0.087_{(0.012)}$ | $7.583_{(0.961)}$ | |
| 2 | 8 | 3 | $0.059_{(0.011)}$ | $5.560_{(0.714)}$ | $23.590_{(0.829)}$ |
| 3 | 4 | 2 | $0.182_{(0.024)}$ | $9.313_{(0.815)}$ | |
| 3 | 4 | 3 | $0.225_{(0.032)}$ | $8.426_{(0.877)}$ | $25.749_{(0.897)}$ |
| 3 | 8 | 2 | $0.178_{(0.028)}$ | $7.000_{(0.884)}$ | |
| 3 | 8 | 3 | $0.154_{(0.036)}$ | $6.280_{(0.711)}$ | $25.451_{(0.871)}$ |

Table 1: **Extended attention variation.** "Layer $i$" shows the mean (and standard deviation) of the attention variation on layer $i$, calculated on $40$ sentences. The embedding width and FFN width are fixed as $50$ in the experiments. We train using sentences from $\mathsf{Dyck}_{k,m}$ of length less than $28$ and test the variation on $40$ randomly sampled sentences with length $19$ (the sampled sentence is fixed across different architectures). The random attention variation baseline here is $3.33$. The numbers in this table are different from previous discussion, since the results here are from a slightly different architecture than the standard GPT-2 architecture: a residue link is appended after the LayerNorm to match our theory better. The models are trained to convergence and have in-distribution accuracy higher than $97\%$.

**Attention pattern visualization for three-layer experiments.** The first-layer attention is close to uniform, while the higher-layer attention shows no clear patterns.

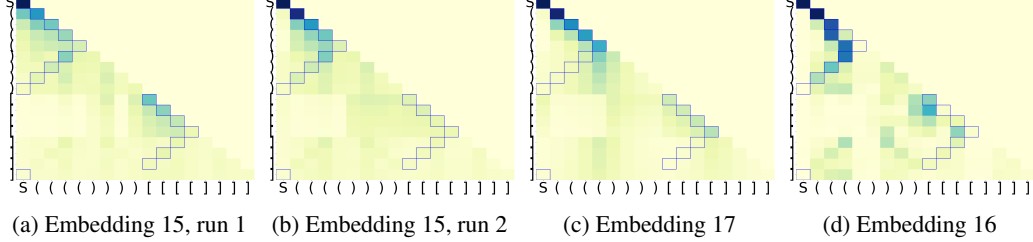

(a) Embedding 15, run 1

(b) Embedding 15, run 2

(c) Embedding 17

(d) Embedding 16

Figure 9: **Second-layer attention patterns of two-layer Transformers with a minimal first layer**: (a), (b) are based on embedding 15 with different random seeds. (c), (d) are based on embedding 17 and 16. Different embedding functions lead to diverse attention patterns, most of which are not stack-like.

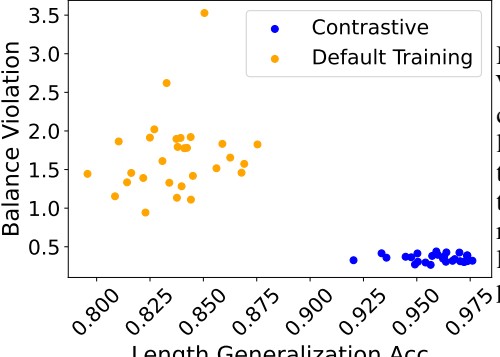

Figure 10: **Relationship Between Balance Violation and Length Generalization.** Accuracy from Transformers with minimal first layer with embedding 15, using both standard training and contrastive regularization (Equation (18)). We again observe that contrastive regularization helps reduce the balance violation and improve the length generalization performance.



(a) $seed = 0$      (b) $seed = 1$      (c) $seed = 2$      (d) $seed = 3$

Figure 11: **Third-layer attention.** The test sentence is fixed and the attention patterns learned by different 3-layer models with the same architectures on the same dataset show large variation visually.

