# OpenReview forum: "Transformers are uninterpretable with myopic methods: a case study with bounded Dyck grammars"
_NeurIPS.cc/2023/Conference — NeurIPS 2023 poster_

### Official Review · Reviewer_ZsLs · 2023-07-01

**Soundness:** 3 good
**Presentation:** 2 fair
**Contribution:** 4 excellent
**Rating:** 7
**Confidence:** 3

**Summary:**

This paper provides mathematical evidence that typical methods for interpreting the inner workings of transformers (looking at attention patterns, looking at feedforward weights) cannot be relied upon. The main argument of the paper is this: There is a task (specifically, language modeling of the bounded-depth Dyck-k language) such that it is possible for a two-layer transformer language model (assuming that it is trained on infinite data) to solve it perfectly using only uniform attention in the second layer (Corollary 1). The first layer can be one of multiple constructions proposed in prior work. Since uniform attention can hardly be considered "interpretable," this shows that a transformer can solve a task perfectly without an interpretable attention pattern, so we cannot reliably draw conclusions from attention patterns alone. Theorem 2 removes the assumption that the model is trained on infinite data. Theorem 3 shows (I think) that for any sufficiently large transformer, it's likely that there is a fully-connected layer in one of the feedforward sublayers such that, if you prune weights in the whole transformer two different ways, resulting in two transformers that have different behavior, that layer can be pruned exactly the same way in both networks, implying that that layer can be interpreted ambiguously. So, we can't rely on analyzing feedforward layers in isolation for interpretability. The authors give some examples of attention patterns in transformers trained on this task that don't have clear interpretations and run experiments validating some aspects of their theory. They also propose a regularization term that helps the transformer generalize to out-of-distribution examples on the Dyck language.

**Strengths:**

This paper, in a mathematically rigorous way, answers a very important question: How much can we rely on naive ways of attempting to interpret the inner workings of transformers, i.e. looking at attention weights or feedforward weights? It turns out the answer is: not much. Although it might have been a bit obvious beforehand that looking at only one component of a transformer, and not the interplay between different parts of the network, is not necessarily reliable for interpretability, the authors prove this fact rigorously, which is an extremely valuable contribution. I think the bounded Dyck language was an excellent choice of case study: not so easy that the solution is trivial, but easy enough that the analysis is manageable. I think the overall logical argumentation of the paper is solid.

A particularly cool result is that it's possible to train a transformer to solve the Dyck language while forcing the attention in the second layer to be uniform, which agrees with the theory. The experiments nicely back up the theoretical work.

The fact that Theorem 3 is a provable Lottery Ticket Hypothesis is also very interesting.

Originality: AFAIK this is an original and creative contribution.

Quality: The methodology and mathematical rigor in this paper seem to be sound.

Clarity: Overall good, but I think this could be improved in places.

Significance: High. It has important implications for anyone working on interpretability and transformers.

**Weaknesses:**

**Edit:** I have read the rebuttal, and it addressed my biggest concerns.

The only reasons I have not immediately given this paper a higher score are that (1) I have some important questions in the Questions section I would like answered (2) I think the clarity of the paper could be improved.

The clarity issues have to do with the fact that the mathematical definitions and notation can get a bit overwhelming, and I would appreciate any attempt by the authors to simplify it and make it more readable.

I think Section 3.2 requires a lot more clarification. I had a very difficult time understanding the logical argument, as there are several levels of nested quantifiers. I did not have time to check the proof in the appendix.

**Questions:**

Most important questions:

1. Eq 5: In this equation, layer norm comes before the residual connection. Isn't post-norm supposed to go after the residual? Is this a mistake?
1. Eq 5: I think there's a mistake with how the causal mask is included -- it should be added to the softmax inputs, not multiplied, and the entries should have values of 0 or -inf, not 1 or 0.

Important questions:

1. Eq 5: Will these results hold true if you use pre-norm instead of post-norm?
1. 162: It might be quite difficult for a transformer to eliminate positional encodings, because of the residual connections. Would this interfere with your construction?
1. Thm 3: Doesn't non-structural pruning have the ability to encode information in $T_{large}'$, merely with the pattern of which weights are pruned? Am I correct in understanding that the last part of Theorem 3 is meant to account for this? And that the significance of Theorem 3 is not that $T$ can approximate $T_{large}$, but that $T_{large}$ could be approximating one of multiple transformers $T_1$ and $T_2$, which have different behaviors? Won't $T_1$ and $T_2$ necessarily have similar behavior because they are similar to $T_{Large,1}$ and $T_{Large,2}$, and $T_{Large,1}$, and $T_{Large,2}$ are similar to $T_{Large}$? I think the answer is no because $T_{Large,1}$ and $T_{Large,2}$ are not necessarily similar to $T_{Large}$, but I wanted to clarify.
1. Sec 3.2: I think this whole section would benefit from a more concise, intuitive description of the logical argument inside Theorem 3, and how it implies the statement made on 247. I'm still quite confused as to how this argument works. If $T_{Large,1}$ and $T_{Large,2}$ coincide on $W$ after pruning, how do you know that the weights pruned in $W$ do not account for the difference in behavior between them? Doesn't this argument only apply to pruned transformers then?
1. 227: What is the main message of Corollary 2? Is it that Approximate Balance still holds true in the presence of l2 regularization, which is a common technique, but to a lesser degree for large $\lambda$? Is there a large enough $\lambda$ that it will fail to hold true?
1. Do Theorems 1-2 work for transformers with more than 2 layers?
1. 237: What is "width"?
1. 279: As mentioned before, don't positional encodings complicate the construction, possibly explaining why there's a change?
1. Fig 3: There are points that make it hard to believe there is a strong negative correlation. For example, some blue points have lower accuracy than orange points that have much higher balance violation. It seems like contrastive regularization helps, but likely in a way that isn't fully explained by balance violation.
1. You showed that the transformer can solve Dyck without an interpretable attention pattern in the second layer. How do we know it doesn't have an interpretable pattern in the first layer? Do your experiments show that as well?

Other questions:

1. 120: Why is there a difference in loss function between the theory and experiments?
1. Eq 5: What about dropout?
1. 184: Which pumping lemma? For CFGs or DFAs? Can you elaborate on how the proof of Thm 1 resembles that of the pumping lemma?
1. 208: Why can the loss not be optimal?
1. 277: How many random restarts for these experiments did you run?
1. Fig 1: How do you pick the models shown in this figure?
1. 299: It doesn't seem that close to me.

Minor questions and comments from reading:

1. 27: Is it generating "grammars" or "strings"?
1. 31: It's not like a stack, it is a stack. It can be recognized by a one-state real-time DPDA.
1. Finite-precision transformers cannot recognize Dyck up to arbitrary depth.
1. 35: works -> papers
1. 37: I like the list of research questions.
1. 51: Can you elaborate more on the significance of the non-stack-like attention patterns, and what non-stack-like attention is?
1. 62: This finding is really interesting.
1. 69: Interesting!
1. 71: Works -> Work
1. 88: works -> work
1. 95: These brackets are balanced; the problem is that they aren't nested.
1. Eq 1: This could probably be made a bit clearer.
1. Eq 2: This is a bit hard to understand. Can you express it as an equivalent PCFG? The fact that all Dyck prefixes are included complicates this. Why is it important to include all prefixes?
1. Eq 3: What is $e_{w_i}$?
1. Eq 5: $g^{(l)}$ is a full feedforward sublayer, not just a feedforward network, right? The phrasing of this section makes it unclear.
1. 135: What about positional encodings?
1. 138: Is the whole LayerNorm operation set to the identity, or just some of its entries?
1. 196: It would be helpful to point out why the proof of sufficiency is not necessary for the main argument of the paper.
1. 200: Could you please include an intuitive proof sketch of Corollary 1 in the main text?
1. 220: Can you elaborate on how this constraint can be satisfied?
1. 238: With respect to what variables is O(1) constant? How is the l2 norm defined on a matrix?
1. 241: What is the significance of using $1 - \delta$ instead of $\delta$?
1. 242: Make it clearer that \epsilon-approximate refers to the equation below.
1. 243: It might be helpful to point out that this holds true when X is one-hot vectors.
1. 280: Cool!
1. 291: Quantitative is misspelled.
1. 306: Can you include these results?
1. 329: Ungrammatical sentence.

**Limitations:**

Yes.

---

> ### Author Rebuttal · Authors · 2023-08-10
>
> We thank the reviewer for their very detailed read and thoughtful comments!
>
> **M1 and l1 - LayerNorm: post or pre-norm**
>
> In our theoretical setup, we have used a nonstandard order of layernorm and residual link. In real implementation, it is much more common to put LayerNorm after the residual addition, which happens after the output of attention modules is calculated (this is true in both post-norm and pre-norm).
>
> Our choice of architecture is to ameliorate the issue of over-smoothing of the softmax operation in the attention. Namely, it can be proven that if the residual is put before the softmax, there in fact **doesn’t exist any Transformer with a minimal first layer** that can reach zero loss on any length $N$. We will include the proof in the camera-ready version.
>
> In the experiments, we try both our theoretical setups and standard GPT-2 setups and observe similar phenomena and accuracies. We choose to report the result with the standard GPT-2 setups and we will add a discussion on these architectural differences in the camera-ready version.
>
> **M2 - Typo in eq 5**
>
> This is indeed a typo, thank you for spotting this! Please note that our conclusions and proofs are not affected by the typo.
>
> **I2 - Whether positional information can be eliminated completely**
>
> It is not a problem even if the position information is present since later layers can always ignore it if necessary. More importantly, the definition of the minimal first layer is to present the smallest family of solutions, i.e. simplicity is a feature not a bug; please see [G1] in our global response for more discussions.
>
> **I3, I4: Clarification on Theorem 3**
>
> We agree that pruning must encode information in the pruned Transformer and you are correct that the last part is accounting for this observation. You are also correct that the significance is that $T_{\text{large}}$ can approximate two entirely different Transformers. It is also correct that $T_{\text{large},i}$ won’t have similar behavior as the $T_{\text{large}}$ the Transformer before pruning.
>
> We are happy to add some remarks on the take-away message before the formal statement of Thm 3. What we are trying to argue is that if we investigate a single component like $W$ in T_{\text{large},i}$, it is feasible that the same observation of $W$ may correspond to a completely different behavior of the Transformer as a whole. Hence, explaining the functionality of the transformer must take into account the interplay of multiple components.
>
> **I5: What’s the implication of Cor 2? Can lambda be arbitrarily large?**
> Your understanding is correct that Cor 2 means approximate balance still holds under regularization.
>
> Lambda cannot be arbitrarily large: when regularization is added, there is a tradeoff between minimizing the Dyck language modeling loss and minimizing the regularizer. When lambda is too large, the focus is on the regularizer and hence the solution is no longer good for the Dyck language modeling task. For example, in the extreme of $\lambda \to \infty$, all learned parameters converge to 0.
> In contrast, our paper focuses on the more interesting setting of having a reasonably small L2 regularization constant (lambda), so that the original loss landscape is only slightly perturbed. When the original loss has multiple (approximate) optima, the regularizer serves as a tie-breaker by selecting the ones with a smaller norm.
>
> **I6: Can Theorem 1-2 be extended to beyond 2 layers?**
>
> Yes, please refer to [G1] in our global response.
>
> **I7: line 237: Width refers to the MLP width.**
>
> **I8: line 279: Yes, this is the correct intuition.**
>
> **I9: contrastive loss doesn’t explain the full story**
>
> Yes, we agree that the effectiveness of contrastive regularization is likely not fully explained by balance violation, and it is interesting future work to understand the training dynamics under various forms of regularization.
>
> **I10: How do we know the first layer doesn't have an interpretable pattern? What do the experiments show?**
> First, we’d like to clarify that our claim is not that “any component in Transformers is not interpretable,” but rather, “some component in Transformers may not be interpretable.” Hence our results do not imply non-interpretability of the first layer. As shown in the appended pdf in general response, the attention pattern of the first layer is close to uniform (loosely aligned with the intuition that the first layer performs counting) and is not very informative beyond this.
>
> **O1 - difference in loss function**
>
> Please see [G3] in the global response.
>
> **O2 - Eq 5: What about dropout?**
>
> We did not include dropout since it affects only the training dynamics, but not the loss optima. For example, it is common practice to disable dropout at inference time.
>
> **O3 - Is the pumping lemma for CFGs or DFAs? How does it relate to the proof of Thm 1?**
>
> Our proof idea is relevant to (but not a direct application of) the pumping lemma for regular languages (i.e. DFAs): although Dyck is context-free, bounded-depth Dyck is regular. The proof idea is to construct a family of prefixes p_m (m = 1,2,3,...), obtained by inserting m pairs of matching brackets to the original prefix p, which is similar to repeating the “middle component” in the pumping lemma.
>
> **O4 - Why can’t the loss be optimal?**
>
> We were referring to the empirical limitation that training dynamics often do not exactly end at an optima in practice, due to finite samples and various sources of randomness. We will clarify this in our camera-ready paper.
>
> **O5 - How many random restarts for these experiments did you run?**
>
> We have 40 runs per setup.
>
> **O6 - Fig 1: How do you pick the models shown in this figure?**
>
> In each setting, we randomly selected models among those that reach at least 97% accuracy.
>
> Thank you very much also for the other comments which will greatly improve the readability of our paper!

---

> > ### Comment · Reviewer_ZsLs · 2023-08-11
> > **Response**
> >
> > Thank you very much for your response.
> >
> > G1, I2, I6, I8, O2: Can you include a more detailed discussion of this point in the paper? The two specific examples you listed in G1 make sense, but the general claim "When moving to more complex architectures, the set of solutions can only grow and complicate interpretability further" is a very strong one, and it requires careful justification. I would recommend inserting a section that clearly lists all of the simplifying assumptions you make about the transformer architecture (e.g. number of layers, details about positional encodings), and explain how the results under these assumptions can still apply to unsimplified, general-case transformers.
> >
> > M1, I1: Thank you for the explanation, and for agreeing to discuss this more in the final version. Doesn't this architectural change significantly limit the applicability of the theory to real transformers? If it's not architecturally possible to construct a minimal layer, don't the main results of Theorems 1 and 2 fall apart?
> >
> > I3, I4: Thanks. I would appreciate it if you rewrote or added to this section in a way that preempts these questions.
> >
> > I5: I see. I was wondering if this Corollary gives us a finite $\lambda$ that breaks this condition, but I think it's weaker than that, because it tells us what happens as it goes to 0 in the limit, right?
> >
> > I10: Got it, makes sense.
> >
> > O1: Great!
> >
> > O3: Thanks. If you have space, could you include this clarification in the paper?
> >
> > O4: Great, thanks.
> >
> > O5: I missed this later in the paper. Thanks!
> >
> > O6: Thanks, this detail is worth mentioning.
> >
> > Typo in Fig 3: leas -> leads

---

> > > ### Author Response · Authors · 2023-08-14
> > > **Thank you for the timely and detailed response!**
> > >
> > > **G1, I2, I6, I8, O2:** Thanks for the helpful advice! We would certainly add the section discussing the choice of architectures. As a further clarification, by “simplicity is a feature not a bug”, we mean that since the set of solutions can only increase when moving to a larger function class, generalizing from minimal-first-layer models to more complex models can only increase the set of non-interpretable solutions, and hence further strengthening our main claim that solutions with non-interpretable attention patterns are abundant.
> > > We will discuss the point more explicitly in our paper, since, as pointed out in the rebuttal, it would be easy to extend these results to more complex structures.
> > >
> > > **M1, I1:** We would like to argue that
> > > 1) While the agenda of this line of research is to make theoretical understanding feasible under increasingly general architectures, the present state of theory is that for many settings, one must make strong assumptions (for example [1,2,3] consider linear attention and [4] considers hard attention). We think our work is a good step towards the above goal, because slightly modifying the position of LayerNorm is less “simplifying” in degree than the assumptions made in the above prior works (linear attention / hard attention).
> > > 2) For the Dyck language considered here, the modified architectures can still learn and generate Dyck to almost perfect accuracy and we observe similarly uninterpretable attention patterns as using standard architectures. Hence we believe it is still valuable to investigate the architecture theoretically in the paper.
> > > 3) We can in fact extend our result (Theorem 1, “sufficient condition” part) to standard architecture and **bounded** input length $N$. We agree that this extended result can be informative about the uninterpretability of attention patterns, and we will include this result in the camera-ready version of the paper.
> > >
> > > References:
> > >
> > > [1] Ahn, Kwangjun, Xiang Cheng, Hadi Daneshmand, and Suvrit Sra. “Transformers Learn to Implement Preconditioned Gradient Descent for In-Context Learning.” arXiv, May 31, 2023.
> > >
> > > [2] Oswald, Johannes von, Eyvind Niklasson, Ettore Randazzo, João Sacramento, Alexander Mordvintsev, Andrey Zhmoginov, and Max Vladymyrov. “Transformers Learn In-Context by Gradient Descent.” arXiv, May 31, 2023.
> > >
> > > [3] Zhang, Ruiqi, Spencer Frei, and Peter L. Bartlett. “Trained Transformers Learn Linear Models In-Context.” arXiv, June 16, 2023.
> > >
> > > [4] Yao, Shunyu, Binghui Peng, Christos Papadimitriou, and Karthik Narasimhan. “Self-Attention Networks Can Process Bounded Hierarchical Languages.” arXiv, March 12, 2023.

---

> > > > ### Comment · Reviewer_ZsLs · 2023-08-14
> > > > **Response**
> > > >
> > > > > since the set of solutions can only increase when moving to a larger function class, generalizing from minimal-first-layer models to more complex models can only increase the set of non-interpretable solutions
> > > >
> > > > How would you prove this in the general case? What if the *proportion* of interpretable vs. non-interpretable solutions can increase?
> > > >
> > > > M1, I1: Thank you, contextualizing this assumption against prior work and the extension to Theorem 1 will be helpful.

---

> > > > > ### Author Response · Authors · 2023-08-17
> > > > >
> > > > > Thank you very much for the discussion and the follow-up questions!
> > > > >
> > > > > We would first like to clarify that our theoretical results do not make a claim on the proportion of interpretable solutions: our results show that multiple solutions exist, some of which are non-interpretable. However, we do not quantify the number of non-interpretable solutions; in fact, it is unclear how such quantification can be done, since to the best of our knowledge, there isn't a universally accepted metric to gauge interpretability.
> > > > >
> > > > > On the other hand, an empirical approach might provide some understanding of the proportion. To this end, our main paper shows results from 2-layer networks, and the PDF attached in our “Global response” additionally provides 3-layer results. As of now, there is no conclusive evidence suggesting that standard training methods lean towards interpretable solutions, and we are not inclined to believe that increasing the network size would increase the proportion of interpretable solutions.
> > > > >
> > > > > We would love to learn about your perspectives on this. If there is a specific model size that you believe would render our empirical results more compelling, we'd appreciate your guidance. Thank you very much!

---

> > > > > > ### Comment · Reviewer_ZsLs · 2023-08-17
> > > > > > **Response**
> > > > > >
> > > > > > I see, thanks for clarifying. My purpose in posing that question stemmed not so much from a desire to quantify the number of interpretable solutions, but to understand the specific constructions that allow you to draw conclusions about more complex models using simpler ones.
> > > > > >
> > > > > > > since the set of solutions can only increase when moving to a larger function class, generalizing from minimal-first-layer models to more complex models can only increase the set of non-interpretable solutions
> > > > > >
> > > > > > Reflecting on it more, I still don't find this earlier statement very satisfying. What I think you're saying is that the set of functions that a more complex transformer can implement is always a superset of that of a simpler transformer -- therefore if a non-interpretable solution exists in the simpler one, it exists in the more complex one. Can you give a proof by construction for this, giving a precise definition of "more complex" models in this context (more layers, different layer norm, etc.)? Maybe the construction is very simple, but it's not obvious to me. Maybe some changes, like layer norm, *delete* non-interpretable solutions. Your theoretical results are on transformers of type B for the purpose of making claims about transformers of type A, and I'm just trying to make sure there's a clear path from B to A.

---

> > > > > > > ### Author Response · Authors · 2023-08-19
> > > > > > >
> > > > > > > Thank you very much for the further clarification!
> > > > > > >
> > > > > > > We’d like to clarify about the construction to larger models, which can have a greater width or depth.
> > > > > > > - For a greater width, our Theorem 1 applies directly, since the construction does not depend on the width.
> > > > > > > - For a greater depth, it suffices to show that additional layers can perform the identity function. To this end, one can utilize the residue link in the Transformer layer and choose the value matrix to be zero and the FFN (with or without residue connection) to be identity. This construction is implicitly assuming LayerNorm will map zero vector to zero vector, which is true for the common PyTorch implementation and for our paper. Also, it is worth noting that this holds for both the architecture we considered in the paper and the standard GPT-2 architecture.
> > > > > > >
> > > > > > > Please let us know if this is clear. Thank you so much again for actively engaging in the discussions!

---

> > > > > > > > ### Comment · Reviewer_ZsLs · 2023-08-19
> > > > > > > > **Response**
> > > > > > > >
> > > > > > > > Thank you. I think all my biggest questions have been addressed, so I'm raising my score.

---

### Official Review · Reviewer_ksf8 · 2023-07-05

**Soundness:** 3 good
**Presentation:** 2 fair
**Contribution:** 3 good
**Rating:** 5
**Confidence:** 3

**Summary:**

The paper studies the methods for interpretability of transformers that focus on individual parts of the model (instead of the entire network) using a synthetic setup of learning a Dyck language. Models that can solve this task need to satisfy the pumping lemma, and this is used to show that the attention pattern of a single layer can be nearly randomized without changing the network functionality. The consequential claim is that interpretability based on individual heads or weight matrices can be incorrect.

**Strengths:**

+ Theoretical characterization shows a perfect balance condition on the attention pattern for 2-layer transformer corresponding to Dyck language. This condition admits non-stack-like attention pattern.

+ As a consequence of lottery ticket hypothesis, it is shown that any transformer can be approximated. by a random larger transformer and thus, interpretability based on local components may not be reliable.

**Weaknesses:**

* The paper appears to have a very narrow focus - examining the claim of stack-like patterns in the attention heads from Ebrahimi et al. Both stack-like and non-stack-like patterns can process Dyck languages. The presented analysis is not a "proof" or very convincing argument that the second is preferred by transformers. The presentation can be shifted to a critical examination rather than non-arguable claims. This is a minor presentation issue.

* The title and initial discussion can emphasize the focus on bounded-depth Dyck language.




**Questions:**

* The fact that we are considering bounded-depth Dyck language makes the classical results from formal automata theory not directly applicable - if I know the maximum depth of Dyck language, does the pumping lemma still apply?

**Limitations:**

There are no concerns with respect to broader societal impacts.

---

> ### Author Rebuttal · Authors · 2023-08-10
>
> Thank you for the comments and questions! We’d like to address the concerns below.
>
> **(W1) The results are not a “proof” that the non-stack-like patterns are preferred by Transformers.**
>
> We’d like to clarify that “proving non-stack-like patterns are preferred” is not the goal of our paper. Instead, our results 1) formally prove that such non-interpretable solutions exist, and 2) empirically demonstrate that the non-interpretable solutions are not merely theoretical artifacts, but trained Transformers frequently converge to such solutions. The implication is that attention patterns alone are not reliable for interpretability. We are not sure if the reviewer had other concerns when referring to “non-arguable claims”; if yes, we’d appreciate it if the reviewer could clarify and we will be happy to discuss.
>
> **(W2) It should be emphasized that the setting is bounded-depth Dyck.**
>
> We interpret this suggestion as emphasizing the “bounded depth” aspect since the title and abstract have both only stated Dyck. Though we do state the bounded depth assumption explicitly in Section 2, Problem Setup, we agree with the reviewer’s suggestion that it will be better to make it clear earlier in the paper—we will modify the abstract and introduction to reflect this suggestion.
>
> **(Q1) Question about the pumping lemma**
>
> By the pumping lemma we are referring to the one for regular languages, which fits our setting since a bounded-depth Dyck language is a regular language (rather than a CFG). We will make this clearer in the future version, thank you for highlighting this!

---

### Official Review · Reviewer_46oE · 2023-07-06

**Soundness:** 3 good
**Presentation:** 3 good
**Contribution:** 2 fair
**Rating:** 5
**Confidence:** 3

**Summary:**

The authors discover interpretability conclusions based off attention weights can be unreliable with a case study on a Dyck language with a shallow transformer. Concretely, the authors show non-distinct attention patterns, even an uniform one can be theoretically learned to solve the Dyck language task which invalidates any interpretability conclusion based off solely on attention weights. The authors also try to prove this from formal theory as well as empirical results. Overall, the paper studies a timely topic and provides some insights, and has some open questions that may be addressed to make the paper more convincing.

**Strengths:**

- The paper overall is easy to follow.
- The research topic about mechanistic interpretability based off attention weights seems like timely, and informs us about future works.
- The formal proof on a shallow two-layer transformer with assumptions seems to valid the point that there are many attention patterns can be learned to solve the task, which makes the conclusions made based off a single set of attention weights less convincing.

**Weaknesses:**

-**Formal proof on a simple setup with assumption is good, but missing connections to larger transformer blocks**

It is great to have a starting point to start looking at the potential attention patterns can be learned to solve structural tasks like Dyck language with a lot of assumptions. But, it limits its connections to larger transformer blocks where there is no assumption about the first layer. After all, multiple stacked transformer blocks approximate high-degree of linearity than a single layer. Can also provide more explanations why their formal proof can be extended to larger transformer blocks?

- **Different Attention Patterns Can Be Learned To Generate Dyck is not convincing enough**

Qualitatively to visualize attention patterns is fine as a start, but quantitatively only measuring attention weights difference among different initialization is bit lacking. It is well-known that, for structural generalization tests like this one, random seeds, learning rate and schedulers can large even affect model's factual behavior, let alone its hidden representations. The fact that attention weights are different is not surprising, and I feel like it is not enough to say different **effective pattern** can be learned to generate Dyck. For instance, if we can fix the attention weights to be uniform and enable gradients on other parameters, and show the model can still solve the task, it would be more convincing, correct me if i am wrong.

- **Balanced attention increases generalizability is a weak evidence about the un-interpretability of attention weights?**

It is interesting to add regularizations to the attention weights can make the model generalizes better. But, by making the attention weights to distribute in a certain pattern can be viewed as a counter argument about the un-interpretability. Making them more balance is an interpretable mechanism to enforce; since it increases model performance, which means that interpretable attention distribution is somewhat better than randomly optimized ones. So, when performance is good, attention weights are better to be interpretable? I think some clarifications are needed here to connect to the main points of the paper.

- **(Un)interpretability of Transformers is a very broad claim**

I think this paper only studies the (Un)interpretability of Transformers' attention weights under a specific use case. So, it is may be better to position the paper to a narrower scope.



**Questions:**

See my comments above.

**Limitations:**

See my comments above.

---

> ### Author Rebuttal · Authors · 2023-08-10
>
> Thank you very much for the detailed reviews and the questions! We are glad that the reviewer appreciates the timeliness of the topic and our formal guarantees. Regarding the reviewer’s concerns, there seem to be some misunderstandings, which we would like to clarify below.
>
> To avoid confusion in our responses, we’d like to first clarify the terms: we will use “(attention) weights” to refer to the model weights (i.e. parameters, such as $W_K$, $W_V$ matrices, or the MLP weights), and use “attention patterns” to refer to the intermediate output $\sigma(XW_K W_Q^\top X^\top)$. Our results show that either attention patterns (Theorem 1,2) or weights (Theorem 3) are not reliable for interpretability.
>
> **(W1) Connection to larger Transformer blocks**
>
> Recall that we are trying to show that there exist more solutions than interpretable ones. Hence, the *simplicity of the model class is a feature and not a bug* (please also see [G1] of our global response): the set of solutions in more complex models (e.g. where the first layer is not necessarily minimal) is a superset of the set of solutions of the simpler models. In other words, we have shown that the solutions are not always interpretable even in a very restricted model class, and so they will not always be interpretable in a broader class. Also, our Theorem 3 shows that a small Transformer can be effectively simulated by simply pruning a large random initialized Transformer, which further strengthens our claims.
>
> **(W2) Effective patterns vs random variations**
>
> We agree with the reviewer that there are inevitable variations introduced by hyperparameter choices and various sources of randomness, which do not necessarily correspond to different “effective patterns”. Nevertheless, as the reviewer pointed out, stack-like patterns and uniform patterns seem indeed sufficiently different. The reviewer suggested checking whether uniform patterns solve the task in practice, which is a great point and in fact *has been addressed in the paper*: it is currently stated as the last sentence of the first paragraph of “Sec 4.1 - Qualitative result”; will highlight this fact more in the camera-ready version. Please also see Appendix B.1 for additional discussions.
>
> **(W3) Does “improved balance improves length generalization” violate the uninterpretability premise?**
>
> There seems to be a misunderstanding about the balance condition: the balance condition characterizes the optimal solutions (which exhibit perfect out-of-distribution generalization), and it doesn’t affect whether a solution is interpretable or not. For example, **both stack-like and uniform attention patterns satisfy the balance condition**, but the uniform attention is not interpretable.
>
> **(W4) (Un)interpretability of Transformers is a broad claim**
>
> We agree with the reviewer that we should be cautious about the naming. In both the abstract and introduction, we have tried to be cautious about the types of interpretability methods our paper is tackling. Furthermore, we discuss in the related work section (Appendix A) that our results do not contradict all methods of mechanistic interpretability. We are happy to make concrete modifications that you think would improve the clarity of the scope of our paper.
>
> We are happy to modify the title to hint at the types of methods we consider: for example, we would suggest “Myopic (Un)interpretability of Transformers: a case study with Dyck grammars”, to indicate the fact that we specifically focus on techniques based on inspecting individual components (either attention patterns or MLP blocks).

---

> > ### Comment · Reviewer_46oE · 2023-08-18
> > **Thanks for your response and I have raised my score.**
> >
> > Thanks for the response. It answers some of concerns. I raised my rating slightly. For the future versions, I would like to see more tasks such as parity. I think it will greatly strengthen the claim with other tasks, especially on those fixed attention weights settings.

---

> > > ### Author Response · Authors · 2023-08-18
> > > **Thank you for the acknowledgement and suggestions!**
> > >
> > > We are glad that our responses helped address some concerns, and thank you for the suggestion of parity!
> > >
> > > We are happy to include parity experiments in our rebuttal, and can reach out to the AC to inquire about the possibility of adding figures if needed.
> > > For clarity, we'd like to confirm the experiments you find most persuasive: recall that parity can be solved by 1 Transformer layer with uniform attention, and in the case of parity, a uniform attention is actually the “interpretable” attention pattern (unlike Dyck). For the experiments:
> > > - Would you find it compelling if we demonstrated whether a 1-layer Transformer with random initialization can converge to patterns that are non-uniform and possibly uninterpretable?
> > > - Would you like to see if a 1-layer Transformer, when set to have uniform attention patterns, can effectively solve parity?
> > >
> > > We greatly appreciate your guidance, thank you!

---

> > > > ### Author Response · Authors · 2023-08-21
> > > > **Additional results on parity**
> > > >
> > > > Thank you again for the suggestion for the parity task! We have run additional experiments with 1-layer Transformers, since parity can be solved by a 1-layer Transformer using uniform attention. The experiments are done in two settings: 1) train all parameters; 2) fix a uniform attention and train the rest of the parameters.
> > > > - For the first setting where all parameters are optimized together, we observe optimization issues: among 40 runs, the best run gets 85.6% accuracy and the mean accuracy is below 60%. Such optimization challenges are consistent with the findings in [1], [2].
> > > > - On the other hand, freezing the attention to be uniform (by fixing $W_k$ to 0) helps alleviate the optimization issues, where the best accuracy is 99.9% and the mean accuracy is 92.2%.
> > > >
> > > > In summary, we found that similar to Dyck, it is also the case for parity that a particular theoretical construction may not be found in practice (i.e. the solutions found depend heavily on optimization) and that theoretical insights help yield better results (e.g. the balanced condition for Dyck, or a uniform attention for parity).
> > > >
> > > > Please let us know if there are other questions and suggestions, thank you very much!
> > > >
> > > > [1] Satwik Bhattamishra, Kabir Ahuja, Navin Goyal. On the Ability and Limitations of Transformers to Recognize Formal Languages.
> > > >
> > > > [2] Bingbin Liu, Jordan T. Ash, Surbhi Goel, Akshay Krishnamurthy, Cyril Zhang. Transformers Learn Shortcuts to Automata.

---

### Official Review · Reviewer_QYzn · 2023-07-06

**Soundness:** 3 good
**Presentation:** 2 fair
**Contribution:** 3 good
**Rating:** 6
**Confidence:** 4

**Summary:**

Recently several efforts have been made to understand the exact mechanism that a trained Transformer uses to make its predictions. A number of those works rely on analysing the attention patterns or other components of Transformers. Broadly, this paper argues that in some cases it may not be possible to identify the mechanism used by Transformers by analysing individual components such as attention patterns.

They confine their analyses to Dyck languages with bounded depths -- which is the task of determining whether a sequence of brackets is well-balanced. For a two-layer Transformer which is expressive enough to recognize Dycks, they first show a condition for attention operation which is necessary and sufficient to construct a Transformer that can generate Dycks optimally (or almost optimally). The first layer in their construction is something called a minimal first layer which they define and then they show that if the attention in the second layer satisfies certain properties then one could construct the ReLU FFNs in such a way that the 2-layer Transformer can generate Dyck and obtain near-optimal mean squared error. The result essentially implies that there exists multiple Transformers with different attention patterns that can generate Dyck near-optimally including uniform attention and therefore attention patterns may not necessarily be informative of the mechanism used by the network.

They then show that Transformers with constant normed weight matrices can be approximated by pruning randomly initialized large Transformers (which have 4x layers and width). Lastly, they conduct experiments with Transformers trained on bounded Dyck languages. They compare the attention patterns of trained Transformers with different initializations and find that there is significant variance in the attention pattern on a string from the Dyck language.

**Strengths:**

(S1) **Interesting theoretical results.** In my opinion both Theorem 2 and 3 are interesting to some degree. The fact that Transformers with various types of attention patterns and FFNs can solve the same problem in almost optimal manner is important to note while working on interpretability.

Dyck languages are also very important from a formal language theory perspective given the Chomsky–Schützenberger representation theorem -- which states that every context-free language can be represented by a regular and a Dyck language. It may be useful to readers to point this somewhere like the introduction.

(S2) **Well-motivated problem.** Recently, mechanistic interpretability as well as experiments with Dyck languages have been the subject of various works, and these results could be useful to researchers working in the area. Although there have been several works in the past regarding the unreliability of attention for interpretability, the results in this paper provide more concrete evidence for a more well-defined task which could be useful.

(S3) **Length generalization experiment.** I also found the experiments with more balanced attention in Section 4.2 somewhat interesting. Although it is for a specific case, it is interesting to see that insights from sections are directly applicable to improve the length-generalization performance.

**Weaknesses:**

(W1) **Implications of theoretical results.** An important thing to note is that the results prove the existence of Transformers with various attention patterns (and other components) that can solve the same task in a near-optimal way. It does not imply that Transformers with different initializations and hyperparameters can converge to each one of them uniformly at random (or a similar distribution).

To be clear, the authors make no such claims as well. The reason I think it is important to note is that similar evidence and arguments regarding generalization have been incorrect before. The fact that deep learning models have a large capacity (VC dimension, Rademacher complexity) which indicates that networks with various different parameterizations can fit the training set perfectly. This could be seen as an argument for the fact that it is not necessary for a network to converge to parameters that will generalize well. However, in practice when neural networks are initialized in a certain way and trained with GD, then they often converge to one that generalizes well and further work was needed to theoretically understand that. Similarly, I think it is important to keep in mind that even if Transformers can solve a task in many different ways, it is still possible that it is not the case when trained with gradient-based methods in practice. I think it could be useful to clearly explain this in the paper so that the reader does not infer incorrect conclusions.


(W2) **Robustness of Experiments.** I do not think W1 is a major weakness or is needed to be immediately addressed, but because of the reasons mentioned in W1, I believe the empirical results are quite important. However, the experiments do not seem sound and rigorous enough to draw conclusions in a robust manner. For quantitative experiments in Section 4.1, is the average attention variation computed for just one input sequence "[[[[]]]](((())))" from Dyck-(2,4)? Let me know if I have interpreted it incorrectly. However, if it is the case, then it would be useful to conduct more extensive experiments indicating attention variation across multiple sampled inputs and for different Dyck languages such as Dyck-(2, 8), Dyck-(3, 24), etc. The number of random initializations (40) also seems quite small to me. It could be useful to see how the average attention variation is affected by the width and depth of the model. It should not be very computationally expensive to compute average attention variation across a large number of inputs (since it requires only inference) and the training time for these tasks should not be huge since it involves only synthetic data.


I am happy to hear the author's response and change my score depending on that.

**Questions:**

(Q1) MSE vs cross entropy: Is there any particular reason why the theoretical results are with mean-squared error and the experiments are with cross-entropy?
(Q2) Dyck-1: Do the results hold for Dyck-1 as well? Previous works have shown how to interpret Transformers [1] and LSTMs [2] trained on Dyck-1. For languages like Dyck-1, a stack-like mechanism is not necessary and these works show that Transformers and LSTMs learn to do it with a counting mechanism. However, since bounded Dyck languages are regular, it is not necessary to emulate stack-like mechanism to recognize bounded Dycks as well. Maybe it could be useful to conduct experiments with Dyck-1 and see if the mechanism is robust. The task setup in [1][2] is different from the one considered in this paper.



[1] On the Ability and Limitations of Transformers to Recognize Formal Languages.
[2] LSTM networks can perform dynamic counting.

---

> ### Author Rebuttal · Authors · 2023-08-10
>
> Thank you very much for your thoughtful comments and suggestions! We appreciate it that you consider our results as well motivated and theoretically interesting. Thank you also for the suggestion on Chomsky–Schützenberger representation theorem which will help better motivate the Dyck setting, and we will update the revised version accordingly.
>
> **(W1) The existence of multiple optima doesn’t mean SGD can find them.**
>
> This is a great point and we completely agree with the reviewer on this. This is also one of the motivations of our work: despite the existence of the stack-like construction given by prior works, in practice it is not common for the model to converge to such a solution, as shown in Figure 1.
>
> Our results extend previous results by providing a much wider set of potential solutions, and we do not think it is probable that each solution would be uniformly likely to be found via standard optimization. As the reviewer pointed out, an interesting next step is to study how the “inductive bias” given by the synergy of the optimization algorithm and the architecture affects the solutions found, which we hope to explore in the future. By showing the existence of qualitatively very different minima, our result is a meaningful first step towards the future work of understanding which of these minima are more likely to be obtained through the training process.
>
> In our paper, we also empirically verified that transformers trained using SGD/Adam do learn qualitatively different solutions. In particular, with common training setups or regularization approaches, there is no implicit bias towards learning the “more interpretable” attention patterns that prior works theoretically constructed. These observations support our overall claim.
>
> We will update our camera-ready paper to clarify this point.
>
>
>
> **(W2) Robustness of the experiments**
>
> Thanks for the excellent suggestions! We have conducted more experiments according to the reviewer’s advice.
> In our current experiments, we perform experiments over bracket types $k$ in [2,3], grammar depth $m$ in [4,8], model layers in [2,3] and we calculate the attention variations on 40 random sampled sentences following the same distribution as training data (fixed set across the random seeds). We currently have $40$ random runs due to time constraint but we will add more runs in the future versions. For the detailed data, please see the appended pdf.
> The takeaway from our result is that:
> - The choice of $k$ and $m$ in the regime of ablation study doesn’t have a huge impact of attention variation or the interpretability of attention patterns. We still observe vastly different attention patterns across runs with different seeds.
> - There exists a consistent trend that the attention variation will increase monotonously with layer. The attention patterns on first layer remain close to uniform for most runs while there could be vast differences between patterns on higher level of the models.
>
> We are still running some experiments on models with different width and are happy to discuss the results after the experiments finished.
>
> **(Q1) MSE vs cross-entropy**
>
> Thanks for the question. The theory actually doesn’t depend on the type of loss as the results characterize models at/near the minimum of the training loss. We will update our results in the following version to make this clear.
>
> **(Q2) Dyck-1**
>
> Dyck-1 is an interesting setup that has been extensively discussed in prior works, though it is also an easy enough task, so that the next-word prediction task is solvable using only one layer of uniform attention (i.e. counting). This was pointed out in [1] as well. In contrast, our Theorems 1 and 2 are stated for second layer attention, and hence are not applicable to Dyck-1.
>
> [1] Satwik Bhattamishra, Kabir Ahuja, Navin Goyal. On the Ability and Limitations of Transformers to Recognize Formal Languages.

---

> > ### Comment · Reviewer_QYzn · 2023-08-18
> >
> > Dear Authors,
> >
> > Thank you for the clarifications and additional experiments! My primary concern was regarding the robustness of the experiments which are addressed to some degree. As discussed, it would be helpful to discuss (W1) in the paper.
> >
> > In light of the clarifications, I have increased my score.

---

### Author Rebuttal · Authors · 2023-08-10

# Global response
We thank all reviewers for their time and effort in reviewing our paper, and for providing so many useful suggestions that will greatly improve the clarity and impact of our paper. Below we’d like to discuss some questions raised by multiple reviewers.

## [G1] Simplified assumptions: simplicity is a feature not a bug
Reviewer 46oE and ZsLs were curious about whether our conclusions or theorems can transfer to more complex Transformers. The answer is yes: the main conclusion of our paper is that interpretability based on a single Transformer component (e.g. an attention pattern or an MLP block) can be unreliable, since the set of optimal solutions can give rise to a large set of attention patterns and pruned MLP weights. When moving to more complex architectures, the set of solutions can only grow and complicate interpretability further, hence our main conclusion still stands.
For example, even though Theorem 1 & 2 are stated for 2-layer Transformers only, the constructed solutions can be trivially extended to multiple layers by e.g. letting the higher layers perform the identity function. Note also that Theorem 3 is already stated for arbitrary number of layers and hence no generalization is needed.

## [G2] Scope of the paper
Reviewer 46oE and ksf8 suggested making the scope of the paper clearer by potentially modifying the title, abstract or the introduction, while agreeing that the setup has been clearly stated in later part of the paper (Section 2). In particular, it is suggested that we should highlight:
(Reviewer 46oE) the fact that we focus on interpretability methods based on individual components;
(Reviewer ksf8) the boundedness of the grammar depth.
We appreciate these suggestions, and propose to:
Update the title to be “Myopic (Un)Interpretability” and clarify in the abstract/introduction that our lower bounds do not apply to methods that study the model as whole, e.g. many mechanistic interpretability methods.
Update the abstract and introduction to clarify that we are working with Dyck languages with a bounded recursion depth.

## [G3] The choice of loss function
In our submitted version, we used different loss functions for theory (MSE) and in practice (cross entropy). As Reviewer QYzn and ZsLs correctly reminded, this discrepancy is in fact not necessary. We will update the camera-ready version to remove this difference.


## Additional experiments
we also provided additional empirical results per Reviewer ZsLs and QYzn’s suggestion. The figures and quantitative results are provided in the attached PDF and will be included in the appendix of the camera-ready version, and the details are deferred to the responses to individual reviewers.

---

### Decision · Program_Chairs · 2023-09-21

**Decision:**

Accept (poster)

**Comment:**

This submission is a case study aiming to illustrate challenges in recent approaches to studying the interpretability of transformers. More precisely, it both theoretically and empirically illustrates that generalizing solutions to the problem of generating bounded-depth Dyck languages can have unintuitive structure. For example, a generalizing solution can employ a uniform attention mask (rather than one that reflects the matching of opening/closing brackets), showcasing that interpretability approaches focused on studying attention patterns in isolation are insufficient.

The reviewers agree that this is a timely paper that can and should influence future research on interpretability of Transformers. The mix of theoretical grounding combined with practical experiments was received positively, indicating that the paper could have robust impact.

Concerns were raised about the experiments (QYzn, 46oE), which the authors addressed in their rebuttal by providing additional data. Finally, reviewers raised a number of issues around the presentation of the results: smaller issues around clarity of the core paper, and a larger issue about the framing of the contribution. The authors have agreed to shift the framing slightly, and were receptive to feedback and clarification questions, so that I expect a future version of the paper to be substantially more polished.

Overall, I believe that this is an interesting paper that deserves presentation and discussion at NeurIPS.